# Near-Optimality of Contrastive Divergence Algorithms

**Pierre Glaser**     **Kevin Han Huang**     **Arthur Gretton**
Gatsby Computational Neuroscience Unit, University College London
pierreglaser@gmail.com, han.huang.20@ucl.ac.uk, arthur.gretton@gmail.com

## Abstract

We perform a non-asymptotic analysis of the contrastive divergence (CD) algorithm, a training method for unnormalized models. While prior work has established that (for exponential family distributions) the CD iterates asymptotically converge at an $O(n^{-1/3})$ rate to the true parameter of the data distribution, we show, under some regularity assumptions, that CD can achieve the parametric rate $O(n^{-1/2})$. Our analysis provides results for various data batching schemes, including the fully online and minibatch ones. We additionally show that CD can be near-optimal, in the sense that its asymptotic variance is close to the Cramér-Rao lower bound.

## 1   Introduction

Describing data using probability distributions is a central task in multiple scientific and industrial disciplines [1, 2, 3]. Since the true distribution of the data is generally unknown, such a task requires finding an estimator of the true distribution among a model class that best describes the available data. An estimator can be characterized at multiple levels of granularity: at the highest level lies *consistency* [4], a property which states that as the number of available data points increases, a given estimator will converge to the one best describing the data distribution. At a lower level, a consistent estimator can be further characterized by its convergence rate, a quantity upper–bounding its distance to the true distribution as a function of the number of samples. A convergence rate can be either *asymptotic*, e.g. hold only in the limit of an infinite sample size, or *non-asymptotic*, in which case the rate also holds for finite sample sizes. In their simplest form, convergence rates are provided in big–$O$ notation, discarding finer grained information such as asymptotically dominated quantities as well as multiplicative constants. These constants play a role in the so–called *asymptotic variance* of the estimator, which is a precise descriptor of an estimator's statistical efficiency. Convergence rates and asymptotic variances have been the subject of extensive research in the statistical literature; in particular, well–known lower bounds exists regarding both the best possible (asymptotic) convergence rate of an estimator and its best possible asymptotic variance. These results set a clear frame of reference to interpret individual convergence rates, and are routinely present in the analysis of modern statistical algorithms such as noise-contrastive estimation [5, 6] or score matching [7, 8, 9].

In this work, we focus on cases where (1) the true data distribution admits a density with respect to some known base measure, and (2) the model class is parametrized by a finite-dimensional parameter. In this setting, provided that the true distribution belongs to the model class, a celebrated result in statistical estimation states that the model maximizing the average log-likelihood both achieves the best possible asymptotic convergence rate (called the *parametric rate*) and the best possible asymptotic variance, called the Cramér-Rao bound (see, e.g. [10]). While this result shows that Maximum Likelihood Estimators (MLE) are asymptotically optimal, fitting them is complicated by computational hurdles when using models with intractable normalizing constants. Such *unnormalized models* are common in the Machine Learning literature due to their high flexibility [11, 12]; their weakness however lies in the fact that expectations under these models have no unbiased approximation. For this reason, popular approximation algorithms such as unbiased gradient-based

38th Conference on Neural Information Processing Systems (NeurIPS 2024).

stochastic optimization of the empirical log-likelihood cannot *a priori* be used, as the gradient of the normalizing constant is given by an expectation under the model distribution.

The Contrastive Divergence (CD) algorithm [13] is a popular approach that circumvents this issue by using a Markov Chain Monte Carlo (MCMC) algorithm to approximate the gradient of the log-likelihood. Unnormalized models trained with Contrastive Divergence have been shown to reach competitive performance in high-dimensional tasks such as image [14, 15, 16], text [17], and protein modeling [18, 19], or neuroscience [20]. A consistency analysis of the Contrastive Divergence algorithm is delicate, however: indeed, the *optimization error* e.g. the difference between the estimate returned by CD and the MLE, is likely to be non-negligible as compared with *statistical error* – the distance between the MLE and the true distribution – and thus cannot be discarded, as often done when analyzing estimators that minimize tractable objectives [8, 5]. Recent work [21] elegantly established asymptotic $O(n^{-1/3})$–consistency of the CD estimator for unnormalized exponential families when using only a *finite* number of MCMC steps. Key to their argument is the fact that the bias of the CD gradient estimate decreases as iterates approach the data distribution. However, as noted by the authors, their work left open the question of whether and under what conditions CD might achieve $O(n^{-1/2})$–consistency.

**Contributions**    In this work, we answer this question by providing a non-asymptotic analysis of the CD algorithm for unnormalized exponential families. While existing convergence bounds [21] were derived for the "full batch" setting, where the CD gradient is estimated using the full dataset at each iteration, our analysis covers both the online setting (where data points are processed one at a time without replacement), and the offline setting with multiple data reuse strategies (including full batch).

In the online case (Section 3), we show, under a restricted set of assumptions compared to Jiang et al. [21], that the CD iterates can converge to the true distribution at the parametric $O(n^{-1/2})$ rate. Our analysis reveals that CD contains two sources of approximation: a bias term, and a variance term. These sources are almost independent of each other, in the sense that decreasing the bias by increasing the number of MCMC steps will not decrease the variance. The impact of these two sources of approximation transparently propagates in our resulting bounds: in particular, as the bias of the CD algorithm goes to 0, our bounds recover well-known results in online stochastic optimization [22]. Finally, we study the asymptotic variance of an estimator obtained by averaging the CD iterates, a classic acceleration technique in stochastic optimization [23]. We show that provided that the number of steps $m$ is sufficiently large, the *asymptotic* variance of this estimator matches (up to a factor 4) the Cramér-Rao bound.

Next, we study the offline setting (Section 4), where the CD gradient is estimated by reusing (potentially random) subsets of a finite dataset. We show that a similar result to the online setup holds, up to an additional correlation term that arises from data reuse, and present several approaches to control this term. We improve over the results of [21] by showing a non-asymptotic and near-parametric rate at $O((\log n)^{1/2} n^{-1/2})$ under their conditions, and also illustrate how different rates can be obtained under a variety of conditions. Our results also show an interesting tradeoff between the effect of initialization and the statistical error as a function of batch size.

In summary, we establish the near–optimality of a variety of Contrastive Divergence algorithms for unnormalized exponential families in the so called "long-run" regime, where the number of MCMC steps is high enough to ensure that the CD gradient bias is sufficiently offset by the convexity of the negative log-likelihood.

## 2    Contrastive Divergence in Unnormalized Exponential Families

**Unnormalized Exponential Families** Exponential families (EF) [24, 25] form a well-studied class of probability distributions, given by

$$p_\psi(\mathrm{d}x) := e^{\psi^\top \phi(x) - \log Z(\psi)} c(\mathrm{d}x), \quad Z(\psi) := \int_{\mathcal{X}} e^{\psi^\top \phi(x)} c(\mathrm{d}x). \tag{1}$$

Here, $\mathcal{X} \ni x$ is the *data* or *sample space*, which we set to be a subset of $\mathbb{R}^d$ for some $d \in \mathbb{N}^*$, although our results are readily extendable to more general measurable spaces. $c$ is a measure on $\mathcal{X}$ called the *base* or *carrier* measure. When $\mathcal{X} \subseteq \mathbb{R}^d$, $c$ is often set to be the corresponding Lebesgue measure. $\psi \in \Psi \subseteq \mathbb{R}^p$ is a finite-dimensional parameter called the *natural parameter*, and $\phi : \mathbb{R}^d \longmapsto \mathbb{R}^p$ is a function called the *sufficient statistics*, which, alongside with the base measure, fully describes an exponential family. Finally, $\log Z(\psi)$, the *log–normalizing* (or *cumulant*) function, is a quantity

ensuring that $p_\psi$ integrates to 1 over $\mathcal{X}$. Crucially, we will not assume that $\log Z(\psi)$ admits a closed form expression for all $\psi$. The latter fact provides the practitioner with a great deal of flexibility in designing the model class: indeed, the only requirement that should be satisfied prior to performing statistical estimation is to have $Z(\psi) < +\infty$ for all $\psi$, something that can be readily verified and is often the case in practice. The drawback of unnormalized EFs is the fact that sampling (and thus approximating expectations under the model) cannot usually be performed in an unbiased manner. Instead, inference in unnormalized EFs is often performed using tools from the Bayesian Inference literature, such as MCMC [26]. Unnormalized EFs belong to the larger class of *unnormalized models* [27, 28, 7, 6], of the form $e^{-E_\psi(x)-\log Z(\psi)}c(\mathrm{d}x)$, $Z(\psi) = \int e^{-E_\psi(x)}c(\mathrm{d}x)$, for some parametrized function $E_\psi : \mathbb{R}^d \longmapsto \mathbb{R}$ referred to as the *energy*. Unnormalized models thus take the flexibility of unnormalized EFs one step further by allowing the (negative) unnormalized log–density to be an arbitrary function $E_\psi$ of $x$ and $\psi$, instead of requiring a linear dependence on $\psi$ as in Equation 1. We focus in this work on unnormalized EFs due to the multiple computational benefits they provide, as explained in the next section, but we believe that extending our analysis to more general unnormalized models is an interesting avenue for future work.

**Statistical Estimation in Unnormalized Exponential Families using Contrastive Divergence** We now review the Contrastive Divergence algorithm, an algorithm used to fit unnormalized models, and our main object of study in this work. The general setting is the following: we assume access to $n$ i.i.d. samples $(X_1, \ldots, X_n)$ drawn from some unknown distribution $p^\star$, which we assume belongs to $\mathcal{P}_\psi$, e.g. $p^\star = p_{\psi^\star}$ for some $\psi^\star \in \Psi$. Given these samples, we aim to perform *statistical estimation*, e.g. find a parameter $\psi_n$ within $\Psi$ that should approach $\psi^\star$ as $n$ grows.

The starting point of the Contrastive Divergence algorithm is the unfortunate realization that Maximum Likelihood Estimation, which corresponds to minimizing the cross-entropy $\mathcal{L}(\psi) := -\mathbb{E}_{p_n} \log \mathrm{d}p_\psi/\mathrm{d}c$ between the model $p_\psi$ and the empirical data distribution $p_n := 1/n \sum_{i=1}^n \delta_{X_i}$, cannot be performed using exact (possibly stochastic) gradient-based optimization, as the gradient $\nabla_\psi \mathcal{L}(\psi)$ of $\mathcal{L}$ with respect to the parameter $\psi$ contains an expectation under the model distribution $p_\psi$. Indeed, the cross entropy and its gradient are given by

$$\begin{cases} \mathcal{L}(\psi) & = -\frac{1}{n} \sum_{i=1}^n \phi(X_i)^\top \psi + \log Z(\psi) \\ \nabla_\psi \mathcal{L}(\psi) & = -\frac{1}{n} \sum_{i=1}^n \phi(X_i) + \mathbb{E}_{p_\psi} \phi. \end{cases} \tag{2}$$

The second line follows from the well known identity $\nabla_\psi \log Z(\psi) := \mathbb{E}_{p_\psi} \phi$; we refer to [25, Proposition 3.1] for a proof. The Contrastive Divergence algorithm circumvents this issue by running approximate stochastic gradient descent (SGD) on $\mathcal{L}$, where the intractable expectation in $\nabla_\psi \log Z$ is estimated using an MCMC algorithm initialized at the empirical data distribution. In more details, given a number of epochs $T$, a sequence of data batches $B_{t,j}$ of size $B$ (e.g. $B_{t,j} \in [\![1, n]\!]^B, 1 \leq t \leq T, 1 \leq j \leq N\lceil n/B \rceil$), and a family of *Markov kernels* $\{k_\psi, \psi \in \Psi\}$ each with invariant distribution $p_\psi$, at the $j^{th}$ minibatch of epoch $t$, $\nabla_\psi \log Z(\psi_{t,j-1})$ is approximated by $\frac{1}{B} \sum_{i \in B_{t,j}} \phi(\tilde{X}_i^m)$, where $\tilde{X}_i^m$ is produced by running the recursion $\tilde{X}_i^k \sim k_{\psi_t}(\tilde{X}_i^{k-1}, \cdot)$, $\tilde{X}_i^0 = X_i$ up to $k = m$. Throughout the paper, we will refer to the conditional distribution of $\tilde{X}_i^m$ given $X_i$ as $k_\psi^m(X_i, \cdot)$. The resulting gradient estimate arising from combining this approximation with the other (tractable) sum over the data samples present in $\nabla_\psi \mathcal{L}(\psi)$, which we refer to as the *CD gradient* and denote as $h_t$, is thus

$$h_{t,j} := \frac{1}{B} \sum_{i \in B_{t,j}} \phi(X_i) - \frac{1}{B} \sum_{i \in B_{t,j}} \phi(\tilde{X}_i^m) = \frac{1}{B} \sum_{i \in B_{t,j}} \left( \phi(X_i) - \phi(\tilde{X}_i^m) \right). \tag{3}$$

Key to the behavior and analysis of the CD algorithm is the strategy employed to generate minibatches $B_{t,j}$. The case where $T = 1$, $B = 1$, and $B_{1,j} = \{j\}$ will be referred to as *online* CD, while the variant where $T > 1$, and each batch $B_{t,j}$ draws $B$ indices (with or without replacement) from $[\![1, n]\!]$ will be referred to as *offline* CD. In online CD, each data point is present in one and one batch only, while in offline CD, data points are reused across batches. From a statistical perspective, we will see that online CD can be analyzed in a remarkably simple way, while offline CD introduces additional correlations that require care to be controlled. Both settings come with their advantages and drawbacks, as we will see in the next section. The CD algorithms we study will employ decreasing step size schedules $(\eta_t)_{t \geq 0}$ of the form $\eta_t = Ct^{-\beta}$, where $C > 0$ is the initial leaning rate and $\beta \in [0, 1]$. We lay out online CD and offline CD in Algorithms 1 and 2. Note that our algorithms include a projection step on the parameter space $\Psi$ to account for the case where $\Psi$ is compact. In the case $\Psi = \mathbb{R}^p$, this step can be omitted. Next we depart from the setting of [21] and start by analyzing online CD.

**Algorithm 1** Online CD

**Input:** $(X_1, \ldots, X_n) \overset{\text{i.i.d.}}{\sim} p_{\psi^\star}$
**Parameters:** Model class $\{p_\psi, \psi \in \Psi\}$, Markov kernels $\{k_\psi, \psi \in \Psi\}$, number of MCMC steps $m$, learning rate schedule $\eta_t := Ct^{-\beta}$, $\beta \in [0,1]$, $C > 0$, initial parameter $\psi_0$
**for** $t = 1, \ldots, n$ **do**
    //Approx. sample from $p_{\psi_{t-1}}$
    $\tilde{X}_t^m \sim k_{\psi_{t-1}}^m(X_t, \cdot)$
    $h_t := \phi(X_t) - \phi(\tilde{X}_t^m)$
    $\psi_t \leftarrow \psi_{t-1} - \eta_t h_t$
    $\psi_t \leftarrow \text{Proj}_\Psi(\psi_t)$
**end for**
**return** $\psi_n$

**Algorithm 2** Offline CD

**Input:** $(X_1, \ldots, X_n) \overset{\text{i.i.d.}}{\sim} p_{\psi^\star}$
**Parameters:** Same as Algorithm 1, plus number of epochs $T$, batch size $B$, batching schedule $B_{t,j}$, initial parameter $\psi_{0,0}$
**for** $t = 1, \ldots, T$ **do**
    **for** $j = 1, \ldots \lceil n/B \rceil$ **do**
        $[\tilde{X}_{t,i,j}^m \sim k_{\psi_{t-1}}^m(X_i, \cdot) \text{ for i in } B_{t,j}]$
        $h_{t,j} := \frac{1}{B} \sum_{i \in B_{t,j}} (\phi(X_i) - \phi(\tilde{X}_{t,i,j}^m))$
        $\psi_{t,j} \leftarrow \psi_{t,j-1} - \eta_t h_{t,j}$
        $\psi_{t,j} \leftarrow \text{Proj}_\Psi(\psi_{t,j})$
    **end for**
**end for**
**return** $\psi_T$

## 3 Non-asymptotic analysis of Online CD

### 3.1 Preliminaries and Assumptions

Recall that the chi-squared divergence between two probability measures $p$ and $q$ is defined as: $\chi^2(p,q) := \int(\frac{dp}{dq}(x) - 1)^2 q(dx)$ if $p \ll q$, and $+\infty$ otherwise. Here, $p \ll q$ denotes that $p$ is absolutely continuous with respect to $q$ and $dp/dq$ is the Radon-Nikodym derivative [29] of $p$ with respect to $q$. Let $L^2(p_\psi)$ be the space of square-integrable functions with respect to $p_\psi$. For a function $f \in L^2(p_\psi)$, we define

$$\alpha(f, \psi) = \frac{\left(\int \left(\int (f - \mathbb{E}_{p_\psi} f)(y) \, k_\psi(x, dy)\right)^2 p_\psi(dx)\right)^{1/2}}{\left(\int (f - \mathbb{E}_{p_\psi} f)(x)^2 p_\psi(dx)\right)^{1/2}} \tag{4}$$

which is a measure of how quick a Markov chain with kernel $k_\psi$ mixes, relative to the function $f$ [30]. With these definitions in hand, we now state the assumptions required by our analysis of online CD. These assumptions form a strict subset of the assumptions considered in prior work [21], which required additional regularity and tail conditions on the Markov kernels $k_\psi$.

**Assumption A1.** $\mathcal{P}_\psi$ is a subset of a *regular* and *minimal* [25, Section 3.2] exponential family with natural parameter domain $\mathcal{D} \subseteq \mathbb{R}^p$, $\Psi$ is a *convex and compact* subset of $\mathcal{D}$, and $\psi^*$ lies in the interior of $\Psi$.

**Assumption A2.** There exists a constant $C_\chi > 0$ such that $\chi^2(p_{\psi^\star}, p_\psi) \leq C_\chi^2 \|\psi - \psi^\star\|^2$

**Assumption A3.** $\alpha := \sup\{\alpha(f, \psi), f \in \{\phi_i\}_{i=1}^p \cup \{\phi_i \phi_j\}_{i,j=1}^p, \psi \in \Psi\} < 1$, where $\phi_i$ is the $i$-th component of the function $\phi$, and $\phi_i^2$ is the $i$-th component of the function $x \longmapsto \phi(x)^2$.

A well known property of EFs [25, Proposition 3.1] is that their negative cross-entropy (against any other measure) is $C^\infty$, convex, and strictly so if the exponential family is minimal (meaning that the set of sufficient statistic functions $\phi_i$ are not linearly dependent). Leaving aside the issue of intractable expectations, this convexity suggests that $\mathcal{L}$ can be efficiently minimized using stochastic approximation algorithms [31, 22]. The compactness of $\Psi$ provided by Assumption A1 thus ensures, by the extreme value theorem [32], the existence of finite positive constants $\mu$ and $L$ defined as:

$$\mu := \min_{\psi \in \Psi} \lambda_{\min}\left(\nabla_\psi^2 \mathcal{L}(\psi)\right), \quad L := \max_{\psi \in \Psi} \lambda_{\max}\left(\nabla_\psi^2 \mathcal{L}(\psi)\right), \tag{5}$$

where $\nabla_\psi^2 \mathcal{L}$ is the Hessian of $\mathcal{L}$ with respect to $\psi$. $\mu$ (called the *strong convexity* constant) and $L$ (a bound controlling the smoothness of the problem) play a critical role in the analysis of convex optimization algorithms [31]. While it is possible to obtain convergence rates in non-smooth or non-strongly-convex settings, our analysis follows the spirit of [21] by leveraging the strong convexity of the problem to compensate for the bias introduced by using CD gradients instead of unbiased stochastic gradients.

Assumption A2 allows link variations in distribution space to variations in parameter space, and will be instrumental to control the bias of the CD gradient. Note that since $\chi^2(p_{\psi^\star}, p_\psi) =$

$e^{\log Z(2\psi - \psi^\star) - (2\log Z(\psi) - \log Z(\psi^\star))} - 1$ provided that $2\psi - \psi^\star \in \mathcal{D}$ (see [33, Lemma 1]), we expect Assumption A2 to hold in many cases of interests. On the other hand, the possible exponential scaling of $C_\chi$ w.r.t $\log Z$ suggests that this constant may be large in some instances.

Assumption A3 is a *restricted* spectral gap condition: it guarantees that the time required by the MCMC algorithm to estimate expectations of $\phi$ and $\phi^2$ under $p_\psi$ will be uniformly bounded. This assumption is weaker than the (unrestricted) uniform spectral gap condition of [21], which requires that $\alpha$ controls the convergence rate of *all* functions in $L^2(p_\psi)$. Note that standard results in stochastic analysis [34] guarantee that $\alpha \leq 1$: thus, it only remains to ensure that $\alpha$ is strictly less than 1. Spectral gaps are strongly dependent on two properties of distribution: their tail behavior and their multimodality. While multimodality poses the risk of pushing the constant $\alpha$ close to 1, very heavy tails distributions may not verify the spectral gap condition at all.

## 3.2 Results

### 3.2.1 Parametric convergence of online CD

In this section, we show that under the assumptions stated in Section 3.1, the iterates $\psi_t$ produced by the online CD algorithm described in Algorithm 1 will converge to the true parameter $\psi^\star$ at the parametric rate $O(n^{-1/2})$. To do so, we follow a well known paradigm in convex optimization [22] by deriving a recursion on the quantity $\delta_t := \mathbb{E} \|\psi_t - \psi^\star\|^2$, which will allow, after unrolling, to obtain convergence rates for the iterates $\psi_t$. We aim to characterize precisely the impact of performing CD as opposed to performing online SGD on $\mathcal{L}$, which would consist of replacing the CD gradient $h_t$ of Algorithm 1 by the unbiased (stochastic) gradient, given by:

$$g_t(\psi) := -\phi(X_t) + \nabla_\psi \log Z(\psi) \tag{6}$$

which satisfies $\mathbb{E}\, g_t(\psi) = \nabla_\psi \mathcal{L}(\psi)$. The only stochasticity in $g_t$ comes from the sampling of a single data point $x_t$ from the true distribution, which is unavoidable in the online setting, and we have $\mathbb{E}\|g_t(\psi^\star)\|^2 = \mathrm{Tr}(\mathrm{Cov}_{p_{\psi^\star}} \phi) =: \sigma_\star^2$. $\sigma_\star^2$ plays a key role in the analysis of Stochastic Gradient Descent [22]. We expect that replacing $g_t$ by $h_t$ will introduce two sources of approximation: a bias term coming from using a finite number of MCMC steps $m$, and an *additional* variance term, coming from using a single sample $\tilde{x}_t^m$ to estimate $\nabla_\psi \log Z(\psi_t)$. With that in mind, we derive a recursion on $\delta_t$ in the following lemma.

**Lemma 3.1.** *Let $(\psi_t)_{0 \leq t \leq n}$ be the iterates from Algorithm 1. Denote $\delta_t = \mathbb{E}\|\psi_t - \psi^\star\|^2$, $\sigma_\star = (\mathbb{E}_{p_{\psi^\star}}\|\phi - \mathbb{E}_{p_{\psi^\star}}\phi\|^2)^{1/2}$, and $\sigma_t = (\mathbb{E}_{p_{\psi_t}}\|\phi - \mathbb{E}_{p_{\psi_t}}\phi\|^2)^{1/2}$. Then, under A1, A2 and A3, for all $t \geq 1$,*

$$\delta_t \leq (1 - 2\eta_t \tilde{\mu}_{m,t-1} + 2\eta_t^2 L^2)\delta_{t-1} + 2\eta_t^2 \tilde{\sigma}_{m,t-1}^2 + 4\alpha^{m/2}\eta_t^2 \|\log Z\|_{3,\infty} C_\chi \delta_{t-1}^{1/2} \tag{7}$$

*where $\|\log Z\|_{3,\infty}$ is a constant, $\tilde{\mu}_{m,t} := \mu - \alpha^m \sigma_t C_\chi$, and $\tilde{\sigma}_{m,t} := (\sigma_\star^2 + \sigma_t^2 + 2\sigma_t^2 \alpha^{2m})^{1/2}$.*

Lemma 3.1 is proved in Appendix D.2, which details the form of $\|\log Z\|_{3,\infty}$, a constant that we expect to scale roughly as $dL$. Loosely speaking, this recursion suggests that as the learning rate $\eta_t$ goes to 0, the two terms scaling in $\eta_t^2$ will be negligible, in which case we will have: $\delta_t \leq (1 - 2\eta_t \tilde{\mu}_{m,t-1})\delta_{t-1} < \delta_{t-1}$, yielding convergence of $\delta_t$ to 0. We make these arguments formal in the next theorem. The reader familiar with the convex optimization literature will note the similarities between this recursion and the one derived in [22], which would apply as is to online SGD on $\mathcal{L}$ using $g_t$. The difference between the two recursions is that the roles of the strong convexity constant $\mu$ and the noise $\sigma_\star$ are now played respectively by

$$\tilde{\mu}_{m,t-1} = \mu - \alpha^m \sigma_{t-1} C_\chi \quad \text{and} \quad \tilde{\sigma}_{m,t-1}^2 = \sigma_\star^2 + \sigma_t^2 + 2\sigma_t^2 \alpha^{2m}$$

These two modifications respectively characterize the impact of the bias and the additional variance introduced by the CD gradient. The last term in Equation 7, scaling in $\alpha^{m/2}\eta_t^2\sqrt{\delta_t}$, is a residual higher order mixed term coming from relating the variance of the Markov chain sample $\tilde{x}_t^m$ to $\sigma_t^2$. This term can be easily controlled as done next, and disappears as $m \to \infty$. Investigating the impact of $m$ in the recursion, we notice that as $m \to \infty$, $\tilde{\mu}_{m,t} \to \mu$. As we will see later, this ensures that CD will converge for a sufficiently high $m$. On the other hand, in that same regime, $\tilde{\sigma}_{m,t}$ *does not* converge to $\sigma_\star$, but rather to $(\sigma_\star^2 + \sigma_t^2)^{1/2}$, showing the irreducible impact of the variance term.

While we precisely investigate the impact of the residual variance term in the next section, we now unify $\sigma_\star$ and $\sigma_t$ by introducing

$$\sigma := \sup_{\psi \in \Psi}(\mathbb{E}_{p_\psi}\|\phi - \mathbb{E}_{p_\psi}\phi\|^2)^{1/2}. \tag{8}$$

$\sigma$ is an upper bound on the noise induced *both* by the CD gradient and by the online setup, and was used in prior work [21]. Note that by the properties of $\log Z$, $\sigma^2$ also equals $\sup_{\psi \in \Psi} \operatorname{tr}(\nabla^2_\psi \mathcal{L}(\psi))$, where $\operatorname{tr}(A)$ is the trace of $A \in \mathbb{R}^{p \times p}$, and thus finite by the extreme value theorem. The following theorem is obtained by invoking standard unrolling arguments in the convex optimization literature. In the next result, we use the function $\varphi_\gamma(t)$, defined as $\varphi_\gamma(t) = \frac{t^\gamma - 1}{\gamma}$ if $\gamma \neq 0$, and $\log t$ if $\gamma = 0$.

**Theorem 3.2.** *Fix* $n \geq 1$. *Let* $(\psi_t)_{0 \leq t \leq n}$ *be the iterates produced by Algorithm 1, and define* $\delta_t := \mathbb{E}\|\psi_t - \psi^\star\|^2$. *Moreover, assume that* $m > \frac{\log(\sigma C_\chi/\mu)}{\log|\alpha|}$, *i.e.* $\tilde{\mu}_m := \mu - \alpha^m \sigma C_\chi > 0$. *Then under Assumptions A1, A2 and A3, for* $\eta_t = Ct^{-\beta}$ *with* $C > 0$, *we have:*

$$\delta_n \leq \begin{cases} 2\exp\left(4\tilde{L}C^2\varphi_{1-2\beta}(n)\right)\exp\left(-\frac{\tilde{\mu}_m C}{4}n^{1-\beta}\right)\left(\delta_0 + \frac{\tilde{\sigma}_m^2}{\tilde{L}^2}\right) + \frac{4C\tilde{\sigma}_m^2}{\tilde{\mu}_m n^\beta}, & \text{if } 0 \leq \beta < 1 \\ \frac{\exp(2\tilde{L}^2C^2)}{n^{\tilde{\mu}C}}\left(\delta_0 + \frac{\tilde{\sigma}_m^2}{\tilde{L}^2}\right) + 2\tilde{\sigma}_m^2 C^2 \frac{\varphi_{\tilde{\mu}_m C/2 - 1}(n)}{n^{\tilde{\mu}_m C/2}}, & \text{if } \beta = 1 \, , \end{cases}$$

*where* $\tilde{\sigma}_m = \sigma^2(2 + 2\alpha^{2m}) + \alpha^{m/2}\|\log Z\|^2_{3,\infty}C^2_\chi$ *and* $\tilde{L} = (L^2 + \alpha^{m/2})^{1/2}$. *Consequently, if* $\eta_n = \frac{C}{n}$ *with an initial learning rate* $C > 2\tilde{\mu}_m^{-1}$, *we have* $\sqrt{\delta_n} \leq 2\tilde{\sigma}_m C\sqrt{\frac{\tilde{\mu}_m C}{\tilde{\mu}_m C - 2}}\frac{1}{\sqrt{n}} + o\left(\frac{1}{\sqrt{n}}\right)$.

Theorem 3.2 is proved in Appendix D.3. It shows that the iterates produced by online CD will converge to the true parameter $\psi^\star$ at the rate $O(n^{-1/2})$ provided that the number of steps $m$ is sufficiently large, improving over the asymptotic $O(n^{-1/3})$ rate of [21], while imposing slightly weaker conditions on the number of steps $m$ (see [21, Theorem 2.1]). This proves that online CD can be asymptotically competitive with other methods for training unnormalized models, such as Noise Contrastive Estimation [6], or Score Matching [7]. However, the asymptotic variance of $\psi_t$ (e.g. the multiplicative factor in front of the $O(n^{-1/2})$ term) is likely to be suboptimal, e.g. much larger than the Crámer-Rao bound, given by the trace of the inverse of the Fisher information matrix [25]. Given the statistical optimality of MLE, and the fact that CD in an approximate MLE method, this motivates the further goal or obtaining a CD estimator with near-optimal statistical properties. In the next section, we achieve this goal by showing that averaging the iterates $\psi_t$ will produce a near statistically-optimal estimator, in a sense that we will make precise.

### 3.2.2 Towards statistical optimality with averaging

Polyak-Ruppert averaging [23] is a simple yet surprisingly effective way to construct an asymptotically optimal estimator $\bar{\psi}_n := \frac{1}{n}\sum_{t=1}^n \psi_i$ from a sequence of iterates $(\psi_t)_{0 \leq t \leq n}$ obtained by running a standard online SGD algorithm [22]. As shown in [22], when the objective is the cross-entropy of a model, and assuming the unbiased stochastic gradients are available, averaging yields an estimator $\bar{\psi}$ with the asymptotic variance $\operatorname{tr}(\mathcal{I}(\psi^\star)^{-1})/n$, where $\mathcal{I}(\psi) := \operatorname{Cov}_{p_{\psi^\star}}\phi$ is the Fisher information matrix of the data distribution $p_{\psi^\star}$. $\mathcal{I}(\psi^\star)^{-1}$ being the Cramér-Rao *lower bound* on asymptotic variances of statistical estimators [10], this estimator $\bar{\psi}_n$ is asymptotically optimal. The following theorem shows conditions under which averaging CD iterates can give rise to a near-optimal estimator.

**Theorem 3.3** (Contrastive Divergence with Polyak-Ruppert averaging). *Let* $(\psi_t)_{t \geq 0}$ *the sequence of iterates obtained by running the CD algorithm with a learning rate* $\eta_t = Ct^{-\beta}$ *for* $\bar{\beta} \in (\frac{1}{2}, 1)$. *Define* $\bar{\psi}_n := \frac{1}{n}\sum_{i=1}^n \psi_i$. *Then, under the same assumptions as Theorem 3.2, and assuming additionally that* $m := m(n) > \frac{(1-\beta)\log n}{2|\log \alpha|}$, *we have, for all* $n \geq 1$,

$$\left(\mathbb{E}\|\bar{\psi}_n - \psi^\star\|^2\right)^{1/2} \leq 2\sqrt{\frac{\operatorname{tr}(\mathcal{I}(\psi^\star)^{-1})}{n}} + o(n^{-1/2})$$

*Consequently, we have that* $\limsup_{n \to \infty} n\mathbb{E}(\|\bar{\psi}_n - \psi^\star\|^2) \leq 4\operatorname{tr}(\mathcal{I}(\psi^\star)^{-1})$.

Theorem 3.3, alongside with a statement which includes the asymptotic order of the residual term, is proved in Appendix D.4. It shows that at the cost of an increase in *computational* complexity of the entire algorithm from $O(n)$ to $O(n \log n)$, $\bar{\psi}_n$ will be a near-optimal statistical estimator of $\psi^\star$.

While this increase in complexity emerges from the bias of CD, the additional variance of CD results in an asymptotic variance inflated by a factor of $4$ compared to the Cramér-Rao bound.

Theorem 3.3 concludes our analysis of online CD. Despite their asymptotic near-optimality, the bounds provided for online CD and its averaged version have weaknesses: the online CD iterates are not robust to choices of C. On the other hand, as shown in Appendix D.4, the bound of the averaged iterates contain higher-order terms that could be large in intermediate sample regimes. Next, we show that offline CD, which processes data points multiple times, can alleviate these issues.

## 4 Non-asymptotic analysis of offline CD

In practice, CD gradient approximation schemes are commonly used within an offline stochastic gradient descent (SGD) algorithm, where one is given the full size-$n$ dataset upfront and each update uses some stochastic subset of the data. We study CD under offline SGD with replacement (SGDw), i.e. Algorithm 2 with batches $B_{t,j}$ being i.i.d. uniform draws of size-$B$ subsets of $[n]$, and include SGD without replacement in Appendix B.2. To do so, we follow the setting of prior work on offline CD [21], which established its asymptotic $O(n^{-\frac{1}{3}})$ consistency. We show that by slightly strengthening a moment assumption used in [21], the offline CD iterates converge to the true parameter at a near-parametric $O((\log n)^{\frac{1}{2}} n^{-\frac{1}{2}})$ rate. Our proof proceeds by controlling a "tail probability" term specific to the offline setting which characterizes the strength of the correlations between the offline CD iterates and the training data. While, as we show, the assumptions of [21] provide a tail control sufficient to obtain a near-parametric rate, other strategies are possible to obtain convergence guarantees. In particular, we show that non-asymptotic convergence can be obtained by either (1) relaxing assumptions on the Markov kernel required by prior work, or (2) making a specific mixing assumption the Markov chain.

### 4.1 Background: Asymptotic consistency of offline CD in subexponential settings

Prior work [21] has established *asymptotic* $O(n^{-\frac{1}{3}})$ consistency of the (averaged) offline CD iterates in the full-batch case. We summarize their results and assumptions below.

**Assumption A4.** There exists $\nu \geq 2$ s.t. for all $m \in \mathbb{N}$, there is $\kappa_{\nu;m} < \infty$ s.t.

$$\sup_{x \in \mathcal{X}} \sup_{\psi \in \Psi} \left( \mathbb{E} \big\| \phi(K_\psi^m(x)) - \mathbb{E}[\phi(K_\psi^m(x))] \big\|^\nu \right)^{1/\nu} \leq \kappa_{\nu;m} .$$

**Assumption A5.** There exists some $C_m > 0$ such that, for all $\psi_1, \psi_2 \in \Psi$, $\sup_{x \in \mathcal{X}} \| \mathbb{E}[\phi(K_{\psi_1}^m(x))] - \mathbb{E}[\phi(K_{\psi_2}^m(x))] \| \leq C_m \|\psi_1 - \psi_2\|$.

**Assumption A6.** There exist some $\sigma_m, \zeta_m > 0$ such that, for any $z \in \mathbb{R}^p$ with $\|z\| \leq \zeta_m$, $\mathbb{E}[e^{z^\top (\phi(K_{\psi^*}^m(X_1)) - \mathbb{E}[\phi(K_{\psi^*}^m(X_1))])}] \leq e^{\sigma_m^2 \|z\|^2 / 2}$.

**Theorem 4.1** (Theorem 2.1 of [21])**.** *Assume assumptions A1,A2, A3, A4 (for $\nu = 2$), A5 and A6. Let $\psi_{t,1}$ be the $t$-th iterate of offline CD with full-batch gradient descent and constant stepsize $\eta_t = C$, i.e the iterates produced by Algorithm 2 using $B_{t,1} = [\![1, n]\!]$. Then for any learning rate $C$ and number of Markov kernel steps $m$ satisfying $\mu - \alpha^m \sigma C_\chi - \frac{C}{2}(L + \alpha^m \sigma C_\chi)^2 > 0$, we have, for some $A_m > 0$,*

$$\lim_{n \to \infty} \mathbb{P} \left( \limsup_{T \to \infty} \left\| \frac{1}{T} \sum_{t=1}^{T} \psi_{t,1}^{\text{SGDw}} - \psi^\star \right\| > A_m n^{-\frac{1}{3}} \right) = 0$$

This result shows convergence of the *averaged* full-batch CD iterates to the true parameter in the large $n$ and $T$ limit. As discussed, this result is asymptotic both in $n$ and $T$: the probability of the error exceeding $A_m n^{-\frac{1}{3}}$ goes to 0 as $n \to \infty$ and $T \to \infty$, but at an unknown rate. Moreover, the $O(n^{-\frac{1}{3}})$ does not match the optimal $O(n^{-\frac{1}{2}})$ rate.

### 4.2 Sharpening offline CD bounds in subexponential settings

#### 4.2.1 Non-asymptotic $\tilde{O}(n^{-1/2})$-consistency

As a first result, we show that under the assumptions of [21] (except for a slightly stronger $\nu > 2$ moment assumption in A4), $\psi_{T,N}^{\text{SGDw}}$ in fact achieves a near-parametric rate. The most general version of our result holds for any learning rate schedule of the form $Ct^{-\beta}, \beta \in [0, 1]$, and for offline SGD

with arbitrary batch sizes $B$, with data drawn either with or without replacement across batches. For simplicity, we first present our result assuming full batch ($B = n$, $N = 1$, $\psi_{t,j}^{\text{SGDw}} = \psi_{t,1}^{\text{SGDw}}$ for $t \geq 1$) SGD with constant step sizes $\eta_t = C$, which is the setting of [21]. Analogue bounds holding for the other mentioned batching and step sizes schedules can be found in Appendix B.

**Theorem 4.2.** *Assume the setup of Theorem 4.1, except that Assumption A4 holds for some $\nu > 2$, and that $\tilde{\mu}_m = \mu - \alpha^m \sigma C_\chi > 4CL^2$ . Let $\delta_{t,j}^{\text{SGDw}} := \mathbb{E}\|\psi_{t,j}^{\text{SGDw}} - \psi^*\|^2$. Then, we have:*

$$\sqrt{\delta_{T,1}^{\text{SGDw}}} \leq E_1^{T,1} \sqrt{\delta_{0,0}^{\text{SGDw}}} + C'(p, \nu, m, \Psi) \Big( \frac{\sqrt{\log n}}{\sqrt{n}} + \frac{1}{\sqrt{n}} \Big) \Big( \frac{e^{\frac{\tilde{\mu}_m C}{2}}}{\tilde{\mu}_m C} + \frac{E_2^{T,1}}{L^2 C^2} \Big) \tag{9}$$

*where $E_1^{T,1}$, $E_2^{T,1}$ are functions decreasing exponentially in $T$, and $C'(p, \nu, m, \Psi)$ is a constant in $n, T$. Consequently,*

$$\lim_{T \to \infty} \sqrt{\delta_{T,1}^{\text{SGDw}}} \leq \frac{e^{\frac{\tilde{\mu}_m C}{2}}}{\tilde{\mu}_m C} C'(p, \nu, m, \Psi, \beta) \Big( \frac{\sqrt{\log n}}{\sqrt{n}} + \frac{1}{\sqrt{n}} \Big) .$$

The precise values of all the constants can be found in Theorem B.1 (for $E_1^{T,1}$, $E_2^{T,1}$) and Lemma B.3 (for $C'(p, \nu, m, \Psi, \beta)$), including their expressions for $N > 1$ and $\beta \in [0, 1]$. We comment on the main differences between our result and the one of [21]. First our bound holds for *any* epoch $T$ and number of samples $n$. Second, fixing $n$ but taking $T \to \infty$, the final bound matches the parametric $O(\sqrt{n})$ up to a $\sqrt{\log(n)}$ factor, a significant improvement over the $O(n^{-\frac{1}{3}})$ rate of [21]. Finally, we control an $L_2$ error, which is a stronger control than a high probability bound by Markov's inequality; we hypothesize this is the reason why a slightly stronger moment assumption is required for our setup, compared to the one used for the high probability bound in [21].

Inspecting Equation 9, we notice the presence of two *transient* terms, and a *stationary term*, reminiscent of the structure of upper bound of Theorem 3.2. The transient terms (i.e. the ones containing $E_1^{T,1}$ and $E_2^{T,1}$) vanish exponentially fast in the total number of CD updates $T$. However, unlike in online CD where the number of updates and the number of samples are tied (e.g. $T = n$), these two values are now *decoupled*, and these terms can be made arbitrarily small by increasing the number of gradient steps $T$ without having to collect more samples $n$. The stationary term, which is the only one remaining in the limit of $T \to \infty$, decreases with $n$ at a rate that is independent of hyperparameters like the step size $C$ or the learning rate schedule $\beta$ (see Lemma B.3). In that sense, offline CD compares favorably to online CD, whose rate is sensitive to $\beta$ and $C$, and averaged online CD, whose bound contains higher-order (in $n$) terms which can be large in the moderate $n$ regime. On the other hand, the stationary term in offline CD is asymptotically suboptimal: its rate is larger (while only up to a log factor) than the best-case $O(\sqrt{n})$ one achieved by online CD algorithms, and the leading constant does not match the optimal one.

#### 4.2.2 Proof of Theorem 4.2

The high-level proof of Theorem 4.2 follows a similar strategy as the online one: first, derive a recursion for the quantity $\delta_{t,1}^{\text{SGDw}} := \mathbb{E}\|\psi_{t,1}^{\text{SGDw}} - \psi^*\|^2$, then unroll it explicitly to obtain a final bound on $\delta_{T,1}^{\text{SGDw}}$. The main difference to online CD is the presence of an additional offline-specific correlation between the iterates and the data. We thus break down the proof into three steps: (1) deriving a controllable, uniform-in-time upper bound of the data-iterate correlations, (2) deriving and unrolling a recursion on $\delta_{t,1}^{\text{SGDw}}$ containing this new term, and (3) controlling that term to obtain a final bound on $\delta_{T,1}^{\text{SGDw}}$.

**Step 1:characterizing the data-iterate correlations in offline CD** In offline CD, at each epoch $t \geq 1$, the iterate $\psi_{t-1,1}^{\text{SGDw}}$ and the data samples $X_i$ are correlated: this is because these samples may have been used in previous epochs $t' < t - 1$ to obtain the $\psi_{t',1}^{\text{SGDw}}$, which themselves influenced $\psi_{t-1,1}$. With such correlations, we now have $\mathbb{P}(X_i|\psi_{t-1,1}^{\text{SGDw}}) \neq \mathbb{P}(X_i)$, preventing us from obtaining an unrollable recursion on $\delta_{t,1}^{\text{SGDw}}$ by first marginalizing $X_i$ out to obtain an upper bound of $\mathbb{E}\left[\|\psi_{t,1}^{\text{SGDw}} - \psi^*\|^2 |\psi_{t-1,1}^{\text{SGDw}}\right]$ that only depends on $\|\psi_{t-1,1}^{\text{SGDw}} - \psi^\star\|$, and then marginalizing over $\psi_{t-1,1}^{\text{SGDw}}$ to obtain a recursion as in Lemma 3.1. As this problem would not have occurred had we used "fresh samples" (e.g. i.i.d copies of $X_i$ not present in the training data) to perform our update, the

core of the proof lies in controlling the following quantity:

$$\Delta(\psi_{t,1}^{\mathrm{SGDw}}) := \left\| \frac{1}{n} \sum_{i \leq n} \left( \mathbb{E}\big[\phi\big(K_{i;\psi_{t,1}^{\mathrm{SGDw}}}^m(X_i)\big)\big|\psi_{t,1}^{\mathrm{SGDw}}, X_i\big] - \mathbb{E}\big[\phi\big(K_{i;\psi_{t,1}^{\mathrm{SGDw}}}^m(X_1')\big) \,\big|\, \psi_{t,1}^{\mathrm{SGDw}}\big]\right)\right\|$$

where $X_1'$ is an i.i.d. copy of $X_1$. $\Delta(\psi_{t,1}^{\mathrm{SGDw}})$ is the expected (over the data and iterates) error between a quantity that allows to obtain a recursion (the rightmost term) and the one actually used by offline CD (the leftmost term). To control it, we upper-bound it using a tail decomposition:

$$\mathbb{E}[\Delta(\psi_{t,1}^{\mathrm{SGDw}})^2] \leq \epsilon^2 + (\sup_t \mathbb{E}[\Delta(\psi_{t,1}^{\mathrm{SGDw}})^\nu])^{2/\nu} \sup_t \mathbb{P}(\Delta(\psi_{t,1}^{\mathrm{SGDw}}) > \epsilon)^{\frac{\nu-2}{\nu}} := \varepsilon_{n,m,T;\nu}^{\mathrm{SGDw}}(\epsilon)^2 \tag{10}$$

We invoke an additional assumption to ensure that $(\mathbb{E}[\Delta(\psi_{t,1}^{\mathrm{SGDw}})^\nu])^{2/\nu}$ is finite; in the results of [21], this is automatically implied by assumptions A4 and A6. For simplicity we assume the same bounding constant $\kappa_{\nu;m}$.

**Assumption A7.** There exists $\nu \geq 2$ s.t. for all $m \in \mathbb{N}$, $\kappa_{\nu;m}$ from A4 moreover verifies

$$\sup_{\psi \in \Psi} \left( \mathbb{E}\big\| \phi(K_\psi^m(X_1)) - \mathbb{E}[\phi(K_\psi^m(X_1))]\big\|^\nu\right)^{1/\nu} \leq \kappa_{\nu;m} .$$

Note the similarity of this assumption with assumption A4: the only difference is that $X_1$ is now a random training point instead of an deterministic (arbitrary) one. Ensuring assumption A7 in addition to assumption A4 thus requires controlling a $\nu$-th order moment, instead of all moments as implied by assumption A6.

**Step 2: Deriving and unrolling the recursion on $\delta_{t,1}^{\mathrm{SGDw}}$**    The right-hand side of Equation (10) does not depend on $t$, allowing for the derivation of an "unrollable" recursion on $\delta_{t,1}^{\mathrm{SGDw}}$ and its subsequent unrolling, which is performed in the following theorem. For simplicity, we again assume $\beta = 0$ and $N = 1$ and defer the general case to Theorem B.1 in appendix.

**Theorem 4.3** (Convergence up to a tail control). *Assume A1, A2, A3, A4 and A7. Let $\eta_t = C$ for some $C > 0$, and assume that $\tilde{\mu}_m = \mu - \alpha^m \sigma C_\chi > 4CL^2$ . Then for any $\epsilon > 0$,*

$$\sqrt{\delta_{T,1}^{\mathrm{SGDw}}} \leq E_1^{T,1} \sqrt{\delta_{0,0}^{\mathrm{SGDw}}} + C\left( \varepsilon_{n,m,T;\nu}^{\mathrm{SGDw}}(\epsilon) + \frac{5\sigma + 5\kappa_{\nu,m}}{\sqrt{n}}\right) \left( \frac{e^{\frac{\tilde{\mu}_m C}{2}}}{\tilde{\mu}_m C} + \frac{E_2^{T,1}}{L^2 C^2}\right)$$

*where $\varepsilon_{n,m,T;\nu}^{\mathrm{SGDw}}(\epsilon)$ is defined in Equation* (10).

Note that in the general, non-full batch $B \leq n$ case, $\frac{5\sigma + 5\kappa_{\nu,m}}{\sqrt{n}}$ is replaced by $\frac{5\sigma + 5\kappa_{\nu,m}}{\sqrt{B}}$ (see Theorem B.1). Under our bounds, obtaining consistency thus requires setting $B \equiv B(n) \underset{n \to \infty}{\to} +\infty$.

**Step 3: Controlling the tail probability term**    Theorem 4.3 is just one step away from the final bound of Theorem 4.2: it remains to control the tail term $\varepsilon_{n,m,T;\nu}^{\mathrm{SGDw}}(\epsilon)$. Under the assumptions of [21], minimizing $\varepsilon_{n,m,T;\nu}^{\mathrm{SGDw}}(\epsilon)$ over $\epsilon$ yields the following result:

**Lemma 4.4.** *Assume the setup of Theorem 4.2. Let $n \in \mathbb{N}$ be sufficiently large s.t. $\frac{\log n}{n} < \frac{\sigma_m^2 \zeta_m^2}{p + \nu - 2}$. Denote $r_\Psi$ as the radius of the smallest sphere in $\mathbb{R}^p$ that contains $\Psi$, which is finite under A1. Then*

$$\inf_{\epsilon > 0} \varepsilon_{\nu;n,m,T}^{\mathrm{SGDw}}(\epsilon) \leq$$

$$\left( \frac{3\sigma_m \sqrt{p((\nu-2)p + 2\nu)}}{\sqrt{\nu - 2}} + \kappa_{\nu;m} 2^{\frac{\nu-2}{2\nu}} (r_\Psi)^{\frac{(\nu-2)p}{2\nu}} \left( 1 + \frac{2C_m(\nu-2)^{1/2}}{\sigma_m p^{1/2}((\nu-2)p + 2\nu)^{1/2}}\right)^{\frac{(\nu-2)p}{2\nu}}\right) \frac{\sqrt{\log n}}{\sqrt{n}}.$$

To obtain this result, we control the moment term $(\sup_t \mathbb{E}[\Delta(\psi_{t,1}^{\mathrm{SGDw}})^\nu])$ using A7, and we control the tail probability term $\sup_t \mathbb{P}(\Delta(\psi_{t,1}^{\mathrm{SGDw}}) > \epsilon)$ as in [21, Lemma 3.1] using an union bound, a covering argument and A6. Theorem 4.2 then follows by plugging Lemma 4.4 into Theorem 4.3.

### 4.3   Consistency of offline CD: beyond subexponential tails.

As discussed above, the general unrolling result of Theorem 4.3 holds without the subexponentiality assumption A6; this assumption was only used in Lemma 4.4 to control $\varepsilon_{n,m,T;\nu}^{\mathrm{SGDw}}(\epsilon)$. We now discuss two alternative ways to control this quantity without requiring subexponential tails. The first generalizes the idea of Jiang et al. [21], while the second exploits mixing of the Markov chain $K_\psi^m(x)$ as $m \to \infty$. As before we only state partial results (full batch, $\beta = 0$) and defer the full explicit bounds to Appendix B.3.

**Control via Markov Inequality**    Our first alternative uses Markov Inequality to yield the following.

**Theorem 4.5.** *Assume the setup of Theorem 4.3 and additionally that A5 holds. Then*

$$\inf_{\epsilon>0} \varepsilon^{\mathrm{SGDw}}_{\nu;n,m,T}(\epsilon) \ \leq \tilde{C}(p,\nu,m,\Psi)\, n^{-\frac{(\nu-2)\nu}{2(\nu^2+(\nu-2)p)}}\ ,$$

$$\lim_{T\to\infty} \sqrt{\delta^{\mathrm{SGDw}}_{T,1}} \ \leq \tilde{C}'(p,\nu,m,\Psi)\big(n^{-\frac{(\nu-2)\nu}{2(\nu^2+(\nu-2)p)}} + \frac{1}{\sqrt{n}}\big),$$

*where $\tilde{C}$ and $\tilde{C}'$ are functions whose explicit expressions are given in Lemma B.4 in the appendix.*

In the case $p=1$ and $\nu=3$, the sub-optimal error from Theorem 4.5 reads $O(n^{-3/20})$. Theorems 4.2 and 4.5 reveal that, depending on the tail condition imposed on the noise introduced by the Markov kernel, the convergence rate of offline CD varies: A subexponential tail, as assumed in prior work, in fact leads to near-parametric rate. Meanwhile, consistency can be obtained without assuming subexponentiality, albeit at a sub-optimal rate.

**Control via Markov chain mixing.** Alternatively, notice that $\mathbb{E}[\Delta(\psi^{\mathrm{SGDw}}_{t,1})^2]$ involves an average of $\mathbb{E}\big[\phi\big(K^m_{\psi^{\mathrm{SGDw}}_{t,1}}(X_i)^2\big)\big|X_i,\psi^{\mathrm{SGDw}}_{t,1}\big] - \mathbb{E}\big[\phi\big(K^m_{\psi^{\mathrm{SGDw}}_{t,1}}(X'_1)\big)\big|\psi^{\mathrm{SGDw}}_{t,1}\big]$. When $m\to\infty$, the effect of initialization vanishes, and one may expect the difference to converge to zero. We defer to Lemma B.5 in the appendix to show that, under a $\phi$-discrepancy mixing condition ([35]) with a mixing coefficient $\tilde{\alpha}\in[0,1)$,

$$\inf_{\epsilon>0} \varepsilon^{\mathrm{SGDw}}_{\nu;n,m,T}(\epsilon) \ = O(\kappa_{\nu;m}\tilde{\alpha}^{\frac{(\nu-2)m}{3\nu-2}}) \quad \text{and} \quad \lim_{T\to\infty}\sqrt{\delta^{\mathrm{SGDw}}_{T,1}} = O\Big(\kappa_{\nu;m}\tilde{\alpha}^{\frac{(\nu-2)m}{3\nu-2}} + \frac{\sigma+\kappa_{\nu;m}}{\sqrt{n}}\Big)\ .$$

As $m\to\infty$, this recovers the parametric rate $O(n^{-1/2})$. This alternative convergence guarantee comes at the cost of requiring $m$, the number of Markov chain steps, to grow with the sample size $n$.

**Remark** (Examples).  In our main results (Theorems 3.2, 3.3 and 4.3) and the tail condition for offline SGD (Theorem 4.2), we employed a weaker set of assumptions than those in [21] (except for the mild $\nu>2$ moment assumption in (A4)). Consequently, our results apply to all three examples studied in [21]: A bivariate Gaussian model with unknown mean and random-scan Gibbs sampler, a fully visible Boltzmann machine with random-scan Gibbs sampler, and an exponential-family random graph model with a Metropolis-Hastings sampler.

# 5   Related Work

Central to this paper is the prior work of Jiang et al. [21], which provided a rigorous theoretical foundation to analyze the convergence of full-batch CD, and which we refine. The study of optimization with biased gradient descent has attracted a lot of attention in recent years [36, 37, 38, 39]. These works, while closely connected to ours, analyze algorithms with different implementation choices than the CD algorithm: i.i.d. noise setup [36], or setup where a persistent Markov chain is maintained through the iterations [36, 37, 38, 39]. The latter is akin to a variant of the CD algorithm, called the persistent CD [40]. In contrast, our analysis focus on the CD algorithm that restarts a batch of Markov chains from the data distribution at every iteration. Finally, there is a rich body of work on convergence guarantees for offline multi-pass SGD [41, 42, 43, 44, 45, 46]. A notable difference of our analysis is that we are primarily concerned with statistical errors associated with convergence to the true parameter $\psi^*$ in number of samples $n$, and not the commonly studied convergence rate in number of epochs $T$. Consequently, most of our work for the offline setup goes into handling the correlations that accumulate by reusing data across epochs.

# 6   Discussion

In this work, we provide a non-asymptotic analysis of the Contrastive Divergence algorithms, showing, in the online setting, their potential to converge at the parametric rate and to have near-optimal asymptotic variance, and proving a near-parametric rates in the offline setting, significantly extending prior results. Our results apply to unnormalized exponential families: despite their flexibility, these models only cover log-densities with linear relationships on the model parameters. We believe that extending our results to more general forms of unnormalized models is an important direction for future work.

## Acknowledgments and Disclosure of Funding

All authors acknowledge support from the Gatsby Charitable Foundation.

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

# Supplementary Material for "Near-Optimality of Contrastive Divergence Algorithms"

The supplementary material provides the proofs of the main results of the paper:

Section B states full explicit bounds for the offline CD algorithm.

Section C collects a list of useful tools for our proofs. These include the properties of $\varphi_\gamma$ introduced before Theorem 3.2 in the main text, as well as several contraction and integrability results.

Section D provides the proofs for the online CD algorithm.

Sections E, F and G contain the proofs about the offline CD algorithm and the tail control.

## A  Notations

Throughout the proofs, we will denote by $P_\psi^m$ the following operator from $L^2(p_\psi)$ to itself:

$$P_\psi^m f(x) := \int k^m(x, x') f(x') p_\psi(x') \mathrm{d}x'. \tag{11}$$

Here, $k^m(x, x')$ is the $m$-iterated version of some Markov transition kernel $k_\psi$, e.g.:

$$k_\psi^m(x, x') := \int k_\psi(x, x_1) \ldots k_\psi(x_{m-2}, x_{m-1}) \ldots k_\psi(x_{m-1}, x') \mathrm{d}x_1 \ldots \mathrm{d}x_{m-1}. \tag{12}$$

$\mathrm{Proj}_\Psi : \mathbb{R}^p \longmapsto \Psi$ denotes the projection operator onto the convex set $\Psi$, e.g.

$$\mathrm{Proj}_\Psi(\psi) := \underset{\psi' \in \Psi}{\mathrm{argmin}} \, \|\psi - \psi'\|.$$

We also frequently use the following function, used in standard convex optimization results [22].

$$\varphi_\gamma(t) = \begin{cases} \frac{t^\gamma - 1}{\gamma} & \text{if } \gamma \neq 0 \\ \log t & \text{if } \gamma = 0 \end{cases}$$

which is defined on $\mathbb{R}_+ \setminus \{0\}$.

## B  Additional results for offline SGD

In this section, we provide the full statements on error bounds for SGD with replacement (SGDw), SGD with reshuffling (SGDo) and tail moment bounds, which complement the results in Section 4. Proofs are deferred to Appendix F, which make use of $L_2$ approximation by auxiliary gradient updates derived in Appendix E.

**Notations**  Denote the SGDw iterates by $(\psi_{t,j}^{\mathrm{SGDw}})_{t \in \mathbb{N}, j \leq N}$ and let $X_1'$ be an i.i.d. copy of $X_1$. Throughout the remaining of the appendix, we define given $\epsilon > 0$ and $n \in \mathbb{N}$

$$\vartheta_{n,m,T}^{\mathrm{SGDw}}(\epsilon) := \sup_{\substack{t \in [T] \\ j \in [N]}} \mathbb{P}\left( \frac{\left\| \sum_{i=1}^n \left( \mathbb{E}\left[\phi\left(K_{\psi_{t-1,j}^{\mathrm{SGDw}}}^m(X_i)\right)\Big| X_i, \psi_{t-1,j}^{\mathrm{SGDw}}\right] - \mathbb{E}\left[\phi\left(K_{\psi_{t-1,j}^{\mathrm{SGDw}}}^m(X_1')\right)\Big| \psi_{t-1,j}^{\mathrm{SGDw}}\right] \right) \right\|}{n} > \epsilon \right).$$

$$= \sup_{t,j} \mathbb{P}(\Delta(\psi_{t,j}^{\mathrm{SGDw}}) > \epsilon)$$

and $\vartheta_{n,m,T}^{\mathrm{SGDo}}(\epsilon)$ analogously. Using these notations, we can redefine the quantity $\varepsilon_{n,m,T;\nu}^{\mathrm{SGDw}}(\epsilon)$ in the main as $\varepsilon_{n,m,T;\nu}^{\mathrm{SGDw}}(\epsilon) := \sqrt{\epsilon^2 + \kappa_{\nu;m}^2 \left(\vartheta_{n,m,T}^{\mathrm{SGDw}}(\epsilon)\right)^{\frac{\nu-2}{\nu}}}$.

## B.1 An explicit finite-sample bound for SGDw

In the result below, we write $\delta_{t,j}^{\mathrm{SGDw}} := \mathbb{E}\left\|\psi_{t,j}^{\mathrm{SGDw}} - \psi^*\right\|^2$ and, for a fixed $\epsilon > 0$, the quantity

$$\sigma_{n,T}^{\mathrm{SGDw}} = \varepsilon_{n,m,T;\nu}^{\mathrm{SGDw}}(\epsilon) + \frac{5\sigma + 5\kappa_m}{\sqrt{B}} = \sqrt{\epsilon^2 + \kappa_m^2\left(\vartheta_{n,m,T}^{\mathrm{SGDw}}(\epsilon)\right)^{\frac{\nu-2}{\nu}}} + \frac{5\sigma + 5\kappa_m}{\sqrt{B}}.$$

**Theorem B.1.** *Assume A1 (where $\Psi$ may be non-compact), A2, A3, A4 and A7. Let $\eta_t = Ct^{-\beta}$ for some $\beta \in [0,1]$ and $C > 0$, and assume that $m > \frac{\log(\sigma C_\chi/\mu)}{\log|\alpha|}$ s.t. $\tilde{\mu}_m = \mu - \alpha^m\sigma C_\chi > 0$ as in Theorem 3.2. Then for any $\epsilon > 0$, $\sqrt{\delta_{T,N}^{\mathrm{SGDw}}}$ is upper bounded by*

$$
\begin{cases}
E_1^{T,N}\sqrt{\delta_{0,0}^{\mathrm{SGDw}}} + C\sigma_{n,T}^{\mathrm{SGDw}}\left(\frac{4e^{\frac{\tilde{\mu}_m CN}{(T+1)^{1/2}}}}{\tilde{\mu}_m C} + 2N(1+\tilde{\mu}_m C)^{N-1}\varphi_{\frac{1}{2}-L^2C^2N}(T+1)\,E_2^{T,N}\right) \\
\hfill \text{for } \beta = \frac{1}{2}, \\[2mm]
E_1^{T,N}\sqrt{\delta_{0,0}^{\mathrm{SGDw}}} + C\sigma_{n,T}^{\mathrm{SGDw}}\left(\frac{4}{\tilde{\mu}_m C} + \frac{3N\left(1+\frac{L^2C^2}{2}\right)^{N-1}e^{2L^2C^2N}\log(T+1)}{(T+1)^{(\tilde{\mu}_m CN)/2}}\right) \quad \text{for } \beta = 1, \\[2mm]
E_1^{T,N}\sqrt{\delta_{0,0}^{\mathrm{SGDw}}} + C\sigma_{n,T}^{\mathrm{SGDw}}\left(\frac{2^{2\beta+1}}{\tilde{\mu}_m C}e^{\frac{\tilde{\mu}_m C}{2(1-\beta)}\frac{N}{(T+1)^\beta}} + \frac{3^\beta(1+\tilde{\mu}_m C)^{N-1}(T+2)^\beta}{L^2C^2}\,E_2^{T,N}\right) \text{ otherwise },
\end{cases}
$$

*where $E_1^{T,N}$ and $E_2^{T,N}$ are two decreasing functions in $T$ defined by*

$$E_1^{T,N} := \exp\left(1 - N\tilde{\mu}_m C\varphi_{1-\beta}(T+1) + \frac{NL^2C^2}{2}\varphi_{1-2\beta}(T+1)\right),$$

$$E_2^{T,N} := \exp\left(-\frac{N\tilde{\mu}_m C}{2}\varphi_{1-\beta}(T+1) + 2NL^2C^2\varphi_{1-2\beta}(T+1)\right).$$

We emphasize that the full result above holds for any $\beta \in [0,1]$, which in particular includes the constant step size $\beta = 0$ regime considered by [21]. When $\beta = 0$, for $E_1^{T,N}$ and $E_2^{T,N}$ to decay to zero as $T \to \infty$, we additionally need the condition

$$\tilde{\mu}_m = \mu - \alpha^m\sigma C_\chi > 4CL^2.$$

This is almost identical to the condition used in [21, Equation 2.5, Theorem 2.1], except that $4L^2$ gets replaced by $\frac{1}{2}(L + \alpha^m\sigma C_\chi)^2$. Notably this says that an additional step size condition is needed for our results to hold in the constant step size regime, but not necessary for a decreasing step size.

## B.2 Results for SGDo

SGD with reshuffling (SGDo, also called SGD without replacement) is an optimization scheme that is also widely used in practice compared to SGDo and online SGD. In the context of CD, it corresponds to Algorithm 2 with batches chosen as

$$(B_{t,1}, \ldots, B_{t,N}) = \pi(\{1, \ldots, n\}),$$

where $\pi$ is a uniform draw of the permutation group on $n$ elements. We denote the iterates of SGDo $(\psi_{t,j}^{\mathrm{SGDo}})_{t\in\mathbb{N}, j\in[N]}$. Analogously to $\vartheta_{n,m,T}^{\mathrm{SGDw}}$, we define, for $X_1'$ an i.i.d. copy of $X_1$, $\epsilon > 0$ and $n \in \mathbb{N}$, the tail probability term

$$\vartheta_{n,m,T}^{\mathrm{SGDo}}(\epsilon) := \sup_{\substack{t\in[T] \\ j\in[N]}} \mathbb{P}\left(\frac{\left\|\sum_{i=1}^n\left(\mathbb{E}\left[\phi\left(K_{\psi_{t-1,j}^{\mathrm{SGDo}}}^m(X_i)\right)\Big|X_i, \psi_{t-1,j}^{\mathrm{SGDo}}\right] - \mathbb{E}\left[\phi\left(K_{\psi_{t-1,j}^{\mathrm{SGDo}}}^m(X_1')\right)\Big|\psi_{t-1,j}^{\mathrm{SGDo}}\right]\right)\right\|}{n} > \epsilon\right).$$

Also denote $\varepsilon_{n,m,T;\nu}^{\mathrm{SGDo}}(\epsilon) = \sqrt{\epsilon^2 + \kappa_m^2\left(\vartheta_{n,m,T}^{\mathrm{SGDo}}(\epsilon)\right)^{\frac{\nu-2}{\nu}}}$ and $\sigma_{n,T}^{\mathrm{SGDw}} = \varepsilon_{n,m,T;\nu}^{\mathrm{SGDo}}(\epsilon) + \frac{5\sigma + 5\kappa_m}{\sqrt{B}}$. The following result says that $\psi_{t,j}^{\mathrm{SGDo}}$ enjoys exactly the same convergence guarantee as $\psi_{t,j}^{\mathrm{SGDo}}$ in Theorem 4.3. The statement is identical to that of Theorem B.1 and is stated in full for completeness; see Appendix F.2 for the proof, which is a slight adaptation of the proof for Theorem B.1. As before we write $\delta_{t,j}^{\mathrm{SGDo}} := \mathbb{E}\left\|\psi_{t,j}^{\mathrm{SGDo}} - \psi^*\right\|^2$.

**Theorem B.2** (Convergence of CD-SGDo). *Assume A1 (where $\Psi$ may be non-compact), A2, A3, A4 and A7. Let $\eta_t = Ct^{-\beta}$ for some $\beta \in [0,1]$ and $C > 0$, and assume that $m > \frac{\log(\sigma C_\chi/\mu)}{\log|\alpha|}$ s.t. $\tilde{\mu}_m = \mu - \alpha^m \sigma C_\chi > 0$ as in Theorem 3.2. Then for any $\epsilon > 0$, $\sqrt{\delta_{T,N}^{\mathrm{SGDo}}}$ is upper bounded by*

$$
\begin{cases}
E_1^{T,N}\sqrt{\delta_{0,0}^{\mathrm{SGDo}}} + C\sigma_{n,T}^{\mathrm{SGDo}}\left(\dfrac{4e^{\frac{\tilde{\mu}_m CN}{(T+1)^{1/2}}}}{\tilde{\mu}_m C} + 2N(1+\tilde{\mu}_m C)^{N-1}\varphi_{\frac{1}{2}-L^2C^2N}(T+1)\,E_2^{T,N}\right) \\
\hspace{9cm} \text{for } \beta = \dfrac{1}{2}\,, \\[2mm]
E_1^{T,N}\sqrt{\delta_{0,0}^{\mathrm{SGDo}}} + C\sigma_{n,T}^{\mathrm{SGDo}}\left(\dfrac{4}{\tilde{\mu}_m C} + \dfrac{3N\left(1+\frac{L^2C^2}{2}\right)^{N-1}e^{2L^2C^2N}\log(T+1)}{(T+1)^{(\tilde{\mu}_m CN)/2}}\right) \hspace{1cm} \text{for } \beta = 1\,, \\[2mm]
E_1^{T,N}\sqrt{\delta_{0,0}^{\mathrm{SGDo}}} + C\sigma_{n,T}^{\mathrm{SGDo}}\left(\dfrac{2^{2\beta+1}}{\tilde{\mu}_m C}e^{\frac{\tilde{\mu}_m C}{2(1-\beta)}\frac{N}{(T+1)^\beta}} + \dfrac{3^\beta(1+\tilde{\mu}_m C)^{N-1}(T+2)^\beta}{L^2C^2}\,E_2^{T,N}\right) \;\; \text{otherwise}\,,
\end{cases}
$$

*where $E_1^{T,N}$ and $E_2^{T,N}$ are two decreasing functions in $T$ defined by*

$$
E_1^{T,N} := \exp\left(1 - N\tilde{\mu}_m C\varphi_{1-\beta}(T+1) + \frac{NL^2C^2}{2}\varphi_{1-2\beta}(T+1)\right),
$$

$$
E_2^{T,N} := \exp\left(-\frac{N\tilde{\mu}_m C}{2}\varphi_{1-\beta}(T+1) + 2NL^2C^2\varphi_{1-2\beta}(T+1)\right).
$$

**Remark.** We also remark that existing works [47, 48] show that the standard SGDo typically gives a faster convergence rate in $T$ than SGDw. An analogous result for the CD setup would involve additional technical hurdles of jointly controlling the correlations across minibatches and from reusing data samples, and we defer this to future work.

## B.3 Explicit tail control

We now provide the full explicit tail control bounds. All results in this section hold directly for $\varepsilon_{\nu;n,m,T}^{\mathrm{SGDo}}(\epsilon)$ and $\delta_{T,N}^{\mathrm{SGDo}}$, and we omit them here. In the result below, we denote $r_\Psi$ as the radius of the smallest sphere in $\mathbb{R}^p$ that contains $\Psi$, which is finite under A1.

**Lemma B.3.** *Assume A5 and A6. Let $n \in \mathbb{N}$ be sufficiently large s.t. $\frac{\log n}{n} < \frac{\sigma_m^2 \zeta_m^2}{p+\nu-2}$. Then*

$$
\inf_{\epsilon>0}\varepsilon_{\nu;n,m,T}^{\mathrm{SGDw}}(\epsilon) \leq
$$

$$
\left(\frac{3\sigma_m\sqrt{p((\nu-2)p+2\nu)}}{\sqrt{\nu-2}} + \kappa_{\nu;m}2^{\frac{\nu-2}{2\nu}}(r_\Psi)^{\frac{(\nu-2)p}{2\nu}}\left(1 + \frac{2C_m(\nu-2)^{1/2}}{\sigma_m p^{1/2}((\nu-2)p+2\nu)^{1/2}}\right)^{\frac{(\nu-2)p}{2\nu}}\right)\frac{\sqrt{\log n}}{\sqrt{n}}.
$$

*In particular, if we additionally assume the conditions of Theorem B.1, we have*

$$
\lim_{T\to\infty}\sqrt{\delta_{T,N}^{\mathrm{SGDw}}} \leq C'(p,\nu,m,\Psi,\beta)\left(\frac{\sqrt{\log n}}{\sqrt{n}} + \frac{1}{\sqrt{B}}\right)
$$

*where*

$$
C'(p,\nu,m,\Psi,\beta) := \frac{8(1+5\sigma+5\kappa_m)}{\tilde{\mu}_m}
$$

$$
\times\left(\frac{3\sigma_m\sqrt{p((\nu-2)p+2\nu)}}{\sqrt{\nu-2}} + \kappa_{\nu;m}2^{\frac{\nu-2}{2\nu}}(r_\Psi)^{\frac{(\nu-2)p}{2\nu}}\left(1 + \frac{2C_m(\nu-2)^{1/2}}{\sigma_m p^{1/2}((\nu-2)p+2\nu)^{1/2}}\right)^{\frac{(\nu-2)p}{2\nu}}\right).
$$

**Lemma B.4.** *Assume the conditions of Theorem 4.3 and additionally that A5 holds. Then*

$$
\inf_{\epsilon>0}\varepsilon_{\nu;n,m,T}^{\mathrm{SGDw}}(\epsilon) \leq \left(3C_m + \kappa_{\nu;m}^{\nu/2}(r_\Psi)^{\frac{(\nu-2)p}{2\nu}}C_m^{-\frac{\nu-2}{2}}3^{\frac{(\nu-2)p}{2\nu}}\right)n^{-\frac{(\nu-2)\nu}{2(\nu^2+(\nu-2)p)}},
$$

$$
\lim_{T\to\infty}\left(\delta_{T,N}^{\mathrm{SGDw}}\right)^{1/2} \leq \tilde{C}'(p,\nu,m,\Psi,\beta)\left(n^{-\frac{(\nu-2)\nu}{2(\nu^2+(\nu-2)p)}} + B^{-1/2}\right),
$$

*where*

$$
\tilde{C}'(p,\nu,m,\Psi) := \frac{8(1+5\sigma+5\kappa_m)}{\tilde{\mu}_m}\left(3C_m + \kappa_{\nu;m}^{\nu/2}(r_\Psi)^{\frac{(\nu-2)p}{2\nu}}C_m^{-\frac{\nu-2}{2}}3^{\frac{(\nu-2)p}{2\nu}}\right).
$$

The next result considers a $\phi$-discrepancy mixing condition ([35]), which is a mixing assumption on $K_\psi^m$ but with respect to a specific test function $\phi$, and we impose it uniformly over $\psi \in \Psi$. We also recall that $X_1^\psi \sim p_\psi$.

**Lemma B.5.** *Assume that there exist $\tilde\alpha \in [0,1)$ and $\tilde{C}_K > 0$ such that, for all $\psi \in \Psi$ and $x \in \mathcal{X}$, $\|\mathbb{E}[\phi(K_\psi^m(x))] - \mathbb{E}[\phi(X_1^\psi)]\| \leq \tilde{C}_K \tilde\alpha^m$. Then*

$$\inf_{\epsilon>0} \varepsilon_{\nu;n,m,T}^{\mathrm{SGDw}}(\epsilon) \;\leq\; \big(1 + 2^{\frac{\nu-2}{2\nu}} \kappa_{\nu;m}(\tilde{C}_K)^{\frac{\nu-2}{2\nu}}\big)\tilde\alpha^{\frac{(\nu-2)m}{3\nu-2}} \;.$$

*In particular, if we additionally assume the conditions of Theorem 4.3, we have*

$$\lim_{T\to\infty} \sqrt{\delta_{T,N}^{\mathrm{SGDw}}} \;\leq\; \frac{8}{\mu - \alpha^m \sigma C_\chi}\Big(\big(1 + 2^{\frac{\nu-2}{2\nu}} \kappa_{\nu;m}(\tilde{C}_K)^{\frac{\nu-2}{2\nu}}\big)\tilde\alpha^{\frac{(\nu-2)m}{3\nu-2}} + \frac{5\sigma + 5\kappa_m}{\sqrt{B}}\Big)\;.$$

## C  Auxiliary Tools

### C.1  Properties of $\varphi_\gamma$

The following lemma collects some identities used in [22].

**Lemma C.1.** $\varphi_\gamma$ *satisfies the following properties:*

(i) $\varphi_\gamma$ *is increasing on $\mathbb{R}_+$ for all $\gamma$ ;*

(ii) $\varphi_\gamma(t) \leq \frac{t^\gamma}{\gamma}$ *for $\gamma > 0$, and $\varphi_\gamma(t) \leq -\frac{1}{\gamma}$ for $\gamma < 0$ ;*

(iii) $\varphi_{1-\beta}(t) \geq t^{1-\beta}$ *for $\beta \in (0,1]$ ;*

(iv) $\varphi_\gamma(t) - \varphi_\gamma(\frac{t}{2}) \geq \frac{1}{2} x^\gamma$ *for $\gamma \in (0,1]$ .*

The next lemma provides some additional results on $\varphi_\gamma$.

**Lemma C.2.** $\varphi_\gamma$ *satisfies the following properties:*

(i) $\varphi_\gamma$ *is positive on $t > 0$ and increasing for every $\gamma \in \mathbb{R}$;*

(ii) *For $1 \leq t_1 \leq t_2$ and $\beta \geq 0$, we have*

$$\varphi_{1-\beta}(t_2+1) - \varphi_{1-\beta}(t_1) \;\leq\; \sum_{t=t_1}^{t_2} t^{-\beta} \;\leq\; 2\big(\varphi_{1-\beta}(t_2+1) - \varphi_{1-\beta}(t_1)\big) \;.$$

*If instead $\beta < 0$, we have*

$$\frac{1}{2}\big(\varphi_{1-\beta}(t_2+1) - \varphi_{1-\beta}(t_1)\big) \;\leq\; \sum_{t=t_1}^{t_2} t^{-\beta} \;\leq\; \varphi_{1-\beta}(t_2+1) - \varphi_{1-\beta}(t_1) \;;$$

(iii) *For $1 \leq t_1 \leq t_2$ and $\gamma \neq 0$, we have*

$$\begin{aligned} t_1^{\gamma-1} &\;\leq\; \varphi_\gamma(t_2) - \varphi_\gamma(t_1) \;\leq\; t_2^{\gamma-1} & \text{if } \gamma \geq 1 \;, \\ t_2^{-(1-\gamma)} &\;\leq\; \varphi_\gamma(t_2) - \varphi_\gamma(t_1) \;\leq\; t_1^{-(1-\gamma)} & \text{if } \gamma \leq 1 \;; \end{aligned}$$

(iv) *Let $1 \leq t_1 < t_2$ and $\kappa, \beta \geq 0$. If $\kappa \neq 1$ and $a > 0$, we have*

$$\sum_{t=t_1}^{t_2} (t+1)^{-\beta} \exp\big(a\,\varphi_{1-\kappa}(t-1)\big) \;\leq\; \frac{(t_2+1)^{\max\{\kappa-\beta,0\}}}{a} \exp\big(a\,\varphi_{1-\kappa}(t_2+1)\big) \;,$$

*and if $\kappa \neq 1$ and $a < 0$, we have*

$$\sum_{t=t_1}^{t_2} (t+1)^{-\beta} \exp\big(a\,\varphi_{1-\kappa}(t)\big) \;\leq\; \frac{(t_2+1)^{\max\{\kappa-\beta,0\}}}{(-a)} \exp\big(a\,\varphi_{1-\kappa}(t_1)\big) \;.$$

*Proof of Lemma C.2.* (i) follows from checking $\gamma > 0$, $\gamma = 0$ and $\gamma < 0$ respectively. The first set of bounds in (ii) follow by noting that $t \mapsto t^{-\beta}$ is decreasing for $\beta \geq 0$:

$$\sum_{t=t_1}^{t_2} t^{-\beta} \;\geq\; \int_{t_1}^{t_2+1} t^{-\beta} dt \;=\; \varphi_{1-\beta}(t_2+1) - \varphi_{1-\beta}(t_1) \;,$$

$$\sum_{t=t_1}^{t_2} t^{-\beta} \;\leq\; 2\sum_{t=t_1}^{t_2} (t+1)^{-\beta} \;\leq\; 2\int_{t_1-1}^{t_2} (t+1)^{-\beta} dt \;=\; 2\big(\varphi_{1-\beta}(t_2+1) - \varphi_{1-\beta}(t_1)\big) \;.$$

The second set of bounds follows from noting that $t \mapsto t^{-\beta}$ is increasing for $\beta < 0$:

$$\sum_{t=t_1}^{t_2} t^{-\beta} \geq \frac{1}{2} \sum_{t=t_1}^{t_2} (t+1)^{-\beta} \geq \frac{1}{2} \int_{t_1-1}^{t_2} (t+1)^{-\beta} dt = \frac{1}{2} \left( \varphi_{1-\beta}(t_2+1) - \varphi_{1-\beta}(t_1) \right),$$

$$\sum_{t=t_1}^{t_2} t^{-\beta} \leq \int_{t_1}^{t_2+1} t^{-\beta} dt = \varphi_{1-\beta}(t_2+1) - \varphi_{1-\beta}(t_1).$$

For (iii), we note that for $\gamma \neq 0$,

$$\varphi_\gamma(t_2) - \varphi_\gamma(t_1) = \frac{t_2^\gamma - t_1^\gamma}{\gamma},$$

so by the mean value theorem,

$$\inf_{t_1 \leq t \leq t_2} t^{\gamma-1} \leq \varphi_\gamma(t_2) - \varphi_\gamma(t_1) \leq \sup_{t_1 \leq t \leq t_2} t^{\gamma-1}.$$

The desired bounds then follow from an explicit computation of the infimum and the maximum in each of the two cases $\gamma \geq 1$ and $\gamma \leq 1$.

For (iv), we first consider the case $\kappa \neq 1$ and $a > 0$. Then

$$\sum_{t=t_1}^{t_2} (t+1)^{-\beta} \exp\left( a\, \varphi_{1-\kappa}(t-1) \right) = e^{-\frac{a}{1-\kappa}} \sum_{t=t_1}^{t_2} (t+1)^{-\beta} \exp\left( \frac{at^{1-\kappa}}{1-\kappa} \right)$$

$$\leq e^{-\frac{a}{1-\kappa}} \max_{t_1 \leq t \leq t_2} \left( \frac{(t+1)^\kappa}{(t+1)^\beta} \right) \sum_{t=t_1}^{t_2} (t+1)^{-\kappa} \exp\left( \frac{at^{1-\kappa}}{1-\kappa} \right)$$

$$\leq e^{-\frac{a}{1-\kappa}} (t_2+1)^{\max\{\kappa-\beta,0\}} \sum_{t=t_1}^{t_2} (t+1)^{-\kappa} \exp\left( \frac{at^{1-\kappa}}{1-\kappa} \right).$$

Since, for $x \geq 0$, $x \mapsto (x+1)^{-\kappa}$ is decreasing and $x \mapsto \exp(ax^{1-\kappa}/(1-\kappa))$ is increasing, we have that for $t_1 \leq t \leq t_2$ and $x \in [t, t+1]$,

$$(t+1)^{-\kappa} \leq x^{-\kappa} \qquad \text{and} \qquad \exp(at^{1-\kappa}/(1-\kappa)) \leq \exp(ax^{1-\kappa}/(1-\kappa)).$$

This implies that

$$\sum_{t=t_1}^{t_2} (t+1)^{-\beta} \exp\left( a\, \varphi_{1-\kappa}(t-1) \right)$$

$$\leq (t_2+1)^{\max\{\kappa-\beta,0\}} e^{-\frac{a}{1-\kappa}} \sum_{t=t_1}^{t_2} \int_t^{t+1} x^{-\kappa} \exp\left( \frac{ax^{1-\kappa}}{1-\kappa} \right) dx$$

$$= (t_2+1)^{\max\{\kappa-\beta,0\}} e^{-\frac{a}{1-\kappa}} \int_{t_1}^{t_2+1} x^{-\kappa} \exp\left( \frac{ax^{1-\kappa}}{1-\kappa} \right) dx$$

$$\leq \frac{(t_2+1)^{\max\{\kappa-\beta,0\}}}{a} e^{-\frac{a}{1-\kappa}} e^{\frac{a(t_2+1)^{1-\kappa}}{1-\kappa}}$$

$$= \frac{(t_2+1)^{\max\{\kappa-\beta,0\}}}{a} \exp\left( a\, \varphi_{1-\kappa}(t_2+1) \right).$$

The main difference in the case $\kappa \neq 1$ and $a < 0$ is that we now use $x \mapsto \exp(a(x+1)^{1-\kappa}/(1-\kappa))$ is decreasing to obtain, for $t_1 \leq t \leq t_2$ and $x \in [t, t+1]$,

$$\exp(a(t+1)^{1-\kappa}/(1-\kappa)) \leq \exp(ax^{1-\kappa}/(1-\kappa)).$$

A similar argument then yields

$$\sum_{t=t_1}^{t_2} (t+1)^{-\beta} \exp\left( a\, \varphi_{1-\kappa}(t) \right)$$

$$\leq (t_2+1)^{\max\{\kappa-\beta,0\}} e^{-\frac{a}{1-\kappa}} \sum_{t=t_1}^{t_2} (t+1)^{-\kappa} \exp\left( \frac{a(t+1)^{1-\kappa}}{1-\kappa} \right)$$

$$\leq (t_2+1)^{\max\{\kappa-\beta,0\}} e^{-\frac{a}{1-\kappa}} \int_{t_1}^{t_2+1} x^{-\kappa} \exp\left( \frac{ax^{1-\kappa}}{1-\kappa} \right) dx$$

$$= \frac{(t_2+1)^{\max\{\kappa-\beta,0\}}}{a} \left( \exp\left( a\, \varphi_{1-\kappa}(t_2+1) \right) - \exp\left( a\, \varphi_{1-\kappa}(t_1) \right) \right)$$

$$\leq \frac{(t_2+1)^{\max\{\kappa-\beta,0\}}}{(-a)} \exp\left( a\, \varphi_{1-\kappa}(t_1) \right).$$

$\square$

We also need the following lemma, which is useful for controlling the accumulation of errors from the noise terms over iterations.

**Lemma C.3.** *For any $a, b \geq 0$, $T, N \in \mathbb{N}$ and $\kappa, \beta \geq 0$ such that $bt^{-\beta} - at^{-\kappa} \leq 1$ for all $1 \leq t \leq T$, we have that*

$$\prod_{t=1}^{T}(1 - bt^{-\beta} + at^{-\kappa})^N \leq \exp\left(-bN\,\varphi_{1-\beta}(T+1) + aN\,\varphi_{1-\kappa}(T+1)\right).$$

*Moreover, for any $\zeta \geq 0$, we have that*

$$\sum_{t=1}^{T} t^{-\zeta}\left(\sum_{j=1}^{N}(1 - bt^{-\beta} + at^{-\kappa})\right)\prod_{s=t+1}^{T}(1 - bs^{-\beta} + as^{-\kappa})^N$$
$$\leq Q_1 + \exp\left(-\frac{bN}{2}\,\varphi_{1-\beta}(T+1) + 4aN\,\varphi_{1-\kappa}(T+1)\right) Q_2\,,$$

*where*

$$Q_1 := \begin{cases} \dfrac{2^{2\zeta+1}(T+3)^{\max\{\beta-\zeta,0\}}}{b}\exp\left(\dfrac{bN}{2(1-\beta)(T+1)^\beta}\right) & \text{if } \beta \neq 1\,, b > 0\,, \\ 2N\varphi_{1-\zeta+bN/2}(T+1)\exp\left(-\dfrac{bN}{2}\,\varphi_{1-\beta}(T+1)\right) & \text{if } \beta = 1 \text{ or } b = 0\,, \end{cases}$$

*and*

$$Q_2 := \begin{cases} \dfrac{3^\zeta(1+a)^{N-1}}{2a}(T+2)^{\max\{\kappa-\zeta,0\}} & \text{if } \kappa \neq 1 \text{ and } a > 0\,, \\ 2N(1+a)^{N-1}\varphi_{1-\zeta-2aN}(T+1) & \text{if } \kappa = 1 \text{ or } a = 0\,. \end{cases}$$

*In the special case $\zeta = \beta = 1 < \kappa$, we have*

$$\sum_{t=1}^{T} t^{-\zeta}\left(\sum_{j=1}^{N}(1 - bt^{-\beta} + at^{-\kappa})^{j-1}\right)\prod_{s=t+1}^{T}(1 - bs^{-\beta} + as^{-\kappa})^N$$
$$\leq \frac{4}{b} + \frac{3N(1+a)^{N-1}e^{\frac{4aN}{\kappa-1}}\log(T+1)}{(T+1)^{\frac{bN}{2}}}\,.$$

*Proof of Lemma C.3.* By assumption, $bt^{-\beta} - at^{-\kappa} \leq 1$ for all $1 \leq t \leq T$. Since $0 \leq 1 - x \leq e^{-x}$ for all $x \leq 1$, we have that for any $1 \leq t_1 \leq t_2 \leq T$,

$$\prod_{t=t_1}^{t_2}(1 - bt^{-\beta} + at^{-\kappa})^N \leq \exp\left(-bN\sum_{t=t_1}^{t_2}t^{-\beta} + aN\sum_{t=t_1}^{t_2}t^{-\kappa}\right). \qquad (13)$$

Applying this to the first quantity of interest followed by noting that $a, b \geq 0$ and using Lemma C.2(ii), we obtain the first bound that

$$\prod_{t=1}^{T}(1 - bt^{-\beta} + at^{-\kappa})^N \leq \exp\left(-bN\sum_{t=1}^{T}t^{-\beta} + aN\sum_{t=1}^{T}t^{-\kappa}\right)$$
$$\leq \exp\left(-bN\,\varphi_{1-\beta}(T+1) + aN\,\varphi_{1-\kappa}(T+1)\right).$$

For the second bound, we define

$$t_0 := \sup\left\{t \leq T \;\middle|\; \frac{b}{2} \leq at^{-(\kappa-\beta)}\right\}.$$

Then by noting that $1 - bt^{-\beta} + at^{-\kappa} \geq 0$ for all $1 \leq t \leq T$ again, we can bound the quantity of interest as

$$\sum_{t=1}^{T} t^{-\zeta} \left( \sum_{j=1}^{N} (1 - bt^{-\beta} + at^{-\kappa})^{j-1} \right) \prod_{s=t+1}^{T} (1 - bs^{-\beta} + as^{-\kappa})^N$$

$$= \sum_{t=t_0+1}^{T} t^{-\zeta} \left( \sum_{j=1}^{N} (1 - bt^{-\beta} + at^{-\kappa})^{j-1} \right) \prod_{s=t+1}^{T} (1 - bs^{-\beta} + as^{-\kappa})^N$$

$$+ \left( \prod_{s=t_0+1}^{T} (1 - bs^{-\beta} + as^{-\kappa}) \right)$$

$$\times \left( \sum_{t=1}^{t_0} t^{-\zeta} \left( \sum_{j=1}^{N} (1 - bt^{-\beta} + at^{-\kappa})^{j-1} \right) \prod_{s=t+1}^{t_0} (1 - bs^{-\beta} + as^{-\kappa})^N \right)$$

$$\leq \sum_{t=t_0+1}^{T} t^{-\zeta} \left( \sum_{j=1}^{N} \left( 1 - \frac{b}{2} t^{-\beta} \right)^{j-1} \right) \prod_{s=t+1}^{T} \left( 1 - \frac{b}{2} s^{-\beta} \right)^N$$

$$+ \left( \prod_{s=t_0+1}^{T} \left( 1 - \frac{b}{2} s^{-\beta} \right)^N \right) \left( \sum_{t=1}^{t_0} t^{-\zeta} \left( \sum_{j=1}^{N} (1 + at^{-\kappa})^{j-1} \right) \prod_{s=t+1}^{t_0} (1 + as^{-\kappa})^N \right)$$

$$\leq N \times \underbrace{\sum_{t=t_0+1}^{T} t^{-\zeta} \prod_{s=t+1}^{T} \left( 1 - \frac{b}{2} s^{-\beta} \right)^N}_{=:S_1}$$

$$+ N(1+a)^{N-1} \times \underbrace{\left( \prod_{s=t_0+1}^{T} \left( 1 - \frac{b}{2} s^{-\beta} \right)^N \right)}_{=:S_3} \times \underbrace{\left( \sum_{t=1}^{t_0} t^{-\zeta} \prod_{s=t+1}^{t_0} (1 + as^{-\kappa})^N \right)}_{=:S_2} .$$

In the last line, we have used that $0 \leq 1 - \frac{b}{2} t^{-\beta} \leq 1$ for $t \geq t_0 + 1$ and $1 + at^{-\kappa} \leq 1 + a$. To control the three quantities, we first note that by (13), we have

$$S_3 \leq \exp\left( -\frac{bN}{2} \sum_{s=1}^{T} s^{-\beta} \right) \exp\left( \frac{bN}{2} \sum_{s=1}^{t_0} s^{-\beta} \right)$$

$$\overset{(a)}{\leq} \exp\left( -\frac{bN}{2} \sum_{s=1}^{T} s^{-\beta} \right) \exp\left( aN \sum_{s=1}^{t_0} s^{-\kappa} \right)$$

$$\overset{(b)}{\leq} \exp\left( -\frac{bN}{2} \varphi_{1-\beta}(T+1) + 2aN\, \varphi_{1-\kappa}(T+1) \right) .$$

In $(a)$ above, we have noted that $\frac{b}{2} \leq as^{-(\kappa-\beta)}$ for $s \leq t_0$; in $(b)$, we have used $t_0 \leq T$ and Lemma C.2(ii) with $a, b \geq 0$. In the special case $\beta = 1 < \kappa$, the above yields

$$S_3 \leq (T+1)^{-\frac{bN}{2}} \exp\left( 2aN \frac{1 - (T+1)^{-(\kappa-1)}}{\kappa - 1} \right)$$

$$\leq (T+1)^{-\frac{bN}{2}} e^{\frac{2aN}{\kappa-1}} .$$

We now control $S_2$. By (13) again, we have

$$S_2 \leq \sum_{t=1}^{t_0} t^{-\zeta} \exp\left( aN \sum_{s=t+1}^{t_0} s^{-\kappa} \right)$$

$$\leq \sum_{t=1}^{T} t^{-\zeta} \exp\left( aN \sum_{s=t+1}^{T} s^{-\kappa} \right)$$

$$\overset{(c)}{\leq} \exp\left( 2aN\varphi_{1-\kappa}(T+1) \right) \times \left( \sum_{t=1}^{T} t^{-\zeta} \exp\left( -2aN\varphi_{1-\kappa}(t+1) \right) \right)$$

$$\overset{(d)}{\leq} 3^{\zeta} \exp\left( 2aN\varphi_{1-\kappa}(T+1) \right) \sum_{t=1}^{T} (t+2)^{-\zeta} \exp\left( -2aN\varphi_{1-\kappa}(t+1) \right) .$$

In $(c)$ above, we have applied Lemma C.2(ii); in $(d)$, we have noted that $\sup_{t\in\mathbb{N}}(t+2)^{\beta}/t^{\beta} = 3^{\beta}$. If $\kappa \neq 1$ and $a > 0$, we can apply Lemma C.2(iv) to get that

$$S_2 \leq \frac{3^{\zeta}}{2aN}(T+2)^{\max\{\kappa-\zeta, 0\}} \exp\left( 2aN\varphi_{1-\kappa}(T+1) \right)$$

$$= \frac{Q_2}{N(1+a+c)^{N-1}} \exp\left( 2aN\varphi_{1-\kappa}(T+1) \right) .$$

If $\kappa = 1$ or $a = 0$, the bound from $(c)$ above reads

$$
\begin{aligned}
S_2 &\leq \exp\big(2aN\varphi_{1-\kappa}(T+1)\big) \sum_{t=1}^{T} t^{-\zeta}(t+1)^{-2aN} \\
&\leq \exp\big(2aN\varphi_{1-\kappa}(T+1)\big) \sum_{t=1}^{T} t^{-\zeta-2aN} \\
&\leq 2\,\varphi_{1-\zeta-2aN}(T+1)\exp\big(2aN\varphi_{1-\kappa}(T+1)\big) \;=\; \frac{Q_2}{N(1+a)^{N-1}} \exp\big(2aN\varphi_{1-\kappa}(T+1)\big)\,,
\end{aligned}
$$

where we have used Lemma C.2(ii) in the last line. Now consider the special case with $\zeta = 1 < \kappa$, the bound from $(d)$ becomes

$$
\begin{aligned}
S_2 &\leq 3\exp\big(2aN\varphi_{1-\kappa}(T+1)\big)\Big(\sum_{t=1}^{T}(t+2)^{-1}\exp\big(-2aN\varphi_{1-\kappa}(t+1)\big)\Big) \\
&\leq 3\exp\Big(2aN\,\frac{1-(T+1)^{-(\kappa-1)}}{\kappa-1}\Big)\sum_{t=1}^{T}(t+2)^{-1} \\
&\leq 3e^{\frac{2aN}{\kappa-1}}\log(T+1)\,.
\end{aligned}
$$

We are left with controlling $S_1$, which follows from a similar strategy as controlling $S_2$:

$$
\begin{aligned}
S_1 &\overset{(13)}{\leq} \sum_{t=t_0+1}^{T} t^{-\zeta}\exp\Big(-\frac{bN}{2}\sum_{s=t+1}^{T}s^{-\beta}\Big) \\
&\leq \sum_{t=1}^{T} t^{-\zeta}\exp\Big(-\frac{bN}{2}\sum_{s=t+1}^{T}s^{-\beta}\Big) \\
&\overset{(a)}{\leq} \exp\Big(-\frac{bN}{2}\varphi_{1-\beta}(T+1)\Big)\sum_{t=1}^{T} t^{-\zeta}\exp\Big(\frac{bN}{2}\varphi_{1-\beta}(t+1)\Big) \qquad (14) \\
&\leq 4^{\zeta}\exp\Big(-\frac{bN}{2}\varphi_{1-\beta}(T+1)\Big)\sum_{t=1}^{T}(t+3)^{-\zeta}\exp\Big(\frac{bN}{2}\varphi_{1-\beta}(t+1)\Big)\,.
\end{aligned}
$$

In $(a)$ above, we used Lemma C.2(ii). For $\beta \neq 1$ and $b \neq 0$, we can apply Lemma C.2(iv) with $\frac{b}{2} > 0$ to obtain

$$
\begin{aligned}
S_1 &\leq 4^{\zeta}\exp\Big(-\frac{bN}{2}\varphi_{1-\beta}(T+1)\Big)\frac{(T+3)^{\max\{\beta-\zeta,0\}}}{bN/2}\exp\Big(\frac{bN}{2}\varphi_{1-\beta}(T+3)\Big) \\
&= \frac{2^{2\zeta+1}(T+3)^{\max\{\beta-\zeta,0\}}}{bN}\exp\Big(\frac{bN}{2(1-\beta)}\big((T+3)^{1-\beta}-(T+1)^{1-\beta}\big)\Big) \\
&\overset{(b)}{\leq} \frac{2^{2\zeta+1}(T+3)^{\max\{\beta-\zeta,0\}}}{bN}\exp\Big(\frac{bN}{2(1-\beta)(T+1)^{\beta}}\Big) \;=\; \frac{Q_1}{N}\,.
\end{aligned}
$$

In $(b)$, we have used Lemma C.2(iii) with $1-\beta \leq 1$. Meanwhile, if $\beta = 1$ or $b = 0$, we have

$$
\begin{aligned}
S_1 &\leq \exp\Big(-\frac{bN}{2}\varphi_{1-\beta}(T+1)\Big)\sum_{t=1}^{T} t^{-\zeta}(t+1)^{bN/2} \\
&\leq \exp\Big(-\frac{b}{2}\varphi_{1-\beta}(T+1)\Big)\sum_{t=1}^{T} t^{-\zeta+bN/2} \\
&\leq 2\varphi_{1-\zeta+bN/2}(T+1)\exp\Big(-\frac{bN}{2}\varphi_{1-\beta}(T+1)\Big) \;=\; \frac{Q_1}{N}\,.
\end{aligned}
$$

For the special case with $\zeta = \beta = 1$, the bound in (14) becomes

$$
\begin{aligned}
S_1 &\leq \exp\Big(-\frac{bN}{2}\varphi_0(T+1)\Big)\sum_{t=1}^{T} t^{-1}\exp\Big(\frac{bN}{2}\varphi_0(t+1)\Big) \\
&= (T+1)^{-\frac{bN}{2}}\sum_{t=1}^{T} t^{-1}(t+1)^{\frac{bN}{2}} \\
&\leq (T+1)^{-\frac{bN}{2}}\sum_{t=1}^{T} t^{-(1-\frac{bN}{2})} \\
&\overset{(c)}{\leq} 2(T+1)^{-\frac{bN}{2}}\varphi_{\frac{bN}{2}}(T+1) \;=\; 2(T+1)^{-\frac{bN}{2}}\frac{(T+1)^{bN/2}-1}{bN/2} \;\leq\; \frac{4}{bN}\,.
\end{aligned}
$$

In $(c)$, we have used Lemma C.2(ii) for both the case $1-\frac{bN}{2} \leq 0$ and $1-\frac{bN}{2} \geq 0$.

Combining the bounds for the general cases, we obtain the first desired inequality that

$$
\begin{aligned}
&\sum_{t=1}^{T} t^{-\zeta}\Big(\sum_{j=1}^{N}(1-bt^{-\beta}+at^{-\kappa})^{j-1}\Big)\prod_{s=t+1}^{T}(1-bs^{-\beta}+as^{-\kappa})^{N} \\
&\leq Q_1 + \exp\Big(-\frac{b}{2}\varphi_{1-\beta}(T+1)+u\varphi_{1-\xi}(T+3)+4a\,\varphi_{1-\kappa}(T+1)\Big)Q_2\,.
\end{aligned}
$$

For the special case $\zeta = \beta = 1 < \kappa, \gamma$, combining the earlier bounds gives

$$\sum_{t=1}^{T} t^{-\zeta} \left( \sum_{j=1}^{N} (1 - bt^{-\beta} + at^{-\kappa})^{j-1} \right) \prod_{s=t+1}^{T} (1 - bs^{-\beta} + as^{-\kappa})^N$$

$$\leq \frac{4}{b} + \frac{3N(1+a)^{N-1} e^{\frac{4aN}{\kappa-1}} \log(T+1)}{(T+1)^{\frac{bN}{2}}} .$$

$\square$

## C.2 Contraction and integrability results

The next result is a standard result in convex analysis, needed to handle projections performed in Algorithms 1 and 2.

**Lemma C.4.** *Let $\Psi$ a be convex subset of $\mathbb{R}^p$. Let $\psi^\star \in \Psi$. Then, for all $\psi \in \mathbb{R}^p$, we have:*

$$\|\mathrm{Proj}_\Psi(\psi) - \psi^\star\| \leq \|\psi - \psi^\star\|$$

*Proof.* We have:

$$\begin{aligned}
\|\psi - \psi^\star\|^2 &= \|\psi - \mathrm{Proj}_\Psi(\psi) + \mathrm{Proj}_\Psi(\psi) - \psi^\star\|^2 \\
&= \|\psi - \mathrm{Proj}_\Psi(\psi)\|^2 + 2\langle \psi - \mathrm{Proj}_\Psi(\psi), \mathrm{Proj}_\Psi(\psi) - \psi^\star \rangle + \|\mathrm{Proj}_\Psi(\psi) - \psi^\star\|^2
\end{aligned}$$

Since by [49, Proposition 1.1.9], we have:

$$\langle \psi - \mathrm{Proj}_\Psi(\psi), \psi' - \mathrm{Proj}_\Psi(\psi) \rangle \leq 0$$

for all $\psi' \in \Psi$, we can use this inequality at $\psi' = \psi^\star \in \Psi$ to obtain:

$$\|\psi - \psi^\star\|^2 \geq \|\psi - \mathrm{Proj}_\Psi(\psi)\|^2$$

and the result follows by taking the square root. $\square$

We now state two lemmas that guarantee an amount of integrability sufficient to our analysis.

**Lemma C.5.** *Let $p, q \in \mathcal{P}(\mathcal{X})$ such that $\frac{\mathrm{d}p}{\mathrm{d}q}$ exists, and such that $\chi^2(p; q) < +\infty$. Assume that $f \in L^2(q)$ Then $|\mathbb{E}_p f| < +\infty$.*

*Proof.* By assumption, $f \in L^2(q)$. Moreover, $\chi^2(p, q) < +\infty$, and thus we have $\frac{\mathrm{d}p}{\mathrm{d}q} - 1 \in L^2(q)$. Thus, the inner product is finite, and

$$\left| \int f\left( \frac{\mathrm{d}p}{\mathrm{d}q} - 1 \right) \mathrm{d}q \right| = \left| \int f \mathrm{d}p - \int f \mathrm{d}q \right| = |\mathbb{E}_p f - \mathbb{E}_q f| := M < +\infty$$

$$\implies M - |\mathbb{E}_q f| < \mathbb{E}_p f < M + |\mathbb{E}_q f|$$

$\square$

**Lemma C.6.** *For all $\psi \in \Psi$, for all $m \geq 1$, and for all $k \geq 1$, we have:*

$$\mathbb{E}_{p_{\psi^\star}} \left\| P_\psi^m \phi \right\|^k < +\infty$$

*Proof.* By analycity of the log partition function $\psi \longmapsto \log Z(\psi)$, we have $\int \|\phi\|^k \, \mathrm{d}p_\psi(x) < +\infty$ for all $\psi \in \Psi$, and thus, the function $x \longmapsto \|\phi\|^k (x) \in L^2(p_\psi)$ for all $\psi$. Consequently, $P_\psi^m \|\phi\|^k \in L^2(p_\psi)$. We can apply Lemma C.5 to $P_\psi^m \|\phi\|^k$ to obtain $\mathbb{E}_{\psi^\star} P^m \|\phi\|^k < +\infty$ for all $k \geq 1$ and for all $m \geq 0$. As a by-product, we obtain $P_\psi^m \|\phi\|^k \in L^2(p_{\psi^\star})$, and thus so $\|P^m \phi\|^k$. $\square$

The following lemma is used multiple time in our analysis.

**Lemma C.7.** *Assume A3. Let $q$ be a positive integer. Let $f := (f_1, \ldots, f_q)$ such that $f_k \in \{\phi_i\}_{i=1}^p \cup \{\phi_i \phi_j\}_{i,j=1}^p$ for $k \in [q]$. Then, for all $\psi \in \Psi$, we have*

$$\left\| \mathbb{E}_{p_{\psi^\star}} \left[ P_\psi^m (f - \mathbb{E}_{p_\psi} f) \right] \right\| \leq \alpha^m C_\chi \left( \mathbb{E}_{p_\psi} \left[ \|f - \mathbb{E}_{p_\psi} f\|^2 \right] \right)^{1/2} \|\psi - \psi^\star\|$$

*Proof.* Let us note first that

$$\|\mathbb{E}_{p_{\psi^\star}} P_\psi^m \left( f - \mathbb{E}_{p_\psi} f \right) \|^2 \overset{(a)}{=} \sum_{i=1}^q \left( \int P_\psi^m \left( f_i - \mathbb{E}_{p_\psi} f_i \right)(x) \left( p_{\psi^\star}(x) - p_\psi(x) \right) \mathrm{d}x \right)^2$$

$$= \sum_{i=1}^q \left( \int P_\psi^m \left( f_i - \mathbb{E}_{p_\psi} f_i \right)(x) \left( \frac{\mathrm{d}p_{\psi^\star}}{\mathrm{d}p_\psi}(x) - 1 \right) p_\psi(x)\mathrm{d}x \right)^2$$

$$\overset{(b)}{\leq} \left( \int \left( \frac{\mathrm{d}p_\psi^\star}{\mathrm{d}p_\psi}(x) - 1 \right)^2 p_\psi(x)\mathrm{d}x \right) \sum_{i=1}^q \int P_\psi^m \left( f_i - \mathbb{E}_{p_\psi} f_i \right)(x)^2 p_\psi(x)\mathrm{d}x$$

$$\leq \chi^2(p_\psi, p_{\psi^\star}) \sum_{i=1}^q \left\| P_\psi^m (f_i - \mathbb{E}_{p_\psi} f_i) \right\|_{L^2(p_\psi)}$$

$$\overset{(c)}{\leq} \alpha^{2m} \chi^2(p_\psi, p_{\psi^\star}) \sum_{i=1}^q \left\| f_i - \mathbb{E}_{p_\psi} f_i \right\|_{L^2(p_\psi)(\mathbb{R}^d)}$$

$$\overset{(d)}{\leq} \alpha^{2m} C_\chi^2 \|\psi - \psi^\star\|^2 \, \mathbb{E}_{p_\psi} \left\| f - \mathbb{E}_{p_\psi} f \right\|^2 .$$

Here, we used the fact that $P_\psi^m$ admits $p_\psi$ as an invariant measure in $(a)$ [34, Eq. (1.2.2)], the Cauchy-Schwarz inequality in $(b)$ $\qquad\qquad\square$

## C.3 Miscellaneous

**Lemma C.8.** *Let $f : \Psi \to \mathbb{R}^p$ be a differentiable function in the interior of $\Psi \subseteq \mathbb{R}^p$. For $\psi \in \Psi$, define $\sigma_{\min}(\psi) := \inf_{\theta \in \Psi, \|\theta\|=1} \theta^\top \nabla f(\psi)\theta$ and $\sigma_{\max}(\psi) := \sup_{\theta \in \Psi, \|\theta\|=1} \theta^\top \nabla f(\psi)\theta$ with respect to the Jacobian matrix $\nabla f(\psi)$. Then for any $\psi_1, \psi_2 \in \Psi$, we have that*

$$\inf_{\psi \in \Psi} \sigma_{\min}(\psi) \leq (\psi_1 - \psi_2)^\top \left( f(\psi_1) - f(\psi_2) \right) \leq \sup_{\psi \in \Psi} \sigma_{\max}(\psi)$$

*Proof of Lemma C.8.* By the mean value theorem, there exists some $a \in (0,1)$ such that

$$(\psi_1 - \psi_2)^\top \left( f(\psi_1) - f(\psi_2) \right) = (\psi_1 - \psi_2)^\top \left( f(1 \times \psi_1 + 0 \times \psi_2) - f(0 \times \psi_1 + 0 \times \psi_2) \right)$$

$$= (\psi_1 - \psi_2)^\top \nabla f(a\psi_1 + (1-a)\psi_2)(\psi_1 - \psi_2)$$

$$= \|\psi_1 - \psi_2\|^2 \frac{(\psi_1 - \psi_2)^\top}{\|\psi_1 - \psi_2\|} \nabla f(a\psi_1 + (1-a)\psi_2) \frac{\psi_1 - \psi_2}{\|\psi_1 - \psi_2\|} .$$

Plugging in the definition of $\sigma_{\max}$ gives the desired upper bound and similarly $\sigma_{\min}$ implies the lower bound. $\qquad\qquad\square$

# D Proofs for Online CD

## D.1 Auxiliary Lemmas for Online CD

We recall the following notations used in the next lemmas, namely $\sigma_\psi := \mathbb{E}_{p_\psi} \|\phi - \mathbb{E}_{p_\psi} \phi\|^2$, $\sigma_\star := \sigma_{\psi^\star}$ and $\sigma := \sup_{\psi \in \Psi} \sigma_\psi$.

We now provide two intermediary lemmas necessary to analyze the impact of variance in the CD gradient. The strategy is similar in both of them: we change the integration from $p_{\psi^\star}$ to $p_\psi$ to obtain contraction, at the cost of an additional term scaling with $C_\chi \|\psi - \psi^\star\|$. We obtain exact constants that we choose to describe in terms of the smoothness parameters of the problem, e.g. the $k^{th}$ derivatives of the log partition function $\log Z$, which, for $k \geq 2$, equals the $k^{th}$ derivative of the negative cross-entropy model w.r.t $p_{\psi^\star}$.

**Second Moment convergence** The following lemmas guarantee the second moment of a sample from $k_\psi^m p_{\psi^\star}$ approaches the second moment of a sample from the target distribution $p_\psi$.

**Lemma D.1.** *Under A1, A2 and A3, for all $\psi \in \Psi$, we have:*

$$\left| \mathbb{E}_{p_{\psi^\star}} P_\psi^m \|\phi\|^2 - \mathbb{E}_{p_\psi} \|\phi\|^2 \right| \leq \alpha^m C_\chi \|\psi - \psi^\star\| \|\log Z\|_{1,\infty}$$

*where*

$$\|\log Z\|_{1,\infty} := \sup_{\psi \in \Psi} \sum_{i=1}^{p} (4\partial_i^1 \log Z(\psi)^2 \partial_i^2 \log Z(\psi) + 2\partial_i^2 \log Z(\psi)^2 + 4\partial_i^1 \log Z(\psi)\partial_i^3 \log Z(\psi)$$
$$+ \partial_i^4 \log Z(\psi))^{1/2} < +\infty$$

*Proof.* Applying Lemma C.7 to each $f_i := \phi_i^2$, we have

$$\left|\mathbb{E}_{p_{\psi^\star}} P_\psi^m \phi_i^2 - \mathbb{E}_{p_\psi} \phi_i^2\right| = \alpha^m C_\chi \|\psi - \psi^\star\| \left(\mathbb{E}_{p_\psi} \left(\phi_i^2 - \mathbb{E}_{p_\psi} \phi_i^2\right)^2\right)^{1/2}$$
$$= \alpha^m C_\chi \|\psi - \psi^\star\| \left(\mathbb{E}_{p_\psi} \phi_i^4 - (\mathbb{E}_{p_\psi} \phi_i^2)^2\right)^{1/2}$$

We map the two moments to derivatives of $\log Z(\psi)$, since the $k^{th}$ derivative of $\log Z(\psi)$ is the $k^{th}$ cumulant. It can be shown, using the multivariate moment to cumulant mapping, that

$$\mathbb{E}\phi_i^4 = \frac{\partial \log Z}{\partial \psi_i}^4 + 6\frac{\partial \log Z}{\partial \psi_i}^2 \frac{\partial^2 \log Z}{\partial \psi_i^2} + 3\left(\frac{\partial^2 \log Z}{\partial \psi_i^2}\right)^2 + 4\frac{\partial \log Z}{\partial \psi_i} \frac{\partial^3 \log Z}{\partial \psi_i^3} + \frac{\partial^4 \log Z}{\partial \psi_i^4}$$
$$= \partial_i^1 \log Z(\psi)^4 + 6\partial_i^1 \log Z(\psi)^2 \partial_i^2 \log Z(\psi) + 3\partial_i^2 \log Z(\psi)^2 + 4\partial_i^1 \log Z(\psi)\partial_i^3 \log Z(\psi)$$
$$+ \partial_i^4 \log Z(\psi)$$

where $\partial_i^k \log Z(\psi)$ denotes the $k^{th}$ derivative of $\log Z$ with respect to $\psi_i$. On the other hand,
$$\mathbb{E}\phi_i^2 = \partial_i^1 \log Z(\psi)^2 + \partial_i^2 \log Z(\psi)$$
$$\implies (\mathbb{E}\phi_i^2)^2 = \partial_i^1 \log Z(\psi)^4 + 2\partial_i^1 \log Z(\psi)^2 \partial_i^2 \log Z(\psi) + \partial_i^2 \log Z(\psi)^2$$

implying

$$\mathbb{E}_{p_\psi} \phi_i^4 - (\mathbb{E}_{p_\psi^2} \phi_i^2)^2 = 4\partial_i^1 \log Z(\psi)^2 \partial_i^2 \log Z(\psi) + 2\partial_i^2 \log Z(\psi)^2 + 4\partial_i^1 \log Z(\psi)\partial_i^3 \log Z(\psi)$$
$$+ \partial_i^4 \log Z(\psi)$$

The result follows by summing over $i$, since:

$$\left|\mathbb{E}_{p_{\psi^\star}} P_\psi^m \|\phi\|^2 - \mathbb{E}_{p_\psi} \|\phi\|^2\right| \leq \sum_{i=1}^{d} |\mathbb{E}_{p_{\psi^\star}} P_\psi^m \phi_i^2 - \mathbb{E}_{p_\psi} \phi_i^2|$$

Note that $\|\log Z\|_{1,\infty}$ is finite since $\Psi$ is compact and $\log Z$ is analytic. $\qquad\square$

**Squared First Moment convergence**    The next lemma provides convergence (in squared absolute value) of the first moment of the $m$-iterated Markov kernel $k_\psi^m$.

**Lemma D.2.** *Under A1, A2 and A3, for all $\psi \in \Psi$, we have*
$$\left|\left\|\mathbb{E}_{p_{\psi^\star}} P_\psi^m \phi\right\|^2 - \left\|\mathbb{E}_{p_\psi} \phi\right\|^2\right| \leq \alpha^m \sigma_\psi^2 + C_\chi \alpha^{m/4} \|\log Z\|_{2,\infty} \|\psi - \psi^\star\|$$

*where*
$$\|\log Z\|_{2,\infty} := \sup_{\psi \in \Psi} \sum_{i=1}^{p} \left(F(\psi)\partial_i^2 \log Z(\psi)\right)^{1/4} + 2\left|\partial_i^1 \log Z(\psi)\partial_i^2 \log Z(\psi)^{1/2}\right|$$

*and*
$$F(\psi) := 15\partial_i^2 \log Z(\psi)^3 + 10\partial_i^3 \log Z(\psi)^2 + 15\partial_i^2 \log Z(\psi)\partial_i^4 \log Z(\psi) + \partial_i^6 \log Z(\psi)$$

*Proof.* We have:
$$(\mathbb{E}_{p_{\psi^\star}} P_\psi^m \phi_i)^2 = (\mathbb{E}_{p_{\psi^\star}} P_\psi^m \phi_i - \mathbb{E}_{p_\psi} \phi_i + \mathbb{E}_{p_\psi} \phi_i)^2$$
$$= (\mathbb{E}_{p_{\psi^\star}} P^m \phi_i - \mathbb{E}_{p_\psi} \phi_i)^2 + 2\mathbb{E}_{p_\psi} \phi_i (\mathbb{E}_{p_{\psi^\star}} P^m \phi_i - \mathbb{E}_{p_\psi} \phi_i)$$
$$+ (\mathbb{E}_\psi \phi_i)^2$$
$$\implies |\mathbb{E}_{p_{\psi^\star}} (P_\psi^m \phi_i)^2 - (\mathbb{E}_{p_\psi} \phi_i)^2| \leq \underbrace{\left|\left(\mathbb{E}_{p_{\psi^\star}} P_\psi^m \left(\phi_i - \mathbb{E}_{p_\psi} \phi_i\right)\right)^2\right|}_{\Delta_1}$$
$$+ 2\underbrace{\left|\mathbb{E}_{p_\psi} \phi_i \,\mathbb{E}_{p_{\psi^\star}} P^m \left(\phi_i - \mathbb{E}_{p_\psi} \phi_i\right)\right|}_{\Delta_2}$$

where

$$\Delta_1 = \underbrace{\mathbb{E}_{p_\psi}(P_\psi^m(\phi_i - \mathbb{E}_\psi \phi_i))^2}_{\Delta_{1,1}} + \underbrace{\mathbb{E}_{p_\psi}\left((P_\psi^m \phi_i - \mathbb{E}_{p_\psi}\phi_i)^2\left(\frac{\mathrm{d}p_{\psi^\star}}{\mathrm{d}p_\psi} - 1\right)\right)}_{\Delta_{1,2}}$$

$$\overset{(a)}{\leq} \alpha^{2m}\mathbb{E}_\psi(\phi_i - \mathbb{E}_{p_\psi}\phi_i)^2 + C_\chi \|\psi - \psi^\star\| \left(\mathbb{E}_{p_\psi}(P^m(\phi_i - \mathbb{E}_{p_\psi}\phi_i)^4\right)^{1/2}$$

$$\overset{(b)}{\leq} \alpha^{2m}\mathbb{E}_\psi(\phi_i - \mathbb{E}_\psi \phi_i)^2 + C_\chi \|\psi - \psi^\star\| \left(\mathbb{E}_{p_\psi}P^m(\phi_i - \mathbb{E}_{p_\psi}\phi_i)^2\right)^{1/4} \left(\mathbb{E}_{p_\psi}P^m(\phi_i - \mathbb{E}_{p_\psi}\phi_i)^6\right)^{1/4}$$

$$\overset{(c)}{\leq} \alpha^{2m}\mathbb{E}_\psi(\phi_i - \mathbb{E}_\psi \phi_i)^2 + \alpha^{m/2}C_\chi \|\psi - \psi^\star\| \left(\mathbb{E}_{p_\psi}(\phi_i - \mathbb{E}_{p_\psi}\phi_i)^2\right)^{1/4} \left(\mathbb{E}_{p_\psi}(\phi_i - \mathbb{E}_{p_\psi}\phi_i)^6\right)^{1/4}.$$

In $(a)$, we used the restricted spectral gap Assumption A3 for $\Delta_{1,1}$, and the Cauchy-Schwarz inequality combined with Assumption A2 for $\Delta_{1,2}$. In $(b)$, we used the Cauchy-Schwarz once again, and in $(c)$ we used the fact that $P_\psi^m$ is a contraction in $L^6(p_\psi)$ and another invocation of the spectral gap assumption A3 As an aside, note that a simpler result can be obtained by making regularity assumption on the mapping $\psi \longmapsto P_\psi^m$. Assuming that $\psi \longmapsto P_\psi^m(x)$ is uniformly $L_m$-Lipschitz across $x \in \mathcal{X}$ for instance (as done in [21, Assumption 5]), the second term $\Delta_{1,2}$ of $\Delta_1$ could have been handled using

$$\Delta_{1,2} \leq 2\mathbb{E}_{p_{\psi^\star}} \left\|P_\psi^m \phi - \mathbb{E}_{\psi^\star}\phi\right\|^2 + 2\left\|\mathbb{E}_{p_\psi}\phi - \mathbb{E}_{p_{\psi^\star}}\phi\right\|^2$$

$$\leq 4(L_m \|\psi - \psi^\star\| + \sigma_\star^2 \alpha^{2m} + 2\left\|\mathbb{E}_{p_\psi}\phi - \mathbb{E}_{p_{\psi^\star}}\phi\right\|^2)$$

$$\leq 4(L_m \|\psi - \psi^\star\| + \sigma_\star^2 \alpha^{2m} + 2L^2 \|\psi - \psi^\star\|)$$

Although this result does not require possibly large constants related to sixth-order moments, it is less tight in the sense that it does not go to 0 as $m \to \infty$. Back to the main proof, and $\Delta_2$ in particular. Applying Lemma C.7 to $f := \phi$, we have

$$\Delta_2 \leq \alpha^m \mathbb{E}_{p_\psi}\phi_i \, C_\chi \|\psi - \psi^\star\| \left(\mathbb{E}_{p_\psi}(\phi_i - \mathbb{E}_{p_\psi}\phi_i)^2\right)^{1/2}$$

$$\leq \alpha^m C_\chi \partial_i^1 \log Z(\psi)\partial_i^2 \log Z(\psi)^{1/2} \|\psi - \psi^\star\|$$

Putting everything together, we have:

$$|\mathbb{E}_{p_{\psi^\star}}(P_\psi^m \phi_i)^2 - \mathbb{E}_{p_\psi}\phi_i^2| \leq \alpha^{2m}\mathbb{E}_{p_\psi}(\phi_i - \mathbb{E}_{p_\psi}\phi_i)^2$$

$$+ C_\chi \|\psi - \psi^\star\| \left(\alpha^{m/2}(\mathbb{E}_{p_\psi}(\phi_i - \log \phi_i)^6)^{1/4}(\mathbb{E}_{p_\psi}(\phi_i - \log \phi_i)^2)^{1/4}\right.$$

$$\left. + 2\alpha^m \partial_i^1 \log Z(\psi)\partial_i^2 \log Z(\psi)^{1/2}\right)$$

$$\leq \alpha^{2m}\mathbb{E}_{p_\psi}(\phi_i - \mathbb{E}_{p_\psi}\phi_i)^2$$

$$+ C_\chi \alpha^{m/2} \|\psi - \psi^\star\| \left((\mathbb{E}_{p_\psi}(\phi_i - \log \phi_i)^6)^{1/4}(\mathbb{E}_{p_\psi}(\phi_i - \log \phi_i)^2)^{1/4}\right.$$

$$\left. + 2\left|\partial_i^1 \log Z(\psi)\partial_i^2 \log Z(\psi)^{1/2}\right|\right)$$

Summing over $i$, we obtain

$$|\mathbb{E}_{p_{\psi^\star}}\|P_\psi^m \phi\|^2 - (\mathbb{E}_{p_\psi}\phi)^2|$$

$$\leq \sum_{i=1}^p |\mathbb{E}_{p_{\psi^\star}}(P_\psi^m \phi_i)^2 - (\mathbb{E}_{p_\psi}\phi_i)^2|$$

$$\leq \alpha^{2m}\sum_{i=1}^p \mathbb{E}_{p_\psi}(\phi_i - \mathbb{E}_{p_\psi}\phi_i)^2 + C_\chi \alpha^{m/2}\|\log Z\|_{2,\infty}\|\psi - \psi^\star\|$$

$$\leq \alpha^{2m}\sigma_\psi^2 + C_\chi \alpha^{m/2}\|\log Z\|_{2,\infty}\|\psi - \psi^\star\|$$

where

$$\|\log Z\|_{2,\infty} = \sup_{\psi \in \Psi} \sum_{i=1}^p \left(\mathbb{E}_{p_\psi}(\phi_i - \mathbb{E}_{p_\psi}\phi_i)^6\right)^{1/4}\left(\mathbb{E}_{p_\psi}(\phi_i - \mathbb{E}_{p_\psi}\phi_i)^2\right)^{1/4}$$

$$+ 2\left|\partial_i^1 \log Z(\psi)\partial_i^2 \log Z(\psi)^{1/2}\right|$$

Similarly to the previous lemma, one can upper bound $\mathbb{E}(\phi_i - \mathbb{E}_{p_\psi}\phi_i)^6$ using the *centered* moment to cumulant formula:

$$\mathbb{E}_{p_\psi}(\phi_i - \mathbb{E}\phi_i)^6 = 15\partial_i^2 \log Z(\psi)^3 + 10\partial_i^3 \log Z(\psi)^2 + 15\partial_i^2 \log Z(\psi)\partial_i^4 \log Z(\psi) + \partial_i^6 \log Z(\psi)$$
$$=: F(\psi)$$

To get a full description of $\|\log Z\|_{2,\infty}$:

$$\|\log Z\|_{2,\infty} = \sup_{\psi \in \Psi} \sum_{i=1}^{p} \left(F(\psi)\partial_i^2 \log Z(\psi)\right)^{1/4} + 2\left|\partial_i^1 \log Z(\psi)\partial_i^2 \log Z(\psi)^{1/2}\right| .$$

$\square$

**Lemma D.3.** *Under A1, A2 and A3, for all $\psi \in \Psi$, we have*

$$\left|\mathbb{E}_{p_{\psi^\star}}\|P_\psi^m\phi\|^2 - \left\|\mathbb{E}_{p_\psi}\phi\right\|^2\right| \le \alpha^m \sigma_\psi^2 + C_\chi \alpha^{m/4} \|\log Z\|_{2,\infty} \|\psi - \psi^\star\|$$

*where*

$$\|\log Z\|_{2,\infty} := \sup_{\psi \in \Psi} \sum_{i=1}^{p} \left(F(\psi)\partial_i^2 \log Z(\psi)\right)^{1/4} + 2\left|\partial_i^1 \log Z(\psi)\partial_i^2 \log Z(\psi)^{1/2}\right|$$

*and*

$$F(\psi) := 15\partial_i^2 \log Z(\psi)^3 + 10\partial_i^3 \log Z(\psi)^2 + 15\partial_i^2 \log Z(\psi)\partial_i^4 \log Z(\psi) + \partial_i^6 \log Z(\psi)$$

*Proof.* We have:

$$\mathbb{E}_{p_{\psi^\star}}(P_\psi^m\phi_i)^2 = \mathbb{E}_{p_{\psi^\star}}(P_\psi^m\phi_i - \mathbb{E}_{p_\psi}\phi_i + \mathbb{E}_{p_\psi}\phi_i)^2$$
$$= \mathbb{E}_{p_{\psi^\star}}(P^m\phi_i - \mathbb{E}_{p_\psi}\phi_i)^2 + 2\mathbb{E}_{p_\psi}\phi_i \, \mathbb{E}_{p_{\psi^\star}}P^m(\phi_i - \mathbb{E}_{p_\psi}\phi_i)$$
$$+ (\mathbb{E}_\psi\phi_i)^2$$
$$\implies |\mathbb{E}_{p_{\psi^\star}}(P_\psi^m\phi_i)^2 - (\mathbb{E}_{p_\psi}\phi_i)^2| \le \underbrace{\left|\mathbb{E}_{p_{\psi^\star}}\left(P^m\left(\phi_i - \mathbb{E}_{p_\psi}\phi_i\right)\right)^2\right|}_{\Delta_1}$$
$$+ 2\underbrace{\left|\mathbb{E}_{p_\psi}\phi_i \, \mathbb{E}_{p_{\psi^\star}}P^m\left(\phi_i - \mathbb{E}_{p_\psi}\phi_i\right)\right|}_{\Delta_2}$$

where

$$\Delta_1 = \underbrace{\mathbb{E}_{p_\psi}(P_\psi^m(\phi_i - \mathbb{E}_\psi\phi_i))^2}_{\Delta_{1,1}} + \underbrace{\mathbb{E}_{p_\psi}\left(\left(P_\psi^m\phi_i - \mathbb{E}_{p_\psi}\phi_i\right)^2\left(\frac{\mathrm{d}p_{\psi^\star}}{\mathrm{d}p_\psi} - 1\right)\right)}_{\Delta_{1,2}}$$

$$\overset{(a)}{\le} \alpha^{2m}\mathbb{E}_\psi(\phi_i - \mathbb{E}_{p_\psi}\phi_i)^2 + C_\chi\|\psi - \psi^\star\|\left(\mathbb{E}_{p_\psi}(P^m(\phi_i - \mathbb{E}_{p_\psi}\phi_i)^4)\right)^{1/2}$$

$$\overset{(b)}{\le} \alpha^{2m}\mathbb{E}_\psi(\phi_i - \mathbb{E}_\psi\phi_i)^2 + C_\chi\|\psi - \psi^\star\|\left(\mathbb{E}_{p_\psi}P^m(\phi_i - \mathbb{E}_{p_\psi}\phi_i)^2\right)^{1/4}\left(\mathbb{E}_{p_\psi}P^m(\phi_i - \mathbb{E}_{p_\psi}\phi_i)^6\right)^{1/4}$$

$$\overset{(c)}{\le} \alpha^{2m}\mathbb{E}_\psi(\phi_i - \mathbb{E}_\psi\phi_i)^2 + \alpha^{m/2}C_\chi\|\psi - \psi^\star\|\left(\mathbb{E}_{p_\psi}(\phi_i - \mathbb{E}_{p_\psi}\phi_i)^2\right)^{1/4}\left(\mathbb{E}_{p_\psi}(\phi_i - \mathbb{E}_{p_\psi}\phi_i)^6\right)^{1/4} .$$

In $(a)$, we used the restricted spectral gap Assumption A3 for $\Delta_{1,1}$, and the Cauchy-Schwarz inequality combined with Assumption A2 for $\Delta_{1,2}$. In $(b)$, we used the Cauchy-Schwarz once again, and in $(c)$ we used the fact that $P_\psi^m$ is a contraction in $L^6(p_\psi)$ and another invocation of the spectral gap assumption A3 As an aside, note that a simpler result can be obtained by making regularity assumption on the mapping $\psi \longmapsto P_\psi^m$. Assuming that $\psi \longmapsto P_\psi^m(x)$ is uniformly $L_m$-Lipschitz across $x \in \mathcal{X}$ for instance (as done in [21, Assumption 5]), the second term $\Delta_{1,2}$ of $\Delta_1$ could have been handled using

$$\Delta_{1,2} \le 2\mathbb{E}_{p_{\psi^\star}}\left\|P_\psi^m\phi - \mathbb{E}_{\psi^\star}\phi\right\|^2 + 2\left\|\mathbb{E}_{p_\psi}\phi - \mathbb{E}_{p_{\psi^\star}}\phi\right\|^2$$
$$\le 4(L_m\|\psi - \psi^\star\| + \sigma_\star^2\alpha^{2m} + 2\left\|\mathbb{E}_{p_\psi}\phi - \mathbb{E}_{p_{\psi^\star}}\phi\right\|^2)$$
$$\le 4(L_m\|\psi - \psi^\star\| + \sigma_\star^2\alpha^{2m} + 2L^2\|\psi - \psi^\star\|)$$

Although this result does not require possibly large constants related to sixth-order moments, it is less tight in the sense that it does not go to $0$ as $m \to \infty$. Back to the main proof, and $\Delta_2$ in particular. Applying Lemma C.7 to $f := \phi$, we have

$$\Delta_2 \leq \alpha^m \mathbb{E}_{p_\psi} \phi_i \, C_\chi \, \|\psi - \psi^\star\| \left( \mathbb{E}_{p_\psi} (\phi_i - \mathbb{E}_{p_\psi} \phi_i)^2 \right)^{1/2}$$
$$\leq \alpha^m C_\chi \partial_i^1 \log Z(\psi) \partial_i^2 \log Z(\psi)^{1/2} \, \|\psi - \psi^\star\|$$

Putting everything together, we have:

$$|\mathbb{E}_{p_{\psi^\star}} (P_\psi^m \phi_i)^2 - \mathbb{E}_{p_\psi} \phi_i^2| \leq \alpha^{2m} \mathbb{E}_{p_\psi} (\phi_i - \mathbb{E}_{p_\psi} \phi_i)^2$$
$$+ C_\chi \|\psi - \psi^\star\| \left( \alpha^{m/2} (\mathbb{E}_{p_\psi} (\phi_i - \log \phi_i)^6)^{1/4} (\mathbb{E}_{p_\psi} (\phi_i - \log \phi_i)^2)^{1/4} \right.$$
$$\left. + 2\alpha^m \partial_i^1 \log Z(\psi) \partial_i^2 \log Z(\psi)^{1/2} \right)$$
$$\leq \alpha^{2m} \mathbb{E}_{p_\psi} (\phi_i - \mathbb{E}_{p_\psi} \phi_i)^2$$
$$+ C_\chi \alpha^{m/2} \|\psi - \psi^\star\| \left( (\mathbb{E}_{p_\psi} (\phi_i - \log \phi_i)^6)^{1/4} (\mathbb{E}_{p_\psi} (\phi_i - \log \phi_i)^2)^{1/4} \right.$$
$$\left. + 2 \left| \partial_i^1 \log Z(\psi) \partial_i^2 \log Z(\psi)^{1/2} \right| \right)$$

Summing over $i$, we obtain

$$|\mathbb{E}_{p_{\psi^\star}} \|P_\psi^m \phi\|^2 - (\mathbb{E}_{p_\psi} \phi)^2|$$
$$\leq \sum_{i=1}^p |\mathbb{E}_{p_{\psi^\star}} (P_\psi^m \phi_i)^2 - (\mathbb{E}_{p_\psi} \phi_i)^2|$$
$$\leq \alpha^{2m} \sum_{i=1}^p \mathbb{E}_{p_\psi} (\phi_i - \mathbb{E}_{p_\psi} \phi_i)^2 + C_\chi \alpha^{m/2} \|\log Z\|_{2,\infty} \|\psi - \psi^\star\|$$
$$\leq \alpha^{2m} \sigma_\psi^2 + C_\chi \alpha^{m/2} \|\log Z\|_{2,\infty} \|\psi - \psi^\star\|$$

where

$$\|\log Z\|_{2,\infty} = \sup_{\psi \in \Psi} \sum_{i=1}^p \left( \mathbb{E}_{p_\psi} (\phi_i - \mathbb{E}_{p_\psi} \phi_i)^6 \right)^{1/4} \left( \mathbb{E}_{p_\psi} (\phi_i - \mathbb{E}_{p_\psi} \phi_i)^2 \right)^{1/4}$$
$$+ 2 \left| \partial_i^1 \log Z(\psi) \partial_i^2 \log Z(\psi)^{1/2} \right|$$

Similarly to the previous lemma, one can upper bound $\mathbb{E}(\phi_i - \mathbb{E}_{p_\psi} \phi_i)^6$ using the *centered* moment to cumulant formula:

$$\mathbb{E}_{p_\psi} (\phi_i - \mathbb{E}\phi_i)^6 = 15\partial_i^2 \log Z(\psi)^3 + 10\partial_i^3 \log Z(\psi)^2 + 15\partial_i^2 \log Z(\psi) \partial_i^4 \log Z(\psi) + \partial_i^6 \log Z(\psi)$$
$$=: F(\psi)$$

To get a full description of $\|\log Z\|_{2,\infty}$:

$$\|\log Z\|_{2,\infty} = \sup_{\psi \in \Psi} \sum_{i=1}^p \left( F(\psi) \partial_i^2 \log Z(\psi) \right)^{1/4} + 2 \left| \partial_i^1 \log Z(\psi) \partial_i^2 \log Z(\psi)^{1/2} \right| .$$

$\square$

We can now use the previous lemmas to obtain an expression on the second moment of the contrastive divergence gradient estimator, relating it to the one of the stochastic log-likelihood gradient estimator.

**Lemma D.4.** *Under A1, A2 and A3, we have:*

$$\mathbb{E} \|h_t(\psi, X_t)\|^2 \leq 2\sigma_\star^2 + 2\sigma_\psi^2 + 2L^2 \|\psi - \psi^\star\|^2 + 4(\sigma_\psi^2 \alpha^{2m} + \alpha^{m/2} \|\log Z\|_{3,\infty} C_\chi \|\psi - \psi^\star\|)$$

*where* $\|\log Z\|_{3,\infty} := 2\max(\|\log Z\|_{1,\infty}, \|\log Z\|_{2,\infty})$.

*Proof.* We rely on the following decomposition:

$$h_t(\psi, X_t) = \underbrace{(\phi(X_t) - \mathbb{E}_{p_{\psi^\star}} \phi)}_{\Delta_{1,1}} + \underbrace{(\mathbb{E}_{p_{\psi^\star}} \phi - \mathbb{E}_{p_\psi} \phi)}_{\Delta_{1,2}} + \underbrace{(\mathbb{E}_{k_\psi^m(X_t, \cdot)} \phi - \phi(k_\psi^m(X_t, \cdot)))}_{\Delta_2}$$
$$+ \underbrace{(\mathbb{E}_{p_\psi} \phi - \mathbb{E}_{k_\psi^m(X_t, \cdot)} \phi)}_{\Delta_3}$$

$\Delta_{1,1} + \Delta_{1,2}$ form the differentiable stochastic gradient $g_t$ of Equation 6. Note that $\Delta_{1,1}$ is mean-zero, and $\Delta_2$ is mean-zero conditionally on $X_t$. Consequently, $\mathbb{E} \langle \Delta_2, \Delta_3 \rangle = \mathbb{E} \langle \Delta_2, \Delta_{1,1} \rangle = \mathbb{E} \langle \Delta_{1,1}, \Delta_{1,2} \rangle = \mathbb{E} \langle \Delta_{1,1}, \Delta_2 \rangle = 0$, and the only mixed-terms that remain to be controlled are $\mathbb{E} \langle \Delta_{1,1}, \Delta_3 \rangle$ and $\mathbb{E} \langle \Delta_{1,2}, \Delta_3 \rangle$. We first control the unmixed terms, and the simple ones first: we have $\mathbb{E} \|\Delta_{1,1}\|^2 = \sigma_\star^2$, as well as $\mathbb{E} \|\Delta_{1,2}\|^2 \leq L^2 \|\psi - \psi^\star\|^2$. For $\Delta_2$, we have:

$$\mathbb{E}_{k_\psi^m(x,\cdot)} \|\Delta_2\|^2 = \mathbb{E}_{k_\psi^m(x,\cdot)} \|\phi(k_\psi^m(x,\cdot))\|^2 - \|\mathbb{E}_{k_\psi^m(x,\cdot)} \phi(k_\psi^m(x,\cdot))\|^2$$
$$= P_\psi^m \|\phi(x)\|^2 - \|P_\psi^m \phi(x)\|^2$$

We can invoke Lemmas D.3 and D.1, which guarantee

$$\left| \mathbb{E}_{p_{\psi^\star}} \|P_\psi^m \phi\|^2 - \|\mathbb{E}_{p_\psi} \phi\|^2 \right| \leq \alpha^{2m} \sigma_\psi^2 + C_\chi \alpha^{m/2} \|\log Z\|_{2,\infty} \|\psi - \psi^\star\|$$

$$\left| \mathbb{E}_{p_{\psi^\star}} P_\psi^m \|\phi\|^2 - \mathbb{E}_{p_\psi} \|\phi\|^2 \right| \leq \alpha^m C_\chi \|\log Z\|_{1,\infty} \|\psi - \psi^\star\|$$

to obtain

$$\mathbb{E}_{p_{\psi^\star}} \mathbb{E}_{k_\psi^m(x,\cdot)} \|\Delta_2\|^2$$
$$= \mathbb{E}_{p_\psi} \|\phi\|^2 - \|\mathbb{E}\phi\|^2 + \alpha^{2m} \sigma_\psi^2 + C_\chi \alpha^{m/2} \left( \|\log Z\|_{1,\infty} + \|\log Z\|_{2,\infty} \right) \|\psi - \psi^\star\|$$
$$= \mathbb{E} \|\phi - \mathbb{E}\phi\|^2 + \alpha^{2m} \sigma_\psi^2 + \alpha^{m/2} \|\log Z\|_{3,\infty} C_\chi \|\psi - \psi^\star\|$$

where $\|\log Z\|_{3,\infty} := 2\max(\|\log Z\|_{1,\infty}, \|\log Z\|_{2,\infty})$.

For $\Delta_3$, notice that $\Delta_3$ is precisely the term $\Delta_1$ in Lemma D.3, and we can thus bound it by

$$\mathbb{E}_{p_{\psi^\star}} \|\Delta_3\|^2 \leq \sigma_\psi^2 \alpha^{2m} + \alpha^{m/2} \|\log Z\|_{2,\infty} C_\chi \|\psi - \psi^\star\|$$

Finally, we simply bound $2\mathbb{E} \langle \Delta_{1,1}, \Delta_3 \rangle$ by $\mathbb{E} \|\Delta_{1,1}\|^2 + \mathbb{E} \|\Delta_3\|^2$, and $2\mathbb{E} \langle \Delta_{1,2}, \Delta_3 \rangle$ by $\mathbb{E} \|\Delta_{1,2}\|^2 + \mathbb{E} \|\Delta_3\|^2$. Putting everything together, we have:

$$\mathbb{E} \|h_t(\psi)\|^2 = \mathbb{E} \|\Delta_{1,1}\|^2 + \mathbb{E} \|\Delta_{1,2}\|^2 + \mathbb{E} \|\Delta_2\|^2 + \mathbb{E} \|\Delta_3\|^2 + 2\mathbb{E} \langle \Delta_{1,1}, \Delta_3 \rangle + 2\mathbb{E} \langle \Delta_{1,2}, \Delta_3 \rangle$$
$$\leq 2\mathbb{E} \|\Delta_{1,1}\|^2 + 2\mathbb{E} \|\Delta_{1,2}\|^2 + \mathbb{E} \|\Delta_2\|^2 + 3\mathbb{E} \|\Delta_3\|^2$$
$$\leq 2\sigma_\star^2 + 2L^2 \|\psi - \psi^\star\|^2 + \sigma_\psi^2 + 4(\sigma_\psi^2 \alpha^{2m} + \alpha^{m/2} \|\log Z\|_{3,\infty} C_\chi \|\psi - \psi^\star\|)$$
$$\leq 2\sigma_\star^2 + 2\sigma_\psi^2 + 2L^2 \|\psi - \psi^\star\|^2 + 4(\sigma_\psi^2 \alpha^{2m} + \alpha^{m/2} \|\log Z\|_{3,\infty} C_\chi \|\psi - \psi^\star\|)$$

$\square$

## D.2 Proof of the SGD recursion (Lemma 3.1)

We are now ready to provide an SGD-style recursion for the expected squared distance to the optimum $\mathbb{E} \left[ \|\psi_t - \psi^\star\|^2 \right]$.

**Lemma** (Restatement of Lemma 3.1)**.** *Let $\psi_t$ be the iterates produced by Algorithm 1. Denote $\delta_t = \mathbb{E} \left[ \|\psi_t - \psi^\star\|^2 \right]$, $\sigma_\star = (\mathbb{E}_{p_{\psi^\star}} \|\phi - \mathbb{E}_{p_{\psi^\star}} \phi\|^2)^{1/2}$, and $\sigma_t = (\mathbb{E}_{p_{\psi_t}} \|\phi - \mathbb{E}_{p_{\psi_t}} \phi\|^2)^{1/2}$. Then, under Assumptions A1, A2 and A3, for all $t \geq 1$, we have:*

$$\delta_t \leq (1 - 2\eta_t \tilde{\mu}_{m,t-1} + 2\eta_t^2 L^2) \delta_{t-1} + \eta_t^2 \tilde{\sigma}_{m,t-1}^2 + 4\alpha^{m/2} \eta_t^2 \|\log Z\|_{2,\infty} C_\chi \delta_{t-1}^{1/2}$$

*where $\|\log Z\|_{3,\infty}$ is a constant, $\tilde{\mu}_{m,t} := \mu - \alpha^m \sigma_t C_\chi$, and $\tilde{\sigma}_{m,t} := (\sigma_\star^2 + \sigma_t^2 + 2\sigma_t^2 \alpha^{2m})^{1/2}$.*

*Proof.* In this proof, we note $(\mathcal{F}_t)_{t \geq 0}$, the increasing family of $\sigma$-algebras generated by the random variables $(X_t)_{t \geq 0} \sim p_{\psi^\star}$ and the Markov chain samples $\tilde{X}_t^m | X_t, \psi_t \sim k_{\psi_t}^m(X_t, \cdot)$. We decompose the integrand of $\delta_t$ as follows:

$$\|\psi_t - \psi_n^\star\|^2 = \|\text{Proj}_\Psi(\psi_{t-1} - \eta_t h_t(\psi_{t-1})) - \psi_n^\star\|^2$$
$$\leq \|\psi_{t-1} - \eta_t h_t(\psi_{t-1}) - \psi_n^\star\|^2 \quad \text{(By Lemma C.4)}$$
$$= \|\psi_{t-1} - \psi^\star\|^2 - 2\eta_t \langle h_t(\psi_{t-1}), \psi_{t-1} - \psi_n^\star \rangle + \eta_t^2 \|h_t(\psi_{t-1})\|^2$$

The first term is (up to an averaging operation) the previous iterate. The middle term will ensure (provided $m$ is large enough) contraction of the expected distance to the optimum. Finally, the third term can be described by Lemma D.4, and essentially behaves like the second moment of a log-likelihood stochastic gradient. Indeed, noting $g(\psi) := -\mathbb{E}_{p_{\psi^\star}}\phi + \mathbb{E}_{p_\psi}\phi$ the expectation of $g_t$ w.r.t $x_t$ (which is the gradient of the negative cross-entropy between $p_\psi$ and $p_{\psi^\star}$), we have:

$$\langle h_t(\psi_{n,t-1}), \psi_{t-1} - \psi_n^\star \rangle = \langle g(\psi_{n,t-1}), \psi_{t-1} - \psi_n^\star \rangle + \underbrace{\langle (h_t(\psi_{n,t-1}) - g(\psi_{n,t-1})), \psi_{t-1} - \psi_n^\star \rangle}_{\Delta}$$

and applying Lemma C.7, we get that

$$h_t(\psi_{t-1}) - g(\psi_{t-1}) = \phi(k^m(X_t, \cdot)) - \mathbb{E}_{p_\psi}\phi$$
$$\implies \mathbb{E}_{p_{\psi^\star}} \mathbb{E}_{k_\psi^m(X_t, \cdot)} [h_t(\psi_{t-1}) - g(\psi_{t-1}) | \mathcal{F}_{t-1}] = P_\psi^m \phi - \mathbb{E}_{p_\psi}\phi \,,$$

meaning

$$|\mathbb{E}[\Delta | \mathcal{F}_{t-1}]| \leq \left\| \mathbb{E}_{p_{\psi^\star}} P_\psi^m (\phi - \mathbb{E}_{p_\psi}\phi) \right\| \|\psi_{t-1} - \psi^\star\|$$
$$\leq \alpha^m C_\chi (\mathbb{E}_{p_{\psi_{t-1}}} \left\| f - \mathbb{E}_{p_{\psi_{t-1}}} f \right\|^2)^{1/2} \|\psi - \psi^\star\|^2$$
$$\leq \alpha^m \sigma_{t-1} C_\chi \|\psi - \psi^\star\|^2$$

On the other hand, by applying Lemma C.8 to $g$, we have:

$$\langle g(\psi_{t-1}), \psi_{t-1} - \psi^\star \rangle = \langle g(\psi_{t-1}) - g(\psi^\star), \psi_{t-1} - \psi^\star \rangle \geq \mu \|\psi_{t-1} - \psi^\star\|^2$$

Combining the above results, we obtain:

$$\mathbb{E}_{p_{\psi^\star}} \mathbb{E}_{k_\psi^m(x, \cdot)} \left[ \|\psi_t - \psi^\star\|^2 | \mathcal{F}_{t-1} \right]$$
$$\leq (1 - 2\eta_t(\mu - \alpha^m \sigma_{t-1} C_\chi)) \|\psi_{t-1} - \psi^\star\|^2$$
$$+ \eta_t^2 (2\sigma_\star^2 + 2\sigma_{t-1}^2 + 2L^2 \|\psi_{t-1} - \psi^\star\|^2 + 4(\sigma_{t-1}^2 \alpha^{2m} + \alpha^{m/2} \|\log Z\|_{3,\infty} C_\chi \|\psi_{t-1} - \psi^\star\|))$$

And the result follows by integrating over $\mathcal{F}_{t-1}$. $\qquad\square$

### D.3   Proof of Online CD convergence

We now prove Theorem 3.2. The recursion of Lemma 3.1 is almost identifiable, up to a cross-term of second order, with the one of an SGD algorithm as presented in the setting of [22, Theorem 1]. To make the identification exact, we use the bound $4\alpha^{m/2}\eta_t^2 \|\log Z\|_{3,\infty} C_\chi \delta_{t-1}^{1/2} \leq 2\alpha^{m/2}\eta_t^2 \delta_t + 2\alpha^{m/2}\eta_t^2 \|\log Z\|_{3,\infty}^2 C_\chi^2$, yielding the following recursion:

$$\delta_t \leq (1 - 2\eta_t(\mu - \alpha^m \sigma C_\chi) + 2\eta_t^2(L^2 + \alpha^{m/2}))\delta_{t-1} + (\sigma^2(2 + 2\alpha^{2m}) + \alpha^{m/2} \|\log Z\|_{3,\infty}^2 C_\chi^2)\eta_t^2$$

where we used the fact that $\tilde{\sigma}_{m,t} \leq \sigma$. This recursion is of the same form as the one studied in [22, Equation 6, Theorem 1] given by:

$$\delta_t \leqslant \left(1 - 2\mu\gamma_t + 2L^2\gamma_t^2\right) \delta_{t-1} + 2\sigma^2\gamma_t^2$$

by identifying:

$$\sigma^2 \leftarrow \sigma^2(2 + 2\alpha^{2m}) + \alpha^{m/2} \|\log Z\|_{3,\infty}^2 =: \tilde{\sigma}_m^2$$
$$L^2 \leftarrow (L^2 + \alpha^{m/2}) =: \tilde{L}^2$$
$$\mu \leftarrow \mu - \alpha^m \sigma C_\chi =: \tilde{\mu}_m$$
$$\gamma_t \leftarrow \eta_t$$

We can use the same unrolling strategy as theirs (the only condition required to proceed is that $\tilde{\mu}_m > \tilde{L}$, which automatically holds since $\mu < L$), and we obtain

$$\delta_n \leqslant \begin{cases} 2\exp\left(4\tilde{L}C^2\varphi_{1-2\beta}(n)\right)\exp\left(-\frac{\tilde{\mu}_m C}{4}n^{1-\beta}\right)\left(\delta_0 + \frac{\tilde{\sigma}_m^2}{\tilde{L}^2}\right) + \frac{4C\tilde{\sigma}_m^2}{\tilde{\mu}_m n^\beta}, & \text{if } 0 \leqslant \beta < 1 \\ \frac{\exp\left(2\tilde{L}^2 C^2\right)}{n^{\tilde{\mu}_m C}}\left(\delta_0 + \frac{\tilde{\sigma}_m^2}{\tilde{L}^2}\right) + 2\tilde{\sigma}_m^2 C^2 \frac{\varphi_{\tilde{\mu}_m C/2-1}(n)}{n^{\tilde{\mu}_m C/2}}, & \text{if } \beta = 1. \end{cases}$$

$\qquad\square$

## D.4 Proof of online CD with averaging (Theorem 3.3)

We first restate the theorem in its complete form.

**Theorem** (Contrastive Divergence with Polyak-Ruppert averaging). *Let $(\psi_t)_{t \geq 0}$ the sequence of iterates obtained by running the CD algorithm with a learning rate $\eta_t = Ct^{-\beta}$ for $\beta \in (\frac{1}{2}, 1)$. Define $\bar{\psi}_n := \frac{1}{n} \sum_{i=1}^n \psi_i$. Then, under the same assumptions as Theorem 3.2 we have, for all $n \geq 1$,*

$$\sqrt{\mathbb{E} \left\| \bar{\psi}_n - \psi^\star \right\|^2} \leq 2\sqrt{\frac{\mathrm{tr}(\mathcal{I}(\psi)^{-1})}{n}} + \mathcal{O}\left( n^{\max\left( -\left(\frac{1}{2} + \frac{\beta}{4}\right), -\beta, \frac{\beta}{2} - 1, -\left(\frac{\beta}{2} + m\frac{|\log \alpha|}{\log n}\right) \right)} \right)$$

*Where $\mathcal{I}(\psi^\star) := \mathrm{Cov}_{p_{\psi^\star}}[\phi]$ is the Fisher information matrix of the data distribution. Additionally, if $m > \frac{(1-\beta)\log n}{2|\log \alpha|}$, we have $\sqrt{\mathbb{E} \left\| \bar{\psi}_n - \psi^\star \right\|^2} \leq 2\sqrt{\frac{\mathrm{tr}(\mathcal{I}(\psi)^{-1})}{n}} + o\left( n^{-1/2} \right)$.*

Throughout the proof, we will denote by $h_n$ the standard online CD gradient defined in Equation 3:

$$h_n(\psi_{n-1}) = -\phi(X_n) + \phi(k_{\psi_{n-1}}^m(\cdot, X_n)), \quad X_n \sim p_{\psi^\star}, \quad \forall n \in \mathbb{N} \setminus \{0\},$$

as well as

$$\bar{h}(\psi_{n-1}) := \mathbb{E}\left[ h(\psi_{n-1}) \,|\, \psi_{n-1} \right] = \mathbb{E}_{p_{\psi^\star}} \phi(X_n) + \mathbb{E}_{p_{\psi^\star}} \mathbb{E}_{k_{\psi_{n-1}}^m} \phi(k_{\psi_{n-1}}^m(X_n, \cdot)).$$

We start by establishing some intermediate lemmas.

**Lemma D.5.** *Under Assumptions A1, A2, A3, the online CD iterates produced by Algorithm 1 using $\eta_t = Ct^{-\beta}$ for $\beta \in (\frac{1}{2}, 1)$ verify*

$$\frac{1}{n} \sqrt{\sum_{i=1}^n \left( \mathbb{E}\left[ \|\psi_i - \psi^\star\|^2 \right] \right)} = \mathcal{O}(n^{-\frac{1}{2} - \frac{\beta}{2}}).$$

*Proof.* Let us note $\delta_n = \mathbb{E}\left[ \|\psi_n - \psi^\star\|^2 \right]$. Summing the r.h.s of Theorem 3.2, we have

$$\sum_{i=1}^n \delta_i \leq \sum_{i=1}^n \frac{4C\tilde{\sigma}_m^2}{\tilde{\mu}_m i^\beta} + 2\left( \delta_0 + \frac{\tilde{\sigma}_m^2}{\tilde{L}^2} \right) \underbrace{\sum_{i=1}^n e^{4\tilde{L}C^2 \varphi_{1-2\beta}(i)} e^{-\frac{\tilde{\mu}_m C}{4} n^{1-\beta}}}_{A_3}$$

$$\implies \frac{1}{n} \sqrt{\sum_{i=1}^n \delta_i} \leq \frac{1}{n} \sqrt{\frac{4C\tilde{\sigma}_m^2}{\tilde{\mu}_m} \varphi_{1-\beta}(n)} + \frac{1}{n} \sqrt{\left( 2(\delta_0 + \frac{\tilde{\sigma}_m^2}{\tilde{L}^2}) A_3 \right)} = \mathcal{O}\left( n^{-\frac{1}{2} - \frac{\beta}{2}} \right).$$

where $A_3$ is finite if $\beta < 1$, and $A_3 = O(n)$ otherwise [22]. $\qquad \square$

**Lemma D.6.** *Under Assumptions A1, A2, A3, the online CD iterates produced by Algorithm 1 using $\eta_t = Ct^{-\beta}$ for $\beta \in (\frac{1}{2}, 1)$ verify*

$$\frac{1}{n} \sqrt{\sum_{i=1}^n \left( \mathbb{E}\left[ \|\psi_i - \psi^\star\|^2 \right] \right)^{1/2}} = \mathcal{O}(n^{-\frac{1}{2} - \frac{\beta}{4}}).$$

*Proof.* Let us note $\delta_n = \mathbb{E}\left[ \|\psi_n - \psi^\star\|^2 \right]$. Applying $\sqrt{x+y} \leq \sqrt{x} + \sqrt{y}$ to the r.h.s of Theorem 3.2, we have

$$\sum_{i=1}^n \delta_i^{1/2} \leq \frac{2C^{1/2}\tilde{\sigma}_m}{2\tilde{\mu}_m^{1/2}} \sum_{i=1}^n i^{-\beta/2} + \sqrt{2\left( \delta_0 + \frac{\tilde{\sigma}_m^2}{\tilde{L}^2} \right)} \underbrace{\sum_{i=1}^n e^{2\tilde{L}^2 C^2 \varphi_{1-2\beta}(i)} e^{-\frac{\tilde{\mu}_m C}{8} i^{1-\beta}}}_{A_4}$$

$$\implies \frac{1}{n} \sqrt{\sum_{i=1}^n \delta_i^{1/2}} \leq \frac{1}{n} \sqrt{\frac{2C^{1/2}\tilde{\sigma}_m}{\tilde{\mu}_m^{1/2}} \varphi_{1-\beta/2}(n)} + \frac{1}{n} \left( \sqrt{2\left( \delta_0 + \frac{\tilde{\sigma}_m^2}{\tilde{L}^2} \right)} A_4 \right)^{1/2} = \mathcal{O}\left( n^{-\frac{1}{2} - \frac{\beta}{4}} \right).$$

where $A_4$ is finite if $\beta < 1$, and $A_4 = O(n)$ otherwise [22]. $\qquad \square$

**Lemma D.7.** *Under Assumptions A1, A2, A3, the online CD iterates produced by Algorithm 1 using $\eta_t = Ct^{-\beta}$ for $\beta \in (\frac{1}{2}, 1)$ verify*

$$\sqrt{\mathbb{E}\left[\left\|\frac{1}{n}\sum_{i=1}^{n} h_i(\psi_{i-1})\right\|^2\right]} = \mathcal{O}\left(n^{\frac{\beta}{2}-1}\right).$$

*Proof.* The result follows from the fact that $h_t$ verifies

$$h_t(\psi_{t-1}) = \frac{1}{\eta_t}(\psi_{t-1} - \psi_t).$$

A similar quantity was handled in the case of standard SGD [22, Theorem 3], and the only condition needed to reuse their steps is that $(\psi_t)_{t\leq n}$ satisfies an upper bound of the same form as the one [22, Theorem 1] derived. This is precisely the nature of our bound of $\psi_t$ estblished in Theorem 3.2, with $\tilde{\mu}_m, \tilde{\sigma}_m, \tilde{L}$. Borrowing on their result, we have:

$$\sqrt{\mathbb{E}\left[\left\|\frac{1}{n}\sum_{i=1}^{n} h_i(\psi_{i-1})\right\|^2\right]} \leq \frac{4\tilde{\sigma}_m\beta}{C^{1/2}n\tilde{\mu}_m}\varphi_{\beta/2}(n) + \frac{4\beta}{Cn\tilde{\mu}_m^{1/2}}\left(\delta_0 + \frac{\tilde{\sigma}_m^2}{\tilde{L}^2}\right)^{1/2}A_2$$

$$+ \frac{1}{n\tilde{\mu}_m^{1/2}}\left(\frac{1}{C} + 2\tilde{L}\right)\delta_0^{1/2} + \frac{2\tilde{L}}{n\tilde{\mu}_m^{1/2}}\frac{2C^{1/2}\tilde{\sigma}_m}{\tilde{\mu}_m^{1/2}}\varphi_{1-\beta}(n)^{1/2}$$

$$+ \frac{4\tilde{L}}{n\tilde{\mu}_m^{1/2}}\left(\delta_0 + \frac{\tilde{\sigma}_m^2}{\tilde{L}^2}\right)^{1/2}A_2^{1/2}$$

where $\tilde{\mu}_m, \tilde{\sigma}_m$ and $\tilde{L}$ are defined in 3.2, and

$$A_2 = \sum_{k=1}^{n} e^{\frac{-\tilde{\mu}_m C}{16}k^{1-\beta} + 16\tilde{L}_1^4 C^4 \varphi_{1-2\beta}(k)} \qquad \square$$

**Lemma D.8.** *Under Assumptions A1, A2, A3, the online CD iterates produced by Algorithm 1 using $\eta_t = Ct^{-\beta}$ for $\beta \in (\frac{1}{2}, 1)$ verify*

$$\frac{1}{n}\sum_{i=1}^{n}\sqrt{\mathbb{E}\left[\|\psi_i - \psi^\star\|^4\right]} = \mathcal{O}(n^{-\beta}).$$

*Proof.* We proceed as in the proof of [22, Theorem 3], first establishing a recurrence for $\mathbb{E}\left[\|\psi_i - \psi^\star\|^4\right]$, and then unrolling it. We have

$$\mathbb{E}\left[\|\psi_n - \psi^*\|^4 \mid \mathcal{F}_{n-1}\right] \leqslant \|\psi_{n-1} - \psi^*\|^4 + 6\eta_n^2\|\psi_{n-1} - \psi^*\|^2\,\mathbb{E}\left[\|h_n(\psi_{n-1})\|^2 \mid \mathcal{F}_{n-1}\right]$$

$$+ \eta_n^4 \mathbb{E}\left[\|h_n(\psi_{n-1})\|^4 \mid \mathcal{F}_{n-1}\right]$$

$$- 4\eta_n\|\psi_{n-1} - \psi^*\|^2 \langle\psi_{n-1} - \psi^*, \mathbb{E}[h_n]\rangle$$

$$+ 4\eta_n^3\|\psi_{n-1} - \psi^*\|\,\mathbb{E}\left[\|h_n(\psi_{n-1})\|^3 \mid \mathcal{F}_{n-1}\right].$$

The second and fourth terms will be controlled using results from our previous sections. For simplicity, we don't attempt to relate the moments of $\|h_n\|^4$ as precisely as before. Instead we use:

$$\mathbb{E}_{p_{\psi^\star}}\left[\|h_n\|^k\right] \leq 2^{k-1}\left(\mathbb{E}_{p_{\psi^\star}}\mathbb{E}_{k_{p_{\psi_{n-1}}}^m}\left[\|\phi(k_{\psi_{n-1}}^m(X_n,\cdot))\|^k\right] + \mathbb{E}_{p_{\psi^\star}}\left[\|\phi(X_n)\|^k\right]\right)$$

And let us note $\tau = \left(\sup_{\psi\in\Psi}\mathbb{E}_{p_\psi}\|\phi\|^8\right)^{1/8}$. We have, for $k \geq 4$,

$$\mathbb{E}_{p_{\psi^\star}}\mathbb{E}_{k_{p_{\psi_{n-1}}}^m}\|\phi(k_{\psi_{n-1}}^m(X_n,\cdot))\|^k \leq \mathbb{E}_{p_{\psi_{n-1}}}\|\phi\|^k + C_\chi(\mathbb{E}_{p_{\psi_{n-1}}}(\|\phi\|^k - \mathbb{E}_{p_{\psi_{n-1}}}\|\phi\|^k)^2)^{1/2}\|\psi_{n-1} - \psi^\star\|$$

$$\leq \tau^k + 2C_\chi\tau^k\|\psi_{n-1} - \psi^\star\|$$

Where we used the fact that $P_{\psi_{n-1}}^m$ is a contraction, and $(\mathbb{E}\,\|\phi\|^k)^{1/k}$ is an increasing function of $k$. On the other hand, we simply have $\mathbb{E}_{p_{\psi^\star}}\|\phi(X_n,\cdot)\|^k \le \tau^k$. Plugging this into the previous equation, we obtain

$$
\begin{aligned}
\mathbb{E}\left[\|\psi_n - \psi^*\|^4 \mid \mathcal{F}_{n-1}\right] \le{}& \|\psi_{n-1} - \psi^*\|^4 \\
&+ 6\eta_n^2 \|\psi_{n-1} - \psi^*\|^2 \left(4\tau^2 + 4C_\chi\tau^2 \|\psi_{n-1} - \psi^\star\|\right) \\
&+ \eta_n^4(16\tau^4 + 16C_\chi\tau^4 \|\psi_{n-1} - \psi^\star\|) \\
&- 4\eta_n \|\psi_{n-1} - \psi^*\|^2 \langle\psi_{n-1} - \psi^*, \mathbb{E}\left[h_n|\mathcal{F}_{n-1}\right]\rangle \\
&+ 4\eta_n^3 \|\psi_{n-1} - \psi^*\| \left(8\tau^3 + 8C_\chi\tau^3 \|\psi_{n-1} - \psi^\star\|\right)
\end{aligned}
$$

To simplify the recursion, we use the four following inequalities:

$$
\tau^2\eta_n^2 \|\psi_{n-1} - \psi^\star\|^3 \le \frac{1}{2}(\tau^2\eta_n^2(\|\psi_{n-1} - \psi^\star\|^2 + \tau^2\eta_n^2 \|\psi_{n-1} - \psi^\star\|^4)
$$

$$
\tau^4\eta_n^4 \|\psi_{n-1} - \psi^\star\| \le (\tau^4\eta_n^4 + \frac{1}{4}\tau^4\eta_n^4 \|\psi_{n-1} - \psi^\star\|^4))
$$

$$
\eta_n^3\tau^3 \|\psi_{n-1} - \psi^\star\| \le \frac{1}{2}(\eta_n^2\tau^2 \|\psi_{n-1} - \psi^\star\|^2 + 16\eta_n^4\tau^4
$$

$$
\tau^3\eta_n^3 \|\psi_{n-1} - \psi^\star\|^2 \le \frac{1}{2}(\eta_n^4\tau^4 + \|\psi_{n-1} - \psi^\star\|^2 \eta_n^2\tau^2)
$$

Injecting them in our recursion, we obtain:

$$
\begin{aligned}
\mathbb{E}\left[\|\psi_n - \psi^*\|^4 \mid \mathcal{F}_{n-1}\right] \le{}& \|\psi_{n-1} - \psi^*\|^4 \\
&+ 12\eta_n^2 \|\psi_{n-1} - \psi^*\|^2 \tau^2 \\
&+ 12C_\chi\tau^2\eta_n^2\|\psi_{n-1} - \psi^\star\|^2 \\
&+ 12C_\chi\tau^2\eta_n^2\|\psi_{n-1} - \psi^\star\|^4 \\
&+ 16\eta_n^4\tau^4 \\
&+ 16C_\chi\eta_n^4\tau^4 \\
&+ 4C_\chi\eta_n^4\tau^4 \|\psi_{n-1} - \psi^\star\|^4 \\
&- 4\eta_n\tilde{\mu}_m \|\psi_{n-1} - \psi^*\|^4 \\
&+ 16\eta_n^2\tau^2 \|\psi_{n-1} - \psi^*\|^2 \\
&+ 16\eta_n^4\tau^4 \\
&+ 16\eta_n^4 C_\chi\tau^4 \\
&+ 16\eta_n^2 C_\chi\tau^2 \|\psi_{n-1} - \psi^\star\|^2 \ ,
\end{aligned}
$$

which, after further simplifications, yields

$$
\mathbb{E}\big[\,\|\psi_n - \psi^*\|^4 \mid \mathcal{F}_{n-1}\big]
$$

$$
\leq \|\psi_{n-1} - \psi^\star\|^4 \left(1 - 4\eta_n\tilde{\mu}_m + 12C_\chi\eta_n^2\tau^2 + 4C_\chi\tau^4\eta_n^4\right)
$$

$$
+ \eta_n^2\|\psi_{n-1} - \psi^\star\|^2 \left(28(1 + C_\chi)\tau^2\right) + 32\eta_n^4(\tau^4(1 + C_\chi))
$$

$$
\leq \|\psi_{n-1} - \psi^\star\|^4 \left(1 - 4\eta_n\tilde{\mu}_m + 12(1 + C_\chi)\eta_n^2\tau^2 + 4(1 + C_\chi)\tau^4\eta_n^4\right)
$$

$$
+ 28\eta_n^2\|\psi_{n-1} - \psi^\star\|^2 \left((1 + C_\chi)\tau\right)^2 + 32\eta_n^4(1 + C_\chi)\tau)^4
$$

$$
\leq \|\psi_{n-1} - \psi^\star\|^4 \left(1 - 4\eta_n\tilde{\mu}_m + 12\eta_n^2((1 + C_\chi)\tau)^2 + 4((1 + C_\chi)\tau)^4\eta_n^4\right)
$$

$$
+ 28\eta_n^2\|\psi_{n-1} - \psi^\star\|^2 \left((1 + C_\chi)\tau\right)^2 + 32\eta_n^4(1 + C_\chi)\tau)^4
$$

$$
\leq \|\psi_{n-1} - \psi^\star\|^4 \left(1 - 4\eta_n\tilde{\mu}_m + 12\eta_n^2(2(1 + C_\chi)\tau)^2 + 16\eta_n^2(2(1 + C_\chi)\tau)^3\right.
$$

$$
+ 4(2(1 + C_\chi)\tau)^4\eta_n^4) + 20\eta_n^2\|\psi_{n-1} - \psi^\star\|^2 \left(2(1 + C_\chi)\tau\right)^2 + 16\eta_n^4(2(1 + C_\chi)\tau)^4
$$

$$
\leq \|\psi_{n-1} - \psi^\star\|^4 \left(1 - 4\eta_n\tilde{\mu}_m + 12\eta_n^2(2(1 + C_\chi)\tau + L)^2 + 16\eta_n^2(2(1 + C_\chi)\tau + L)^3\right.
$$

$$
+ 4(2(1 + C_\chi)\tau + L)^4\eta_n^4) + 20\eta_n^2\|\psi_{n-1} - \psi^\star\|^2 \left(2(1 + C_\chi)\tau\right)^2 + 16\eta_n^4(2(1 + C_\chi)\tau)^4
$$

$$
\leq \|\psi_{n-1} - \psi^\star\|^4 \left(1 - 4\eta_n\tilde{\mu}_m + 12\eta_n^2\tilde{L}_1^2 + 16\eta_n^2\tilde{L}_1^3 + 4\tilde{L}_1^4\eta_n^4\right)
$$

$$
+ 20\eta_n^2\|\psi_{n-1} - \psi^\star\|^2 \tilde{\tau}_1^2 + 16\eta_n^4\tilde{\tau}_2^4
$$

where we defined $\tilde{\tau}_1 := 2(1 + C_\chi)\tau$ and $\tilde{L}_1 := 2(1 + C_\chi)\tau + L$. This recursion is of the form of the one studied in [22, Equation 32] (note that by design, $\tilde{L} \geq \tilde{\mu}_m$.) The steps performed to bound $\mathbb{E}\left[\|\psi_i - \psi^\star\|^4\right]$ thus follow from their derivations, and we obtain:

$$
\frac{1}{n}\sqrt{\sum_{i=1}^n \mathbb{E}\|\psi_{i-1} - \psi^\star\|^4}
$$

$$
\leqslant \frac{C\tilde{\tau}_1^2}{2n}\left(C^{1/2}\varphi_{1-3\beta/2}(n) + \tilde{\mu}_m^{-1/2}\varphi_{1-\beta}(n)\right)
$$

$$
+ \frac{\sqrt{20}C^{1/2}\tilde{\tau}_1}{2n}A_1\exp\left(24\tilde{L}_1^4C^4\right)\left(\delta_0 + \frac{\tilde{\mu}_m\mathbb{E}\|\theta_0 - \theta^*\|^4}{20C\tilde{\tau}_1^2} + 2\tilde{\tau}_1^2C^3\tilde{\mu}_m + 8\tilde{\tau}_1^2C^2\right)^{1/2}
$$

$$
= \mathcal{O}\left(n^{-\beta}\right),
$$

where

$$
A_1 = \sum_{k=1}^n e^{\frac{-\tilde{\mu}C}{16}k^{1-\beta} + 16\tilde{L}_1^4C^4\varphi_{1-2\beta}(k)}
$$

and we have $A(1) < +\infty$ if $\beta < 1$, and $A(1) = O(n)$ otherwise. $\qquad\square$

**Lemma D.9.** *For all $\psi \in \Psi$, we have:*

$$
\left\|\mathrm{Cov}\left[\phi(k_\psi^m(X_n, \cdot))\right] - \mathrm{Cov}_{p_\psi}\left[\phi(X_n)\right]\right\|_{\mathrm{F}}
$$

$$
\leq \alpha^m C_\chi(\bar{\tau}^{1/2} + 2\|\log Z\|_{4,\infty}\sigma)\|\psi - \psi^\star\| + \alpha^{2m}C_\chi^2\sigma^2\|\psi - \psi^\star\|^2.
$$

*where $\bar{\tau} := \sup_{\psi \in \Psi}\mathbb{E}_{p_\psi}\left[\|\phi\phi^\top - \mathbb{E}_{p_\psi}[\phi\phi^\top]\|_F^2\right] < +\infty$ and $\|\log Z\|_{4,\infty} := \sup_{\psi \in \Psi}\|\mathbb{E}_\psi\phi\| \leq \sup_{\psi \in \Psi}\sum_{i=1}^d \partial_i^2 \log Z(\psi)^2$.*

*Proof.* We have

$$
\mathrm{Cov}\left[\phi(k_\psi^m(X_n, \cdot))\right] = \mathbb{E}_{p_{\psi^\star}}P_\psi^m\left[\phi\phi^\top\right] - \left(\mathbb{E}_{p_{\psi^\star}}[P_\psi^m\phi]\right)\left(\mathbb{E}_{p_{\psi^\star}}[P_\psi^m\phi]^\top\right)
$$

Looking at the second moment first, we have

$$
\mathbb{E}_{p_{\psi^\star}}\left[P_\psi^m\phi\phi^\top\right] - \mathbb{E}_{p_\psi}\phi\phi^\top = \underbrace{\mathbb{E}_{p_{\psi^\star}}P_\psi^m(\phi\phi^\top - \mathbb{E}_{p_\psi}\phi\phi^\top)}_{\Delta_1}.
$$

Applying lemma C.7 to the $\mathbb{R}^{d^2}$-valued function $f$ given by $f_{ij} := \phi_i\phi_j - \mathbb{E}_{p_\psi}\phi_i\phi_j$,

$$
\|\Delta_1\|_{\mathrm{F}} \leq \|\psi - \psi^\star\|\alpha^m C_\chi\sqrt{\mathbb{E}_{p_\psi}\left[\|\phi\phi^\top - \mathbb{E}_{p_\psi}\phi\phi^\top\|_F^2\right]} \leq \bar{\tau}^{1/2}\alpha^m C_\chi\|\psi - \psi^\star\|.
$$

We now investigate the first moment. We have

$$\mathbb{E}_{p_{\psi^\star}}\left[P_\psi^m \phi\right] = \underbrace{\mathbb{E}_{p_{\psi^\star}}\left[P_\psi^m \phi\right] - \mathbb{E}_{p_\psi}\left[\phi\right]}_{\Delta_{2,1}} + \mathbb{E}_{p_{\psi_n}}\phi$$

$$\implies \underbrace{(\mathbb{E}_{p_{\psi^\star}} P_\psi^m \phi)(\mathbb{E}_{p_{\psi^\star}} P_\psi^m \phi)^\top - \mathbb{E}_{p_\psi}\phi\mathbb{E}_{p_\psi}\phi^\top}_{\Delta_2} = \Delta_{2,1}\Delta_{2,1}^T + \Delta_{2,1}\mathbb{E}_{p_\psi}\phi^\top + \mathbb{E}_{p_\psi}\phi\Delta_{2,1}^\top$$

and thus, applying Lemma C.7 on $\Delta_{2,1}$, we have:

$$\begin{aligned}
\|\Delta_2\|_F &\leq \left\|\Delta_{2,1}\Delta_{2,1}^\top + \Delta_{2,1}\mathbb{E}_{p_\psi}\left[\phi^\top\right] + \mathbb{E}_{p_\psi}\left[\phi\right]\Delta_{2,1}^\top\right\|_F \\
&\leq \|\Delta_{2,1}\Delta_{2,1}^\top\|_F + 2\|\Delta_{2,1}\|\|\mathbb{E}_{p_\psi}\left[\phi\right]\| \\
&\leq \alpha^{2m}\sigma^2 C_\chi^2\|\psi - \psi^\star\|^2 + 2\|\log Z\|_{4,\infty}\alpha^m C_\chi\sigma\|\psi - \psi^\star\|,
\end{aligned}$$

where we used that $\|\Delta_{2,1}\Delta_{2,1}^\top\|_F = \|\Delta_{2,1}\|^2$. We can now combine our two matrix moment bounds to obtain

$$\begin{aligned}
\left\|\text{Cov}\left[\phi(k_\psi^m(X_n,\cdot))\right] - \text{Cov}_{p_\psi}\left[\phi(X_n)\right]\right\|_F &= \|\Delta_1 + \Delta_2\|_F \\
&\leq \alpha^m C_\chi(\bar\tau^{1/2} + 2\|\log Z\|_{4,\infty}\sigma)\|\psi - \psi^\star\| + \alpha^{2m}C_\chi^2\sigma^2\|\psi - \psi^\star\|^2.
\end{aligned}$$

$\square$

**Lemma D.10.** *Under Assumptions A1, A2, A3 it holds that*

$$\begin{aligned}
\sqrt{\mathbb{E}\left[\left\|\frac{1}{n}\sum_{i=1}^n f''(\psi^\star)^{-1}(\overline{h}(\psi_{i-1}) - h_i(\psi_{i-1}))\right\|^2\right]} &\leq 2\sqrt{\frac{\text{tr}(\mathcal{I}(\psi^\star)^{-1})}{n}} \\
&+ \frac{\|\mathcal{I}(\psi^\star)^{-2}\|_F^{1/2}}{n}\left((M + \alpha^m C_\chi(\bar\tau^{1/2} + 2\|\log Z\|_{4,\infty}\sigma))^{1/2}\left(\sum_{i=1}^n \delta_{i-1}^{1/2}\right)^{1/2}\right. \\
&\left.+ \alpha^{2m}C_\chi^2\sigma^2\left(\sum_{i=1}^n \delta_{i-1}\right)^{1/2}\right)
\end{aligned}$$

*where* $M := \sup_{\psi\in\Psi}\left\|\nabla^3\log Z(\psi)\right\|_{\text{op}(\|\cdot\|_F,\|\cdot\|_F)} < +\infty$.

*Proof.*
$$\begin{aligned}
f''(\psi^\star)^{-1}&(\overline{h}(\psi_{n-1}) - h_n(\psi_{n-1})) \\
&= f''(\psi^\star)^{-1}\underbrace{(\phi(X_n) - \mathbb{E}_{p_{\psi^\star}}\phi)}_{\Delta_{1,n}} + f''(\psi^\star)^{-1}\underbrace{(\phi(k_{\psi_{n-1}}^m(X_n)) - \mathbb{E}_{p_{\psi^\star}}\mathbb{E}_{k_{\psi_{n-1}}^m}\phi(k_{\psi_{n-1}}^m(X_n,\cdot)))}_{\Delta_{2,n}} \\
&= f''(\psi^\star)^{-1}\Delta_{1,n} + f''(\psi^\star)^{-1}\Delta_{2,n}
\end{aligned}$$

Noting $\Delta$ the l.h.s of Lemma D.10, we have, summing over $[n]$, and using Minkowski's inequality,

$$\Delta \leq \frac{1}{n}\sqrt{\mathbb{E}\left\|\sum_{i=1}^n f''(\psi^\star)^{-1}\Delta_{1,i}\right\|^2} + \frac{1}{n}\sqrt{\mathbb{E}\left\|\sum_{i=1}^n f''(\psi^\star)^{-1}\Delta_{2,i}\right\|^2}$$

Note that this step was made possible because we are looking at the square-root of the variance, which is unlike the recursion in Lemma 3.1. This allows to separate the terms and use fewer intermediaries than in the proof of Lemma 3.1.

Since both $\Delta_{1,n}$ and $\Delta_{2,n}$ are martingale differences with respect to the filtration $\mathcal{F}_{n-1}$, the covariance terms vanish, and we have

$$
\begin{aligned}
\Delta &\leq \frac{1}{n}\sqrt{\sum_{i=1}^n \mathbb{E}\left[\|f''(\psi^\star)^{-1}\Delta_{1,i}\|^2\right]} + \frac{1}{n}\sqrt{\sum_{i=1}^n \mathbb{E}\left[\|f''(\psi^\star)^{-1}\Delta_{2,i}\|^2\right]} \\
&\leq \frac{1}{n}\sqrt{\sum_{i=1}^n \operatorname{tr}(f''(\psi^\star)^{-1}\mathbb{E}\left[\Delta_{1,i}\Delta_{1,i}^\top f''(\psi^\star)^{-1})\right]} \\
&\quad + \frac{1}{n}\sqrt{\sum_{i=1}^n \operatorname{tr}(f''(\psi^\star)^{-1}\mathbb{E}\left[\Delta_{2,i}\Delta_{2,i}^\top f''(\psi^\star)^{-1})\right]} \\
&\leq \sqrt{\frac{\operatorname{tr}(\mathcal{I}(\psi^\star)^{-1})}{n}} + \frac{1}{n}\sqrt{\sum_{i=1}^n \operatorname{tr}(f''(\psi^\star)^{-1}\operatorname{Cov}\left[\phi(k_{\psi_{i-1}}^m(x_i))\right]f''(\psi^\star)^{-1})} \\
&\leq \sqrt{\frac{\operatorname{tr}(\mathcal{I}(\psi^\star)^{-1})}{n}} + \frac{1}{n}\Big(\sum_{i=1}^n \operatorname{tr}(f''(\psi^\star)^{-1}(\operatorname{Cov}_{p_{\psi^\star}}\phi + (\operatorname{Cov}_{p_{\psi_{i-1}}}\phi - \operatorname{Cov}_{p_{\psi^\star}}\phi) \\
&\quad + (\operatorname{Cov}\phi(k_{\psi_{i-1}}^m(x_i)) - \operatorname{Cov}_{p_{\psi_{i-1}}}\phi))f''(\psi^\star)^{-1})\Big)^{1/2} \\
&\leq 2\sqrt{\frac{\operatorname{tr}(\mathcal{I}(\psi^\star)^{-1})}{n}} + \frac{\sqrt{\operatorname{tr}(\mathcal{I}(\psi^\star)^{-2})}}{n}\Big(\sum_{i=1}^n \|(\operatorname{Cov}_{p_{\psi_{n-1}}}\phi - \operatorname{Cov}_{p_{\psi^\star}}\phi\|_{\mathrm{F}} \\
&\quad + \|\operatorname{Cov}\phi(k_\psi^m(x_i)) - \operatorname{Cov}_{p_\psi}\phi)\|_{\mathrm{F}}\Big)^{1/2} \\
&\overset{(a)}{\leq} 2\sqrt{\frac{\operatorname{tr}(\mathcal{I}(\psi^\star)^{-1})}{n}} + \frac{\sqrt{\operatorname{tr}(\mathcal{I}(\psi^\star)^{-2})}}{n}\Big((M + \alpha^m C_\chi(\bar{\tau}^{1/2} + 2\|\log Z\|_{4,\infty}\sigma)^{1/2})\times \\
&\quad \sqrt{\sum_{i=1}^n \|\psi_{i-1} - \psi^\star\|} + \alpha^m C_\chi\sigma\sqrt{\sum_{i=1}^n \|\psi_{i-1} - \psi^\star\|^2}\Big) \\
&\overset{(b)}{\leq} 2\sqrt{\frac{\operatorname{tr}(\mathcal{I}(\psi^\star)^{-1})}{n}} + \frac{\sqrt{\operatorname{tr}(\mathcal{I}(\psi^\star)^{-2})}}{n}\Big((M + \alpha^m C_\chi(\bar{\tau}^{1/2} + 2\|\log z\|_{4,\infty}\sigma))^{1/2}\sqrt{\sum_{i=1}^n \delta_i^{1/2}} \\
&\quad + \alpha^m C_\chi\sigma\sqrt{\sum_{i=1}^n \delta_i}\Big)
\end{aligned}
$$

In $(a)$ we used Lemma D.10, the cyclicity of the trace, $\operatorname{tr}(A^\top B) \leq \|AB\|_{\mathrm{F}} \leq \|A\|_{\mathrm{F}}\|B\|_{\mathrm{F}}$, and the fact that since $\operatorname{Cov}_{p_{\psi_{i-1}}}[\phi(x_i)] = \nabla_\psi^2 \mathcal{L}(\psi_{i-1})$, by analycity of $\mathcal{L}$, there exists a constant $M := \sup_{\psi \in \Psi}\|\nabla^3 \log Z(\psi)\|_{\mathrm{op}(\|\cdot\|,\|\cdot\|_F)}$ such that

$$
\|\nabla_\psi^2 \mathcal{L}(\psi_{i-1}) - \nabla_\psi^2 \mathcal{L}(\psi^\star)\|_{\mathrm{F}} \leq M\|\psi_{i-1} - \psi^\star\|.
$$

In $(b)$ we used Jensen's inequality to get $\mathbb{E}\left[\|\psi_{i-1} - \psi^\star\|\right] \leq \sqrt{\mathbb{E}\left[\|\psi_{i-1} - \psi^\star\|^2\right]} = \delta_{i-1}^{1/2}$. $\qquad\square$

We are now ready to prove Theorem 3.3.

*Proof of Theorem 3.3.* It holds that:

$$
\begin{aligned}
f''(\psi^\star)(\psi_{n-1} - \psi^\star) &= f'(\psi_{n-1}) - f'(\psi^\star) + (f''(\psi^\star)(\psi_{n-1} - \psi^\star) - f'(\psi_{n-1}) + f'(\psi^\star)) \\
&= h_n(\psi_{n-1}) - f'(\psi^\star) + (f''(\psi^\star)(\psi_{n-1} - \psi^\star) - f'(\psi_{n-1}) + f'(\psi^\star)) \\
&\quad + (f'(\psi_{n-1}) - \bar{h}(\psi_{n-1})) + (\bar{h}(\psi_{n-1}) - h_n(\psi_{n-1})).
\end{aligned}
$$

Applying on both sides: (a) a summation over $i \in [n]$, (b) a multiplication by $f''(\psi^\star)^{-1}$, (c) $\sqrt{\mathbb{E}[\|\cdot\|^2]}$, and using Minkowski's inequality on the r.h.s, we obtain

$$\sqrt{\mathbb{E}\left[\left\|\frac{1}{n}\sum_{i=1}^n \psi_i - \psi^\star\right\|^2\right]} \leq \underbrace{\sqrt{\mathbb{E}\left[\left\|\frac{1}{n}\sum_{i=1}^n f''(\psi^\star)^{-1}h_i(\psi_{i-1})\right\|^2\right]}}_{(i)}$$

$$+ \underbrace{\sqrt{\mathbb{E}\left[\left\|\frac{1}{n}\sum_{i=1}^n f''(\psi^\star)^{-1}(f''(\psi^\star)(\psi_{i-1}-\psi^\star) - f'(\psi_{i-1}))\right\|^2\right]}}_{(ii)}$$

$$+ \underbrace{\sqrt{\mathbb{E}\left[\left\|\frac{1}{n}\sum_{i=1}^n f''(\psi^\star)^{-1}(f'(\psi_{i-1}) - \bar{h}(\psi_{i-1}))\right\|^2\right]}}_{(iii)}$$

$$+ \underbrace{\sqrt{\mathbb{E}\left[\left\|\frac{1}{n}\sum_{i=1}^n f''(\psi^\star)^{-1}(\overline{h}(\psi_{i-1}) - h_i(\psi_{i-1}))\right\|^2\right]}}_{(iv)}.$$

$(i)$ and $(ii)$ have direct analogues in the proofs of prior work [22] on the convergence of (unbiased) SGD with Polyak-Ruppert averaging, and will be bounded similarly. $(iii)$ captures the bias of the CD algorithm, while $(iv)$ captures the variance.

**Bounding $(i)$**    Using Lemma D.7, we have $(i) = \mathcal{O}(n^{\frac{\beta}{2}-1})$.

**Bounding $(ii)$**    Since $\log Z(\psi)$ is analytic, there exists some constant $M'$ such that

$$\|f''(\psi^\star)(\psi_{i-1}-\psi^\star) - f'(\psi_{i-1})\| \leq M'\|\psi_{i-1}-\psi^\star\|^2.$$

Thus, we have:

$$(ii) \leq \frac{M'}{n}\sqrt{\mathbb{E}\left[\left(\sum_{i=1}^n \|\psi_i - \psi^\star\|^2\right)^2\right]} \leq \frac{M'}{n}\sum_{i=1}^n \sqrt{\mathbb{E}\left[\|\psi_i - \psi^\star\|^4\right]} = \mathcal{O}(n^{-\beta})$$

where the second-to-last inequality used Minkowski's inequality, and the last applied Lemma D.8.

**Bounding $(iii)$**    By Minkowski's inequality, we have:

$$(iii) \leq \frac{1}{n}\sum_{i=1}^n \sqrt{\mathbb{E}\left[\|f'(\psi_{n-1}) - \bar{h}(\psi_{n-1})\|^2\right]}$$

Moreover, using lemma C.7, we have:

$$\|f'(\psi_{n-1}) - \bar{h}(\psi_{n-1})\| = \left\|\mathbb{E}_{p_{\psi_{n-1}}}\phi - \mathbb{E}_{p_{\psi^\star}}\mathbb{E}_{k^m_{\psi_{n-1}}}\phi\right\|$$

$$\leq \alpha^m\sqrt{\mathbb{E}_{p_{\psi_{n-1}}}\left\|\phi - \mathbb{E}_{p_{\psi_{n-1}}}\phi\right\|^2}C_\chi\|\psi_{n-1}-\psi^\star\|$$

$$\leq \alpha^m\sigma C_\chi\|\psi_{n-1}-\psi^\star\|.$$

We thus obtain

$$(iii) \leq \frac{\alpha^m C_\chi}{n}\sum_{i=1}^n (\mathbb{E}\|\psi_{i-1}-\psi^\star\|^2)^{1/2} = \frac{\alpha^m C_\chi}{n}\sum_{i=1}^n \delta_{i-1}^{1/2}.$$

Recalling that $\delta_i$ satisfies Theorem 3.2, we have that $\delta_n^{\frac{1}{2}} = \mathcal{O}(n^{-\beta/2})$, and we thus have $\sum_{i=1}^n \delta_i^{1/2} = \mathcal{O}(n^{1-\frac{\beta}{2}})$. By squaring the result of Lemma D.6, we have $\sum_{i=1}^n \delta_i^{1/2} = \mathcal{O}(n^{-1-\frac{\beta}{2}})$, and thus, we obtain that $(iii) = \mathcal{O}(\alpha^m n^{-\beta/2}) = \mathcal{O}(n^{-(\frac{\beta}{2}+m\frac{|\log \alpha|}{\log n})})$.

**Bounding** $(iv)$    We have

$$(iv)$$

$$\overset{(a)}{\leq} 2\sqrt{\frac{\operatorname{tr}\left(\mathcal{I}(\psi^\star)^{-1}\right)}{n}} + \frac{\|\mathcal{I}(\psi^\star)^{-2}\|_{\mathrm{F}}^{1/2}}{n}\left((M + \alpha^m C_\chi(\bar{\tau} + 2\|\log Z\|_{4,\infty}\sigma))^{1/2}\sqrt{\sum_{i=1}^n \delta_{i-1}^{1/2}}\right.$$

$$\left.+\alpha^m C_\chi \sigma \sqrt{\sum_{i=1}^n \delta_{i-1}}\right)$$

$$\overset{(b)}{\leq} 2\sqrt{\frac{\operatorname{tr}\left(\mathcal{I}(\psi^\star)^{-1}\right)}{n}} + \mathcal{O}(n^{-\frac{1}{2}-\frac{\beta}{4}}).$$

Where in $(a)$, we used Lemma D.10 and in $(b)$, we used Lemma D.6 and D.5.

**Final bound**    Putting everything together, we have that:

$$\sqrt{\mathbb{E}\left\|\overline{\psi}_n - \psi^\star\right\|^2} \leq 2\sqrt{\frac{\operatorname{tr}(\mathcal{I}(\psi)^{-1})}{n}} + \mathcal{O}(n^{\max\left(-\left(\frac{1}{2}+\frac{\beta}{4}\right), -\beta, \frac{\beta}{2}-1, -\left(\frac{\beta}{2}+m\frac{|\log\alpha|}{\log n}\right)\right)}).$$

If, furthermore, $m > \frac{(1-\beta)\log n}{2|\log\alpha|}$, we have

$$\max\left(-\left(\frac{1}{2} + \frac{\beta}{4}\right), -\beta, \frac{\beta}{2} - 1, -\left(\frac{\beta}{2} + m\frac{|\log\alpha|}{\log n}\right)\right) < -\frac{1}{2},$$

which concludes the proof. $\qquad\qquad\qquad\qquad\qquad\qquad\qquad\qquad\qquad\qquad\qquad\qquad\square$

## E    $L_2$ approximation by auxiliary gradient updates

In this section, we consider different gradient update schemes starting from some random initialization $\theta_{\mathrm{init}}$, and control the $L_2$ distance between the different updates and the deterministic target $\psi^* \in \Psi$.

**Notation.**    Recall the notation that $X_1, \ldots, X_n \overset{\mathrm{i.i.d.}}{\sim} p_{\psi^*}$, $B = n/N$ and for $\psi \in \Psi$, $K_\psi(x) \sim k_\psi^m(x, \bullet)$. We also write, for $m \in \mathbb{N} \cup \{\infty\}$,

$$K_{1;\psi}^m(x), \ldots, K_{n;\psi}^m(x) \overset{\mathrm{i.i.d.}}{\sim} k_\psi^m(x, \bullet)$$

Let $\theta^{\mathrm{init}}$ be some $\Psi$-valued random initialization that is possibly correlated with $X_1, \ldots, X_n$. We capture the effect of correlation through the following quantities: For $\epsilon > 0$ and $\nu > 2$, let

$$\vartheta_{n,m}^{\mathrm{init}}(\epsilon) := \mathbb{P}\left(\frac{\left\|\sum_{i=1}^n \mathbb{E}\left[\phi\left(K_{\theta^{\mathrm{init}}}^m(X_i)\right) \mid X_i, \theta^{\mathrm{init}}\right] - \mathbb{E}\left[\phi\left(K_{\theta^{\mathrm{init}}}^m(X_i')\right) \mid \theta^{\mathrm{init}}\right]\right\|}{n} > \epsilon\right),$$

and    $\varepsilon_{n,m;\nu}^{\mathrm{init}}(\epsilon) := \sqrt{\epsilon^2 + \kappa_{\nu;m}^2\left(\vartheta_{n,m}^{\mathrm{init}}(\epsilon)\right)^{\frac{2}{\nu-2}}}.$

We also consider the i.i.d. samples, drawn independently of $X_1, \ldots, X_n$ and on a given $\psi \in \Psi$, as

$$X_1^\psi, \ldots, X_n^\psi \overset{\mathrm{i.i.d.}}{\sim} p_\psi.$$

For notational clarity, we shall use $\theta_{m,B}$ to denote parameters arising from an one-step update, where the subscripts $m, B$ represent performing the one-step update with length-$m$ Markov chains and with batch size $B$. This is to be distinguished from $\psi_t$ elsewhere in the text, which denotes the parameter from the actual multi-step CD algorithm and the subscript $t$ denotes the $t$-th CD iterate.

**Gradient update schemes.**    We consider five different updates. Let $X_1'$ be an i.i.d. copy of $X_1$ drawn independently of all other random variables. The SGD-with-replacement update is given by

$$\theta_{m,B}^{\mathrm{SGDw}} := F_{m,B}^{\mathrm{SGDw}}(\theta^{\mathrm{init}}), \quad \text{where } F_{m,B}^{\mathrm{SGDw}}(\psi) := \psi - \frac{\eta}{B}\sum_{i \in S^w}\left(\phi(X_i) - \phi\left(K_{i;\psi}^m(X_i)\right)\right)$$

and $S^w$ is a uniformly drawn size-$B$ subset of $[n]$. The SGD-without-replacement update, after renormalizing the learning rate, is given by the $N$-fold function composition

$$\theta_{m,B}^{\mathrm{SGDo}} \;:=\; F_{m,B;N}^{\mathrm{SGDo}} \circ \ldots \circ F_{m,B;1}^{\mathrm{SGDo}}(\theta^{\mathrm{init}})\,,$$
$$\text{where} \quad F_{m,B;j}^{\mathrm{SGDo}}(\psi) \;:=\; \psi - \frac{\eta}{NB}\sum_{i \in S_j^o}\Big(\phi(X_i) - \phi\big(K_{i;\psi}^m(X_i)\big)\Big) \quad \text{for each } j \in [N]\,,$$

and $S_j^o$'s are disjoint size-$B$ random subsets of $[n]$, defined by $(S_1^o,\ldots,S_N^o) = \pi([n])$ for a uniformly drawn element $\pi$ of the permutation group on $n$ objects. The full-batch gradient update is given by

$$\theta_m^{\mathrm{GD}} \;:=\; F_m^{\mathrm{GD}}(\theta^{\mathrm{init}})\,, \quad \text{where} \quad F_m^{\mathrm{GD}}(\psi) \;:=\; \psi - \frac{\eta}{n}\sum_{i \le n}\Big(\phi(X_i) - \phi\big(K_{i;\psi}^m(X_i)\big)\Big)\,.$$

The full-batch gradient update with an infinite-length Markov chain is given by

$$\theta_\infty^{\mathrm{GD}} \;:=\; F_\infty^{\mathrm{GD}}(\theta^{\mathrm{init}})\,, \quad \text{where} \quad F_\infty^{\mathrm{GD}}(\psi) \;:=\; \psi - \frac{\eta}{n}\sum_{i \le n}\Big(\phi(X_i) - \phi(X_1^\psi)\Big)\,.$$

The population gradient update with an infinite-length Markov chain is given by

$$\theta^{\mathrm{pop}} \;:=\; f^{\mathrm{pop}}(\theta^{\mathrm{init}})\,, \quad \text{where} \quad f^{\mathrm{pop}}(\psi) \;:=\; \psi - \eta\,\mathbb{E}\big[\phi(X_1) - \phi(X_1^\psi)\big]\,,$$

where we use the lowercase $f$ to emphasize that $f^{\mathrm{pop}}$ is a deterministic function.

The forthcoming results are summarized below:

$$\theta_{m,B}^{\mathrm{SGDw}} \stackrel{\text{Lemma E.1}}{\approx} \theta_m^{\mathrm{GD}} \stackrel{\text{Lemma E.2}}{\approx} \theta_\infty^{\mathrm{GD}} \stackrel{\text{Lemma E.3}}{\approx} \theta^{\mathrm{pop}} \stackrel{\text{Lemma E.4}}{\approx} \psi^*\,.$$

**Lemma E.1.** *Let $\mathcal{F}_n$ be the sigma algebra generated by $\{X_i, K_{i;\theta^{\mathrm{init}}}^m(X_i) \mid 1 \le i \le n\}$. Then*

$$\mathbb{E}\big[\theta_{m,B}^{\mathrm{SGDw}} - \theta_m^{\mathrm{GD}} \,\big|\, \theta^{\mathrm{init}}, \mathcal{F}_n\big] \;=\; 0 \qquad \text{almost surely}\,.$$

*Moreover, under A1 and A7, we have*

$$\mathbb{E}\big\|\theta_{m,B}^{\mathrm{SGDw}} - \theta_m^{\mathrm{GD}}\big\|^2 \;\le\; \frac{4\eta^2(\sigma^2 + \kappa_{\nu;m}^2)}{B}\,\mathbb{I}_{\{B < n\}}\,.$$

*Proof of Lemma E.1.* Write $A := (A_1,\ldots,A_n)$, where

$$A_i \;:=\; \big(\phi(X_i) - \phi\big(K_{i;\theta^{\mathrm{init}}}^m(X_i)\big)\big) - \mathbb{E}\big[\phi(X_1) - \phi\big(K_{i;\theta^{\mathrm{init}}}^m(X_1)\big)\big]\,.$$

Since $S^w$ is uniformly drawn from all size-$B$ subsets of $[n]$ and independently of all other variables, we have that almost surely

$$\mathbb{E}\big[\theta_{m,B}^{\mathrm{SGDw}} - \theta_m^{\mathrm{GD}} \,\big|\, \theta^{\mathrm{init}}, \mathcal{F}_n\big] \;=\; \mathbb{E}\Big[\frac{\eta}{B}\sum_{i \in S^w} A_i - \frac{\eta}{n}\sum_{i \le n} A_i \,\Big|\, A\Big] \;=\; 0\,.$$

To prove the remaining bound, we note that the above relation implies $\theta_{m,B}^{\mathrm{SGDw}} - \theta_m^{\mathrm{GD}}$ is zero-mean. By the law of total variance, we have

$$\mathbb{E}\big\|\theta_{m,B}^{\mathrm{SGDw}} - \theta_m^{\mathrm{GD}}\big\|^2 \;=\; \mathrm{Tr}\,\mathrm{Cov}\big[\theta_{m,B}^{\mathrm{SGDw}} - \theta_m^{\mathrm{GD}}\big] \;=\; \mathrm{Tr}\,\mathbb{E}\,\mathrm{Cov}\big[\theta_{m,B}^{\mathrm{SGDw}} - \theta_m^{\mathrm{GD}} \,\big|\, \theta^{\mathrm{init}}, \mathcal{F}_n\big]$$
$$=\; \eta^2\,\mathrm{Tr}\,\mathbb{E}\bigg[\mathbb{E}\Big[\Big(\frac{1}{B}\sum_{i \in S^w} A_i\Big)\Big(\frac{1}{B}\sum_{i \in S^w} A_i\Big)^\top \,\Big|\, A\Big] - \Big(\frac{1}{n}\sum_{i \le n} A_i\Big)\Big(\frac{1}{n}\sum_{i \le n} A_i\Big)^\top\bigg]\,.$$

To compute the covariance, recall that $S^w$ is a uniformly drawn size-$B$ subset of $[n]$ with $B = n/N$. Let $\mathcal{P}_N([n])$ be the collection of all partitions of $[n]$ into $N$ size-$B$ subsets. We can generate $S^w$ by the following two-step process:

(i) Uniformly draw a partition $P' = (P_1',\ldots,P_N')$ from $\mathcal{P}_N([n])$;
(ii) Uniformly sample an index $K$ from $[N]$ and set $S^w = P_K'$.

Then we have, almost surely

$$
\mathbb{E}\Big[\Big(\tfrac{1}{B}\sum_{i\in S^w}A_i\Big)\Big(\tfrac{1}{B}\sum_{i\in S^w}A_i\Big)^\top\,\Big|\,A\Big] - \Big(\tfrac{1}{n}\sum_{i\le n}A_i\Big)\Big(\tfrac{1}{n}\sum_{i\le n}A_i\Big)^\top
$$
$$
= \frac{1}{|\mathcal{P}_N([n])|}\sum_{P'\in\mathcal{P}_N([n])}\Big(\frac{1}{N}\sum_{k\le N}\frac{1}{B^2}\sum_{i,j\in P'_k}A_iA_j^\top - \frac{1}{n^2}\sum_{i,j\le n}A_iA_j^\top\Big)
$$
$$
= \frac{1}{|\mathcal{P}_N([n])|}\sum_{P'\in\mathcal{P}_N([n])}\Big(\frac{1}{NB^2}\sum_{k\le N}\sum_{i,j\in P'_k}A_iA_j^\top - \frac{1}{N^2B^2}\sum_{k,l\le N}\sum_{i\in P'_k,j\in P'_l}A_iA_j^\top\Big)
$$
$$
= \frac{1}{|\mathcal{P}_N([n])|}\sum_{P'\in\mathcal{P}_N([n])}\Big(\frac{N-1}{N^2B^2}\sum_{k\le N}\sum_{i,j\in P'_k}A_iA_j^\top - \frac{1}{N^2B^2}\sum_{k\ne l}\sum_{i\in P'_k,j\in P'_l}A_iA_j^\top\Big).
$$

By noting that $A_i$'s are exchangeable, we obtain

$$
\mathbb{E}\big\|\theta^{\mathrm{SGDw}}_{m,B}-\theta^{\mathrm{GD}}_m\big\|^2 = \eta^2\mathrm{Tr}\,\mathbb{E}\Big[\frac{N-1}{NB}A_1A_1^\top + \frac{(N-1)(B-1)}{NB}A_1A_2^\top - \frac{N-1}{N}A_1A_2^\top\Big]
$$
$$
= \eta^2\mathrm{Tr}\,\mathbb{E}\Big[\frac{N-1}{NB}A_1A_1^\top - \frac{N-1}{NB}A_1A_2^\top\Big]
$$
$$
= \frac{\eta^2(N-1)}{NB}\big(\mathbb{E}\|A_1\|^2 - \mathbb{E}\langle A_1,A_2\rangle\big)
$$
$$
\overset{(a)}{\le} \frac{2\eta^2}{B}\mathbb{E}\|A_1\|^2
$$
$$
= \frac{2\eta^2}{B}\mathbb{E}\Big\|\big(\phi(X_1)-\mathbb{E}[\phi(X_1)]\big) - \Big(\phi\big(K^m_{i;\theta^{\mathrm{init}}}(X_1)\big)-\mathbb{E}\Big[\phi\big(K^m_{i;\theta^{\mathrm{init}}}(X_1)\big)\Big]\Big)\Big\|^2
$$
$$
\le \frac{4\eta^2}{B}\Big(\mathrm{Tr}\,\mathrm{Cov}[\phi(X_1)] + \mathrm{Tr}\,\mathrm{Cov}\Big[\phi\big(K^m_{i;\theta^{\mathrm{init}}}(X_1)\big)\Big]\Big)
$$
$$
\overset{(b)}{\le} \frac{4\eta^2(\sigma^2+\kappa^2_{\nu;m})}{B}.
$$

In $(a)$, we have used a Cauchy-Schwarz inequality; in $(b)$, we have used A1 and A7. Finally we note that if $B=n$, $\theta^{\mathrm{SGDw}}_{m,B}=\theta^{\mathrm{GD}}_m$ almost surely, which implies the desired bound. $\qquad\square$

**Lemma E.2.** *Denote $\mathcal{A}_n$ as the sigma algebra generated by $\theta^{\mathrm{init}},X_1,\ldots,X_n$. Under A1, A2, A3 and A7, we have that for any $\epsilon>0$ and $\nu>2$,*

$$
\mathbb{E}\big\|\mathbb{E}\big[\theta^{\mathrm{GD}}_m-\theta^{\mathrm{GD}}_\infty\,\big|\,\mathcal{A}_n\big]\big\|^2 \le \eta^2\Big(\alpha^m\sigma C_\chi\sqrt{\mathbb{E}\|\theta^{\mathrm{init}}-\psi^*\|^2} + \varepsilon^{\mathrm{init}}_{n,m;\nu}(\epsilon)\Big)^2,
$$
$$
\mathbb{E}\big\|\theta^{\mathrm{GD}}_m-\theta^{\mathrm{GD}}_\infty\big\|^2 \le \eta^2\Big(\big(\alpha^m\sigma C_\chi\sqrt{\mathbb{E}\|\theta^{\mathrm{init}}-\psi^*\|^2} + \varepsilon^{\mathrm{init}}_{n,m;\nu}(\epsilon)\big)^2 + \frac{\kappa^2_{\nu;m}+\sigma^2}{n}\Big).
$$

*Proof of Lemma E.2.* The main challenge arises from the possible correlation between $\theta^{\mathrm{init}}$ and $X_1,\ldots,X_n$. First note that for any $\epsilon>0$, $\nu>2$ and a real-valued random variable $Y$, by Hölder's inequality, we have

$$
\mathbb{E}[Y^2] = \mathbb{E}\big[Y^2\,\mathbb{I}_{\{Y\le\epsilon\}} + Y^2\,\mathbb{I}_{\{Y>\epsilon\}}\big]
$$
$$
\le \epsilon^2 + \mathbb{E}[Y^2\mathbb{I}_{\{Y>\epsilon\}}] \le \epsilon^2 + (\mathbb{E}[Y^\nu])^{2/\nu}\mathbb{P}(Y>\epsilon)^{(\nu-2)/\nu}. \tag{15}
$$

Also note the useful inequality that for two real-valued random vectors (possibly correlated) $V_1,V_2$, we have

$$
\mathbb{E}[\|V_1+V_2\|^2] \le \mathbb{E}[(\|V_1\|+\|V_2\|)^2] \le \mathbb{E}\|V_1\|^2 + 2\sqrt{(\mathbb{E}\|V_1\|^2)(\mathbb{E}\|V_2\|^2)} + \mathbb{E}\|V_2\|^2
$$
$$
= \big(\sqrt{\mathbb{E}\|V_1\|^2} + \sqrt{\mathbb{E}\|V_2\|^2}\big)^2. \tag{16}
$$

Now to control the first quantity of interest, by using a triangle inequality, we have

$$
\mathbb{E}\big\|\mathbb{E}\big[\theta^{\mathrm{GD}}_m-\theta^{\mathrm{GD}}_\infty\,\big|\,\mathcal{A}_n\big]\big\|^2 = \mathbb{E}\Big\|\mathbb{E}\Big[\frac{\eta}{n}\sum_{i\le n}\big(\phi\big(K^m_{i;\theta^{\mathrm{init}}}(X_i)\big)-\phi\big(X_i^{\theta^{\mathrm{init}}}\big)\big)\,\Big|\,\mathcal{A}_n\Big]\Big\|^2
$$
$$
\overset{(16)}{\le} \eta^2\Big(\sqrt{\mathbb{E}[\Delta_1^2]} + \sqrt{\mathbb{E}[\Delta_2^2]}\Big)^2,
$$

where

$$\Delta_1 := \left\| \mathbb{E}\Big[\frac{1}{n}\sum_{i\leq n}\big(\phi\big(K_{i;\theta^{\mathrm{init}}}^m(X_i)\big) - \mathbb{E}\big[\phi\big(K_{i;\theta^{\mathrm{init}}}^m(X_1')\big)\,\big|\,\theta^{\mathrm{init}}\big]\big)\,\Big|\,\mathcal{A}_n\Big]\right\|$$

$$= \left\|\frac{1}{n}\sum_{i\leq n}\big(\mathbb{E}\big[\phi\big(K_{i;\theta^{\mathrm{init}}}^m(X_i)\big)\,\big|\,\theta^{\mathrm{init}},X_i\big] - \mathbb{E}\big[\phi\big(K_{i;\theta^{\mathrm{init}}}^m(X_1')\big)\,\big|\,\theta^{\mathrm{init}}\big]\big)\right\|,$$

$$\Delta_2 := \left\|\mathbb{E}\Big[\frac{1}{n}\sum_{i\leq n}\big(\mathbb{E}\big[\phi\big(K_{i;\theta^{\mathrm{init}}}^m(X_1')\big)\,\big|\,\theta^{\mathrm{init}}\big] - \phi\big(X_i^{\theta^{\mathrm{init}}}\big)\big)\,\Big|\,\mathcal{A}_n\Big]\right\|$$

$$= \left\|\mathbb{E}\big[\phi\big(K_{1;\theta^{\mathrm{init}}}^m(X_1')\big) - \phi\big(X_1^{\theta^{\mathrm{init}}}\big)\,\big|\,\theta^{\mathrm{init}}\big]\right\|,$$

and $X_1'$ is an i.i.d. copy of $X_1$ and in particular independent of $\theta^{\mathrm{init}}$. $\Delta_1$ is controlled via (15):

$$\mathbb{E}[\Delta_1^2] \leq \epsilon^2 + (\mathbb{E}[\Delta_1^\nu])^{2/\nu}\,\mathbb{P}(\Delta_1 > \epsilon)^{\nu/(\nu-2)}$$

$$\overset{(a)}{\leq} \epsilon^2 + \left(\mathbb{E}\Big\|\phi(K_{1;\theta^{\mathrm{init}}}^m(X_1)) - \mathbb{E}\big[\phi(K_{1;\theta^{\mathrm{init}}}^m(X_1'))\,\big|\,\theta^{\mathrm{init}}\big]\Big\|^\nu\right)^{2/\nu}$$

$$\times \mathbb{P}\left(\frac{\big\|\sum_{i\leq n}\big(\mathbb{E}[\phi(K_{i;\theta^{\mathrm{init}}}^m(X_i))\,|\,\theta^{\mathrm{init}},X_i] - \mathbb{E}[\phi(K_{i;\theta^{\mathrm{init}}}^m(X_1'))\,|\,\theta^{\mathrm{init}}]\big)\big\|}{n} > \epsilon\right)^{\frac{\nu-2}{\nu}}$$

$$\overset{(b)}{\leq} \epsilon^2 + \kappa_{\nu;m}^2\big(\vartheta_{n,m}^{\mathrm{init}}(\epsilon)\big)^{\frac{\nu-2}{\nu}}$$

$$= \big(\varepsilon_{n,m;\nu}^{\mathrm{init}}(\epsilon)\big)^2. \tag{17}$$

In $(a)$, we have plugged in the definition of $\Delta_1$ and applied a Jensen's inequality with respect to the empirical average; in $(b)$, we have used A7 to bound the $\nu$-th moment term as

$$\left(\mathbb{E}\Big\|\phi(K_{1;\theta^{\mathrm{init}}}^m(X_1)) - \mathbb{E}\big[\phi(K_{1;\theta^{\mathrm{init}}}^m(X_1'))\,\big|\,\theta^{\mathrm{init}}\big]\Big\|^\nu\right)^{1/\nu}$$

$$= \left(\mathbb{E}\Big[\mathbb{E}\big[\big\|\phi(K_{1;\theta^{\mathrm{init}}}^m(X_1)) - \mathbb{E}\big[\phi(K_{1;\theta^{\mathrm{init}}}^m(X_1'))\,\big|\,\theta^{\mathrm{init}}\big]\big\|^\nu\,\big|\,\theta^{\mathrm{init}}\big]\Big]\right)^{1/\nu}$$

$$\leq \sup_{\psi\in\Psi}\left(\mathbb{E}\big[\big\|\phi(K_{1;\psi}^m(X_1)) - \mathbb{E}\big[\phi(K_{1;\psi}^m(X_1'))\big]\big\|^\nu\big]\right)^{1/\nu}$$

$$= \sup_{\psi\in\Psi}\left(\mathbb{E}\big[\big\|\phi(K_{1;\psi}^m(X_1)) - \mathbb{E}\big[\phi(K_{1;\psi}^m(X_1))\big]\big\|^\nu\big]\right)^{1/\nu} \leq \kappa_{\nu;m}$$

and recalled the definitions of $\vartheta_{n,m}^{\mathrm{init}}$ and $\varepsilon_{n,m;\nu}^{\mathrm{init}}$. On the other hand,

$$\mathbb{E}[\Delta_2^2] = \mathbb{E}\left\|\int_{\mathbb{R}^d}\phi(x)(K_{1;\theta^{\mathrm{init}}}^m p_{\psi^*})(x)dx - \mathbb{E}\big[\phi(X_1^{\theta^{\mathrm{init}}})\big|\theta^{\mathrm{init}}\big]\right\|^2$$

$$\overset{(a)}{=} \mathbb{E}\left\|\int_{\mathbb{R}^d}(K_{1;\theta^{\mathrm{init}}}^m\phi)(x)\,p_{\psi^*}(x)dx - \mathbb{E}_{p_{\theta^{\mathrm{init}}}}[\phi]\right\|^2$$

$$\overset{(b)}{=} \mathbb{E}\left\|\int_{\mathbb{R}^d}\big(K_{1;\theta^{\mathrm{init}}}^m\big(\phi - \mathbb{E}_{p_{\theta^{\mathrm{init}}}}[\phi]\big)\big)(x)\times p_{\psi^*}(x)dx\right\|^2$$

$$\overset{(c)}{=} \mathbb{E}\left\|\int_{\mathbb{R}^d}\big(K_{1;\theta^{\mathrm{init}}}^m\big(\phi - \mathbb{E}_{p_{\theta^{\mathrm{init}}}}[\phi]\big)\big)(x)\times (p_{\psi^*}(x) - p_{\theta^{\mathrm{init}}}(x))dx\right\|^2$$

$$\overset{(d)}{=} \sum_{l=1}^d \mathbb{E}\left(\int_{\mathbb{R}^d}\big(K_{t1;\theta^{\mathrm{init}}}^m\big(\phi - \mathbb{E}_{p_{\theta^{\mathrm{init}}}}[\phi]\big)\big)(x)^\top e_l\times p_{\theta^{\mathrm{init}}}(x)\times\frac{p_{\psi^*}(x) - p_{\theta^{\mathrm{init}}}(x)}{p_{\theta^{\mathrm{init}}}(x)}\,dx\right)^2$$

$$\overset{(e)}{\leq} \sum_{l=1}^d \mathbb{E}\left[\left(\int_{\mathbb{R}^d}\big(\big(K_{t1;\theta^{\mathrm{init}}}^m\big(\phi - \mathbb{E}_{p_{\theta^{\mathrm{init}}}}[\phi]\big)\big)(x)^\top e_l\big)^2 p_{\theta^{\mathrm{init}}}(x)dx\right.\right.$$

$$\left.\left.\times\int_{\mathbb{R}^d}\Big(\frac{p_{\psi^*}(x) - p_{\theta^{\mathrm{init}}}(x)}{p_{\theta^{\mathrm{init}}}(x)}\Big)^2 p_{\theta^{\mathrm{init}}}(x)dx\right)^2\right]$$

$$\overset{(f)}{\leq} \sum_{l=1}^d \mathbb{E}\left(\alpha^{2m}\,\mathrm{Tr}\,\mathrm{Cov}\big[\phi(X_1^{\theta^{\mathrm{init}}})\,\big|\,\theta^{\mathrm{init}}\big]\,\chi^2(p_{\psi^*},p_{\theta^{\mathrm{init}}})\right)^2$$

$$\overset{(g)}{\leq} \alpha^{2m}\sigma^2 C_\chi^2\,\mathbb{E}\|\theta^{\mathrm{init}} - \psi^*\|^2.$$

In $(a)$, we have used that $(Kf)(x) = \int K(x,y)f(y)dy$; in $(b)$, we have used that the Markov operator leaves the constant function invariant; in $(c)$, we used that $K_{1;\theta^{\mathrm{init}}}$ leaves $p_{\theta^{\mathrm{init}}}$ invariant; in

$(d)$, we denoted $(e_l)_{l \leq d}$ as the standard basis vectors of $\mathbb{R}^d$ and multiplied and divided by $p_{\theta^{\text{init}}}(x)$; in $(e)$, we have used a Cauchy-Schwarz inequality; in $(f)$, we have used the definition of the spectral gap $\alpha$ in A3; in $(g)$, we have used A1 and A2. Combining the bounds gives the first inequality that

$$\mathbb{E}\left\|\mathbb{E}\left[\theta_m^{\text{GD}} - \theta_\infty^{\text{GD}} \mid \mathcal{A}_n\right]\right\|^2 \leq \eta^2\left(\alpha^m \sigma C_\chi \sqrt{\mathbb{E}\|\theta^{\text{init}} - \psi^*\|^2} + \varepsilon_{n,m;\nu}^{\text{init}}(\epsilon)\right)^2.$$

Now we handle the second quantity by conditioning on $\theta^{\text{init}}$ and perform a bias-variance decomposition:

$$
\begin{aligned}
\mathbb{E}\|\theta_m^{\text{GD}} - \theta_\infty^{\text{GD}}\|^2 &= \eta^2\, \mathbb{E}\left[\mathbb{E}\left[\left\|\frac{1}{n}\sum_{i \leq n}\left(\phi\left(K_{i;\theta^{\text{init}}}^m(X_i)\right) - \phi(X_i^{\theta^{\text{init}}})\right)\right\|^2 \,\Big|\, \mathcal{A}_n\right]\right] \\
&= \eta^2(Q_B + Q_V),
\end{aligned}
$$

where

$$
\begin{aligned}
Q_B &:= \mathbb{E}\left\|\mathbb{E}\left[\frac{1}{n}\sum_{i \leq n}\left(\phi\left(K_{i;\theta^{\text{init}}}^m(X_i)\right) - \phi(X_i^{\theta^{\text{init}}})\right)\,\Big|\, \mathcal{A}_n\right]\right\|^2, \\
Q_V &:= \mathbb{E}\left[\text{Tr}\left(\text{Cov}\left[\frac{1}{n}\sum_{i \leq n}\phi\left(K_{i;\theta^{\text{init}}}^m(X_i)\right)\Big|\mathcal{A}_n\right] + \text{Cov}\left[\frac{1}{n}\sum_{i \leq n}\phi(X_i^{\theta^{\text{init}}})\Big|\theta^{\text{init}}\right]\right)\right].
\end{aligned}
$$

Note that the covariance terms separate because $X_i^{\theta^{\text{init}}}$ is independent of $K_{i;\theta^{\text{init}}}^m(X_i)$ conditioning on $\theta^{\text{init}}$. $\eta^2 Q_B$ is exactly the quantity controlled above, so it suffices to bound the variance term $Q_V$. By explicitly computing the second covariance term while noting that $X_1^{\theta^{\text{init}}}, \ldots, X_n^{\theta^{\text{init}}}$ are conditionally i.i.d. given $\theta^{\text{init}}$ and $K_{i;\theta^{\text{init}}}^m(X_i)$'s are conditionally independent across $1 \leq i \leq n$ given $\mathcal{A}_n$, we have

$$
\begin{aligned}
Q_V &= \frac{\sum_{i \leq n}\mathbb{E}[\text{Tr}\,\text{Cov}[\phi(K_{i;\theta^{\text{init}}}^m(X_i)) \mid \theta^{\text{init}}, X_i]]}{n^2} + \frac{\mathbb{E}[\text{Tr}\,\text{Cov}[\phi(X_1^{\theta^{\text{init}}})|\theta^{\text{init}}]]}{n} \\
&\overset{(a)}{=} \frac{\mathbb{E}[\text{Tr}\,\text{Cov}[\phi(K_{1;\theta^{\text{init}}}^m(X_1)) \mid \theta^{\text{init}}, X_1]]}{n} + \frac{\mathbb{E}[\text{Tr}\,\text{Cov}[\phi(X_1^{\theta^{\text{init}}})|\theta^{\text{init}}]]}{n} \\
&\leq \frac{\kappa_{\nu;m}^2 + \sigma^2}{n},
\end{aligned}
$$

where we have used A4, A7 and A1 in the last line. Combining the bounds, we obtain that

$$
\begin{aligned}
\mathbb{E}\|\theta_m^{\text{GD}} - \theta_\infty^{\text{GD}}\|^2 &= \eta^2(Q_B + Q_V) \\
&\leq \eta^2\left(\left(\alpha^m \sigma C_\chi \sqrt{\mathbb{E}\|\theta^{\text{init}} - \psi^*\|^2} + \varepsilon_{n,m;\nu}^{\text{init}}(\epsilon)\right)^2 + \frac{\kappa_{\nu;m}^2 + \sigma^2}{n}\right).
\end{aligned}
$$

$\square$

**Lemma E.3.** *Under A1, $\mathbb{E}\|\theta_\infty^{\text{GD}} - \theta^{\text{pop}}\|^2 \leq \frac{4\eta^2\sigma^2}{n}$.*

*Proof of Lemma E.3.* Since both $F_\infty^{\text{GD}}$ and $f^{\text{pop}}$ involve infinite-length Markov chains, the initializations do not matter and we can decouple the stochasticity of $X_i$ and $K_{i;\theta^{\text{init}}}$. In particular,

$$
\begin{aligned}
\mathbb{E}\|\theta_\infty^{\text{GD}} - \theta^{\text{pop}}\|^2 &= \eta^2\, \mathbb{E}\left\|\frac{1}{n}\sum_{i \leq n}\left(\phi(X_i) - \phi(X_i^{\theta^{\text{init}}})\right) - \mathbb{E}\left[\phi(X_1) - \phi(X_1^{\theta^{\text{init}}})\right]\right\|^2 \\
&\leq \eta^2\left(Q_1' + 2\sqrt{Q_1' Q_2'} + Q_2'\right),
\end{aligned}
$$

where

$$
\begin{aligned}
Q_1' &:= \mathbb{E}\left\|\frac{1}{n}\sum_{i \leq n}\left(\phi(X_i) - \mathbb{E}[\phi(X_1)]\right)\right\|^2 = \frac{\text{Tr}\,\text{Cov}[\phi(X_1)]}{n} = \frac{\text{Tr}\,\nabla_\theta^2 \log Z(\psi^*)}{n} \leq \frac{\sigma^2}{n}, \\
Q_2' &:= \mathbb{E}\left\|\frac{1}{n}\sum_{i \leq n}\left(\phi(X_i^{\theta^{\text{init}}}) - \mathbb{E}[\phi(X_1^{\theta^{\text{init}}})]\right)\right\|^2 \\
&= \frac{\mathbb{E}[\text{Tr}\,\text{Cov}[\phi(X_1^{\theta^{\text{init}}})|\theta^{\text{init}}]]}{n} = \frac{\mathbb{E}[\text{Tr}\,\nabla_\theta^2 \log Z(\theta^{\text{init}})]}{n} \leq \frac{\sigma^2}{n}.
\end{aligned}
$$

In the computations above, we have used the relation $\nabla_\theta^2 \log Z(\theta) = \text{Cov}_{X \sim p_\theta}[\phi(X)]$ and the assumption $\sup_{\theta \in \Psi} \text{tr}(\nabla_\theta^2 \log Z(\theta)) = \sigma^2$ from A1. This implies the desired bound. $\square$

**Lemma E.4.** *Under A1,* $\mathbb{E}\|\theta^{\mathrm{pop}} - \psi^*\|^2 \leq \left(1 - 2\mu\eta + L^2\eta^2\right)\mathbb{E}\|\theta^{\mathrm{init}} - \psi^*\|^2$ .

*Proof of Lemma E.4.* Recall that

$$f^{\mathrm{pop}}(\theta') \;=\; \theta - \eta\,\mathbb{E}\big[\phi(X_1) - \phi(X_1^{\theta'})\big] \;=\; \theta - \eta\left(\nabla_\psi \log Z(\psi^*) - \nabla_\psi \log Z(\theta')\right) .$$

By construction, $f^{\mathrm{pop}}$ is deterministic and $f^{\mathrm{pop}}(\psi^*) = \psi^*$. By plugging in the recursions and expanding the square, we get that

$$
\begin{aligned}
\mathbb{E}\left\|\theta^{\mathrm{pop}} - \psi^*\right\|^2 \;&=\; \mathbb{E}\left\|f^{\mathrm{pop}}(\theta^{\mathrm{init}}) - f^{\mathrm{pop}}(\psi^*)\right\|^2 \\
&= \mathbb{E}\left\|(\theta^{\mathrm{init}} - \psi^*) - \eta(\nabla_\psi \log Z(\theta^{\mathrm{init}}) - \nabla_\psi \log Z(\psi^*))\right\|^2 \\
&= \mathbb{E}\left\|\theta^{\mathrm{init}} - \psi^*\right\|^2 - 2\eta\,\mathbb{E}\big[\langle \theta^{\mathrm{init}} - \psi^*\,,\,\nabla_\psi \log Z(\theta^{\mathrm{init}}) - \nabla_\psi \log Z(\psi^*)\rangle\big] \\
&\quad + \eta^2\,\mathbb{E}\left\|\nabla_\psi \log Z(\theta^{\mathrm{init}}) - \nabla_\psi \log Z(\psi^*)\right\|^2 \\
&\leq \mathbb{E}\left\|\theta^{\mathrm{init}} - \psi^*\right\|^2 - 2\mu\eta\,\mathbb{E}\left\|\theta^{\mathrm{init}} - \psi^*\right\|^2 + L^2\eta^2\,\mathbb{E}\left\|\theta^{\mathrm{init}} - \psi^*\right\|^2 .
\end{aligned}
$$

In the last line, we have recalled $\inf_{\psi\in\Psi}\lambda_{\min}(\nabla^2_\psi \log Z(\psi)) = \mu$ and $\sup_{\theta\in\Psi}\lambda_{\max}(\nabla^2_\psi \log Z(\psi)) = L$ by A1 and applied Lemma C.8. Combining the coefficients gives the desired statement. $\qquad\square$

## F   Proofs for offline SGD

We prove Theorem B.1 (which directly implies Theorem 4.3) and Theorem B.2 in this section. The key ingredient of both proofs is Lemma F.1 below, which provides an iterative error bound for the SGD-with-replacement scheme by combining different approximation bounds in Appendix E. Throughout this section, we denote $\delta^{\mathrm{SGDw}}_{t,j} := \mathbb{E}\left\|\psi^{\mathrm{SGDw}}_{t,j} - \psi^*\right\|^2$.

**Lemma F.1.** *Under A1, A2, A3, A4 and A7, we have that for* $1 \leq j \leq N-1$,

$$\sqrt{\delta^{\mathrm{SGDw}}_{t,j}} \;\leq\; \left(1 - \eta_t\left(\mu - \alpha^m \sigma C_\chi - \frac{L^2}{2}\eta_t\right)\right)\sqrt{\delta^{\mathrm{SGDw}}_{t,j-1}} \;+\; \eta_t\left(\varepsilon^{\mathrm{SGDw}}_{n,m,t;\nu}(\epsilon) + \frac{5\sigma + 5\kappa_{\nu;m}}{\sqrt{B}}\right) ,$$

*where* $1 - \eta_t\left(\mu - \alpha^m \sigma C_\chi - \frac{L^2}{2}\eta_t\right) > 0$.

*Proof of Lemma F.1.* We first remark that in view of Lemma C.4, the projection step in Algorithm 2 does not increase $\delta^{\mathrm{SGDw}}_{t,j}$, so it suffices to bound $\delta^{\mathrm{SGDw}}_{t,j}$ as if projection is not performed on $\psi^{\mathrm{SGDw}}_{t,j}$. To apply the results from Appendix E, we identify $\theta^{\mathrm{init}} = \psi^{\mathrm{SGDw}}_{t,j-1}$ and $\eta = \eta_t$, which allows us to write $\psi^{\mathrm{SGDw}}_{t,j} = \theta^{\mathrm{SGDw}}_{m,B}$. This also implies $\mathbb{E}\|\theta^{\mathrm{init}} - \psi^*\|^2 = \delta^{\mathrm{SGDw}}_{t,j-1}$ and $\varepsilon^{\mathrm{init}}_{n,m;\nu}(\epsilon) \leq \varepsilon^{\mathrm{SGDw}}_{\nu;n,m,t}(\epsilon)$.

By adding and subtracting the auxiliary gradient updates followed by expanding the square, we obtain

$$
\begin{aligned}
\delta_{t,j}^{\mathrm{SGDw}} &= \mathbb{E}\left\|\theta_{m,B}^{\mathrm{SGDw}} - \theta_m^{\mathrm{GD}} + \theta_m^{\mathrm{GD}} - \theta_\infty^{\mathrm{GD}} + \theta_\infty^{\mathrm{GD}} - \theta^{\mathrm{pop}} + \theta^{\mathrm{pop}} - \psi^*\right\|^2 \\
&\stackrel{(a)}{=} \mathbb{E}\left\|\theta_{m,B}^{\mathrm{SGDw}} - \theta_m^{\mathrm{GD}}\right\|^2 + \mathbb{E}\left\|\theta_m^{\mathrm{GD}} - \theta_\infty^{\mathrm{GD}}\right\|^2 + \mathbb{E}\left\|\theta_\infty^{\mathrm{GD}} - \theta^{\mathrm{pop}}\right\|^2 + \mathbb{E}\left\|\theta^{\mathrm{pop}} - \psi^*\right\|^2 \\
&\quad + 2\,\mathbb{E}\left\langle\theta_{m,B}^{\mathrm{SGDw}} - \theta_m^{\mathrm{GD}},\ \theta_m^{\mathrm{GD}} - \theta_\infty^{\mathrm{GD}}\right\rangle + 2\,\mathbb{E}\left\langle\theta_{m,B}^{\mathrm{SGDw}} - \theta_m^{\mathrm{GD}},\ \theta_\infty^{\mathrm{GD}} - \theta^{\mathrm{pop}}\right\rangle \\
&\quad + 2\,\mathbb{E}\left\langle\mathbb{E}\left[\theta_{m,B}^{\mathrm{SGDw}} - \theta_m^{\mathrm{GD}}\,\big|\,\theta^{\mathrm{init}}, \mathcal{F}_n\right],\ \theta^{\mathrm{pop}} - \psi^*\right\rangle + 2\,\mathbb{E}\left\langle\theta_m^{\mathrm{GD}} - \theta_\infty^{\mathrm{GD}},\ \theta_\infty^{\mathrm{GD}} - \theta^{\mathrm{pop}}\right\rangle \\
&\quad + 2\,\mathbb{E}\left\langle\mathbb{E}\left[\theta_m^{\mathrm{GD}} - \theta_\infty^{\mathrm{GD}}\,\big|\,\mathcal{A}_n\right],\ \theta^{\mathrm{pop}} - \psi^*\right\rangle + 2\,\mathbb{E}\left\langle\theta_\infty^{\mathrm{GD}} - \theta^{\mathrm{pop}},\ \theta^{\mathrm{pop}} - \psi^*\right\rangle \\
&\stackrel{(b)}{\leq} \frac{4\eta_t^2(\sigma^2 + \kappa_{\nu;m}^2)}{\sqrt{B}}\mathbb{I}_{\{B<n\}} + \eta_t^2\left(\left(\alpha^m\sigma C_\chi\sqrt{\delta_{t,j-1}^{\mathrm{SGDw}}} + \varepsilon_{n,m,t;\nu}^{\mathrm{SGDw}}(\epsilon)\right)^2 + \frac{\kappa_{\nu;m}^2 + \sigma^2}{n}\right) \\
&\quad + \frac{4\eta_t^2\sigma^2}{n} + (1 - 2\mu\eta_t + L^2\eta_t^2)\delta_{t,j-1}^{\mathrm{SGDw}} \\
&\quad + 2\frac{2\eta_t\sqrt{\sigma^2 + \kappa_{\nu;m}^2}}{\sqrt{B}}\mathbb{I}_{\{B<n\}}\eta_t\left(\alpha^m\sigma C_\chi\sqrt{\delta_{t,j-1}^{\mathrm{SGDw}}} + \varepsilon_{n,m,t;\nu}^{\mathrm{SGDw}}(\epsilon) + \frac{\sqrt{\kappa_{\nu;m}^2 + \sigma^2}}{\sqrt{n}}\right) \\
&\quad + 2\frac{2\eta_t\sqrt{\sigma^2 + \kappa_{\nu;m}^2}}{\sqrt{B}}\mathbb{I}_{\{B<n\}}\frac{2\eta_t\sigma}{\sqrt{n}} + 0 \\
&\quad + 2\eta_t\left(\alpha^m\sigma C_\chi\sqrt{\delta_{t,j-1}^{\mathrm{SGDw}}} + \varepsilon_{n,m,t;\nu}^{\mathrm{SGDw}}(\epsilon) + \frac{\sqrt{\kappa_{\nu;m}^2 + \sigma^2}}{\sqrt{n}}\right)\frac{2\eta_t\sigma}{\sqrt{n}} \\
&\quad + 2\eta_t\left(\alpha^m\sigma C_\chi\sqrt{\delta_{t,j-1}^{\mathrm{SGDw}}} + \varepsilon_{n,m,t;\nu}^{\mathrm{SGDw}}(\epsilon)\right)\sqrt{(1 - 2\mu\eta_t + L^2\eta_t^2)\delta_{t,j-1}^{\mathrm{SGDw}}} \\
&\quad + 2\frac{2\eta_t\sigma}{\sqrt{n}}\sqrt{(1 - 2\mu\eta_t + L^2\eta_t^2)\delta_{t,j-1}^{\mathrm{SGDw}}} \\
&\stackrel{(c)}{\leq} \left(1 - 2\mu\eta_t + L\eta_t^2 + \eta_t^2\alpha^{2m}\sigma^2 C_\chi^2 + 2\eta_t\alpha^m\sigma C_\chi\sqrt{1 - 2\mu\eta_t + L^2\eta_t^2}\right) \times \delta_{t,j-1}^{\mathrm{SGDw}} \\
&\quad + 2\Bigg(\eta_t^2\alpha^m\sigma C_\chi\varepsilon_{n,m,t;\nu}^{\mathrm{SGDw}}(\epsilon) + \frac{2\eta_t^2(\sqrt{\sigma^2 + \kappa_{\nu;m}^2} + \sigma)\,\alpha^m\sigma C_\chi}{\sqrt{B}} \\
&\qquad\quad + \eta_t\varepsilon_{n,m,t;\nu}^{\mathrm{SGDw}}(\epsilon)\sqrt{1 - 2\mu\eta_t + L_2\eta_t^2} + \frac{2\eta_t\sigma\sqrt{1 - 2\mu\eta_t + L^2\eta_t^2}}{\sqrt{n}}\Bigg) \times \sqrt{\delta_{t,j-1}^{\mathrm{SGDw}}} \\
&\quad + \eta_t^2\Bigg(\frac{4(\sigma^2 + \kappa_{\nu;m}^2)}{B} + \left(\varepsilon_{n,m,t;\nu}^{\mathrm{SGDw}}(\epsilon)\right)^2 + \frac{5\kappa_{\nu;m}^2 + 9\sigma^2}{B} + \frac{4\sqrt{\sigma^2 + \kappa_{\nu;m}^2}\,\varepsilon_{n,m,t;\nu}^{\mathrm{SGDw}}(\epsilon)}{\sqrt{B}} \\
&\qquad\quad + \frac{8\sigma\sqrt{\sigma^2 + \kappa_{\nu;m}^2}}{B} + \frac{4\sigma\varepsilon_{n,m,t;\nu}^{\mathrm{SGDw}}(\epsilon)}{\sqrt{B}}\Bigg) \\
&=: (A_1) \times \delta_{t,j-1}^{\mathrm{SGDw}} + (A_2) \times \sqrt{\delta_{t,j-1}^{\mathrm{SGDw}}} + (A_3)\,.
\end{aligned}
$$

In $(a)$, we have expanded the square, used $\mathcal{F}_n$ defined in Lemma E.1 and $\mathcal{A}_n$ defined in Lemma E.2, and noted that $\theta^{\mathrm{pop}} - \psi^*$ is almost surely constant given $\theta^{\mathrm{init}}$; in $(b)$, we have applied Lemmas E.1, E.2, E.3 and E.4 under A1, A2, A3, A4 and A7, and used $\sqrt{a+b} \leq \sqrt{a} + \sqrt{b}$ for $a, b \geq 0$; in $(c)$, we have grouped the terms by powers of $\delta_{t,j-1}^{\mathrm{SGDw}}$, bounded the indicator function from above by 1 and used $B \leq n$ to replace $n$ in the denominator by $B$. While the computation above is complicated, we remark that the key steps are taking the conditional expectations for the cross-terms in $(b)$, which gives us a tighter bound than directly applying a triangle inequality of the form $\mathbb{E}\|Y_1 + Y_2\|^2 \leq (\sqrt{\mathbb{E}\|Y_1\|^2} + \sqrt{\mathbb{E}\|Y_2\|^2})$. To further simplify the bounds, we seek to bound each

coefficient by a square, which yields

$$(A_1) = \left(\eta_t \alpha^m \sigma C_\chi + \sqrt{1 - 2\mu\eta_t + L\eta_t^2}\right)^2,$$

$$(A_2) \leq 2\left(\eta_t \alpha^m \sigma C_\chi + \sqrt{1 - 2\mu\eta_t + L\eta_t^2}\right) \times \eta_t\left(\varepsilon_{n,m,t;\nu}^{\text{SGDw}}(\epsilon) + \frac{4\sigma + 2\kappa_{\nu;m}}{\sqrt{B}}\right),$$

$$(A_3) \leq \eta_t^2\left(\left(\varepsilon_{n,m,t;\nu}^{\text{SGDw}}(\epsilon)\right)^2 + \frac{8\sigma^2 + 4\kappa_{\nu;m}^2}{\sqrt{B}}\varepsilon_{n,m,t;\nu}^{\text{SGDw}}(\epsilon) + \frac{21\sigma^2 + 17\kappa_{\nu;m}^2}{B}\right)$$

$$\leq \eta_t^2\left(\varepsilon_{n,m,t;\nu}^{\text{SGDw}}(\epsilon) + \frac{5\sigma + 5\kappa_{\nu;m}}{\sqrt{B}}\right)^2.$$

This implies

$$\delta_{t,j}^{\text{SGDw}} \leq (A_1) \times \delta_{t,j-1}^{\text{SGDw}} + (A_2) \times \sqrt{\delta_{t,j-1}^{\text{SGDw}}} + (A_3)$$

$$\leq \left(\left(\eta_t \alpha^m \sigma C_\chi + \sqrt{1 - 2\mu\eta_t + L\eta_t^2}\right)\sqrt{\delta_{t,j-1}^{\text{SGDw}}} + \eta_t\left(\varepsilon_{n,m,t;\nu}^{\text{SGDw}}(\epsilon) + \frac{5\sigma + 5\kappa_{\nu;m}}{\sqrt{B}}\right)\right)^2.$$

Now note that since $\mu \leq L$ by the definitions in A1, $2\mu\eta_t - L^2\eta_t^2 \leq 2L\eta_t - L^2\eta_t^2 \leq 1$. By using $\sqrt{1-x} \leq 1 - \frac{x}{2}$ for all $x \leq 1$, we get that

$$\sqrt{1 - 2\mu\eta_t + L^2\eta_t^2} \leq 1 - \mu\eta_t + \frac{L^2}{2}\eta_t^2.$$

Substituting this into the earlier bound and taking a square-root, we obtain the desired bound that

$$\sqrt{\delta_{t,j}^{\text{SGDw}}} \leq \left(1 - \eta_t\left(\mu - \alpha^m \sigma C_\chi - \frac{L^2}{2}\eta_t\right)\right)\sqrt{\delta_{t,j-1}^{\text{SGDw}}} + \eta_t\left(\varepsilon_{n,m,t;\nu}^{\text{SGDw}}(\epsilon) + \frac{5\sigma + 5\kappa_{\nu;m}}{\sqrt{B}}\right).$$

Moreover, since $L \geq \mu$ by definition from A1, we have

$$1 - \eta_t\left(\mu - \alpha^m \sigma C_\chi - \frac{L^2}{2}\eta_t\right) = \left(\frac{L}{\sqrt{2}}\eta_t - 1\right)^2 + (\sqrt{2}L - \mu)\eta_t + \alpha^m \sigma C_\chi \eta_t > 0,$$

which finishes the proof. $\qquad \square$

## F.1 Proof of Theorem B.1

Under A1, A2, A3, A4 and A7, Lemma F.1 implies

$$\sqrt{\delta_{t,j}^{\text{SGDw}}} \leq \left(1 - \tilde{\mu}_m C t^{-\beta} + \frac{L^2 C^2}{2} t^{-2\beta}\right)\sqrt{\delta_{t,j-1}^{\text{SGDw}}} + C t^{-\beta} \sigma_{n,T}^{\text{SGDw}}$$

for $1 \leq t \leq T$ and $1 \leq j \leq N$, where we have used $\tilde{\mu}_m = \mu - \alpha^m \sigma C_\chi$, $\eta_t = C t^{-\beta}$ and

$$\varepsilon_{n,m,t;\nu}^{\text{SGDw}}(\epsilon) + \frac{5\sigma + 5\kappa_{\nu;m}}{\sqrt{B}} \leq \varepsilon_{n,m,T;\nu}^{\text{SGDw}}(\epsilon) + \frac{5\sigma + 5\kappa_{\nu;m}}{\sqrt{B}} = \sigma_{n,T}^{\text{SGDw}}.$$

By an induction on $j = 1, \ldots, N$, we have

$$\sqrt{\delta_{t,N}^{\text{SGDw}}} \leq \left(1 - \tilde{\mu}_m C t^{-\beta} + \frac{L^2 C^2}{2} t^{-2\beta}\right)^N \sqrt{\delta_{t,0}^{\text{SGDw}}}$$

$$+ C \sigma_{n,T}^{\text{SGDw}} t^{-\beta} \sum_{j=1}^{N}\left(1 - \tilde{\mu}_m C t^{-\beta} + \frac{L^2 C^2}{2} t^{-2\beta}\right)^{j-1}.$$

By noting that $\delta_{t,0}^{\text{SGDw}} = \delta_{t-1,N}^{\text{SGDw}}$ almost surely for $t \geq 2$ and using another induction on $t = 1, \ldots, T$,

$$\sqrt{\delta_{T,N}^{\text{SGDw}}} \leq Q_0 \sqrt{\delta_{0,0}^{\text{SGDw}}} + C \sigma_{n,T}^{\text{SGDw}} \sum_{t=1}^{T} Q_t A_t t^{-\beta}, \qquad (18)$$

where

$$Q_t := \prod_{s=t+1}^{T}\left(1 - \tilde{\mu}_m C s^{-\beta} + \frac{L^2 C^2}{2} s^{-2\beta}\right)^N \quad \text{and} \quad A_t := \sum_{j=1}^{N}\left(1 - \tilde{\mu}_m C t^{-\beta} + \frac{L^2 C^2}{2} t^{-2\beta}\right)^{j-1}.$$

Note that by Lemma F.1, we also have that $1 - \tilde{\mu}_m C t^{-\beta} + \frac{L^2 C^2}{2} t^{-2\beta} > 0$. This allows us to apply Lemma C.3 to obtain

$$\kappa_0 \leq \exp\left(1 - N\tilde{\mu}_m C \varphi_{1-\beta}(T+1) + \frac{NL^2 C^2}{2}\varphi_{1-2\beta}(T+1)\right) = E_1^{T,N}.$$

To control $\sum_{t=1}^{T} Q_t A_t t^{-\beta}$, recall that $\beta \in [0,1]$, $\tilde{\mu}_m > 0$, $\frac{L^2 C^2}{2} > 0$, and that

$$E_2^{T,N} = \exp\left(-\frac{N\tilde{\mu}_m C}{2}\varphi_{1-\beta}(T+1) + 2NL^2 C^2 \varphi_{1-2\beta}(T+1)\right).$$

We can now apply Lemma C.3: If $\beta \notin \{\frac{1}{2}, 1\}$, then

$$\sum_{t=1}^{T} Q_t A_t t^{-\beta} \leq \frac{2^{2\beta+1}}{\tilde{\mu}_m C} e^{\frac{\tilde{\mu}_m C}{2(1-\beta)} \frac{N}{(T+1)^\beta}} + \frac{3^\beta (1+\tilde{\mu}_m C)^{N-1}(T+2)^\beta}{L^2 C^2} E_2^{T,N}.$$

If $\beta = \frac{1}{2}$, i.e. $2\beta = 1$, we have

$$\sum_{t=1}^{T} Q_t A_t t^{-\beta} \leq \frac{4}{\tilde{\mu}_m C} e^{\frac{\tilde{\mu}_m C N}{(T+1)^{1/2}}} + 2N(1+\tilde{\mu}_m C)^{N-1} \varphi_{\frac{1}{2}-L^2 C^2 N}(T+1) E_2^{T,N}.$$

If $\beta = 1$, we get that

$$\sum_{t=1}^{T} Q_t A_t t^{-\beta} \leq \frac{4}{\tilde{\mu}_m C} + \frac{3N\left(1+\frac{L^2 C^2}{2}\right)^{N-1} e^{2L^2 C^2 N} \log(T+1)}{(T+1)^{(\tilde{\mu}_m CN)/2}}$$

Substituting the bounds into (18), we get the desired bound that $\sqrt{\delta_{T,N}^{\mathrm{SGDw}}}$ is upper bounded by

$$
\begin{cases}
E_1^{T,N}\sqrt{\delta_{0,0}^{\mathrm{SGDw}}} + C\sigma_{n,T}^{\mathrm{SGDw}}\left(\dfrac{4e^{\frac{\tilde{\mu}_m CN}{(T+1)^{1/2}}}}{\tilde{\mu}_m C} + 2N(1+\tilde{\mu}_m C)^{N-1}\varphi_{\frac{1}{2}-L^2 C^2 N}(T+1)\,E_2^{T,N}\right) \\
\hfill \text{for } \beta = \frac{1}{2}\,, \\[2ex]
E_1^{T,N}\sqrt{\delta_{0,0}^{\mathrm{SGDw}}} + C\sigma_{n,T}^{\mathrm{SGDw}}\left(\dfrac{4}{\tilde{\mu}_m C} + \dfrac{3N\left(1+\frac{L^2 C^2}{2}\right)^{N-1} e^{2L^2 C^2 N}\log(T+1)}{(T+1)^{(\tilde{\mu}_m CN)/2}}\right) \hfill \text{for } \beta = 1\,, \\[2ex]
E_1^{T,N}\sqrt{\delta_{0,0}^{\mathrm{SGDw}}} + C\sigma_{n,T}^{\mathrm{SGDw}}\left(\dfrac{2^{2\beta+1}}{\tilde{\mu}_m C} e^{\frac{\tilde{\mu}_m C}{2(1-\beta)} \frac{N}{(T+1)^\beta}} + \dfrac{3^\beta(1+\tilde{\mu}_m C)^{N-1}(T+2)^\beta}{L^2 C^2}\,E_2^{T,N}\right) \text{ otherwise }.
\end{cases}
$$

$\square$

## F.2 Proof of Theorem B.2

Recall that $\delta_{t,j}^{\mathrm{SGDo}} := \mathbb{E}\left\|\psi_{t,j}^{\mathrm{SGDo}} - \psi^*\right\|^2$. To apply the results from Appendix E to $\psi_{t,j}^{\mathrm{SGDo}}$, we condition on $S_t^o$, which in particular fixes $S_{t,j}^o$, the last size-$B$ subset of $[n]$ chosen. We then identify $\theta^{\mathrm{init}} = \psi_{t,j-1}^{\mathrm{SGDo}}$, $\eta = \eta_t$ and the dataset used as $\mathcal{D}_{t,j}^o := (X_i : i \in S_{t,j}^o)$, which allows us to identify $\psi_{t,j}^{\mathrm{SGDo}} = \theta_m^{\mathrm{GD}}$, the full-batch gradient descent update using $\mathcal{D}_{t,j}^o$. Meanwhile, we note that almost surely $\theta_m^{\mathrm{GD}} = \theta_{m,B}^{\mathrm{SGDw}}$, the SGD-with-replacement iterate that uses the full dataset $\mathcal{D}_{t,j}^o$. Observe that the proof of Lemma F.1 holds with $\delta_{t,j-1}^{\mathrm{SGDo}}$ replaced by any random initialization $\theta^{\mathrm{init}}$ possibly correlated with $X_1, \ldots, X_n$, which allows us to obtain

$$\sqrt{\delta_{t,j}^{\mathrm{SGDo}}} \leq \left(1 - \eta_t\left(\mu - \alpha^m \sigma C_\chi - \frac{L^2}{2}\eta_t\right)\right)\sqrt{\delta_{t,j-1}^{\mathrm{SGDo}}} + \eta_t\left(\varepsilon_{n,m,t;\nu}^{\mathrm{SGDw}}(\epsilon) + \frac{5\sigma + 5\kappa_{\nu;m}}{\sqrt{B}}\right).$$

Since the error recursion for $\delta_{t,j}^{\mathrm{SGDo}}$ is identical to that of $\delta_{t,j}^{\mathrm{SGDw}}$ in Lemma F.1, the proof of Theorem 4.3 follows directly, thereby yielding an identical result for $\delta_{T,N}^{\mathrm{SGDo}}$ as with $\delta_{T,N}^{\mathrm{SGDw}}$ in Theorem 4.3. $\square$

# G  Proofs for tail probability bounds in offline SGD

We present the proofs for results in Appendix B.3 that control the tail probability terms $\vartheta_{\nu;n,m,T}^{\mathrm{SGDw}}$ and $\varepsilon_{\nu;n,m,T}^{\mathrm{SGDw}}$.

*Proof of Lemma B.3.* For $\delta > 0$, let $N_\delta$ be the $\delta$-covering number of $\Psi$, which satisfies $N_\delta \leq \left(r_\Psi(1+2/\delta)\right)^p$ (Example 5.8, [50]). Note also that by the Jensen's inequality applied to $\mathbb{E}[\bullet\,|X_1]$ and Assumption A6, there exist some $\sigma_m, \zeta_m > 0$ such that, for any $z \in \mathbb{R}^p$ with $\|z\| \leq \zeta_m$,

$$\mathbb{E}\big[e^{z^\top\left(\mathbb{E}[\phi(K_{\psi^*}^m(X_1))|X_1] - \mathbb{E}[\phi(K_{\psi^*}^m(X_1))]\right)}\big] \leq \mathbb{E}\big[e^{z^\top\left(\phi(K_{\psi^*}^m(X_1)) - \mathbb{E}[\phi(K_{\psi^*}^m(X_1))]\right)}\big] \leq e^{\sigma_m^2\|z\|^2/2}.$$

Meanwhile under A5, recycling the proof of Lemma 3.1 of [21] shows that if $C_m\delta/\sqrt{p} < \sigma_m^2\zeta_m$,

$$\mathbb{P}\Big(\sup_{\psi\in\Psi}\Big\|\frac{1}{n}\sum_{i=1}^n\mathbb{E}\big[\phi\big(K_\psi^m(X_i)\big)\big|X_i\big]-\mathbb{E}\big[\phi(K_\psi^m(X_i))\big]\Big\| > 3C_m\delta\Big) \le 2N_\delta p\exp\Big(-\frac{nC_m^2\delta^2}{2p\sigma_m^2}\Big)$$

$$\le 2(r_\Psi)^p\exp\Big(p\log(1+2\delta^{-1})-\frac{nC_m^2\delta^2}{2p\sigma_m^2}\Big).$$

Since the probability above is an upper bound to $\vartheta_{n,m,T}^{\mathrm{SGDw}}(3C_m\delta)$, we get that if $C_m\delta/\sqrt{p} < \sigma_m^2\zeta_m$,

$$\big(\varepsilon_{\nu;n,m,T}^{\mathrm{SGDw}}(3C_m\delta)\big)^2 = 9C_m^2\delta^2 + \kappa_{\nu;m}^2\Big(\vartheta_{n,m,T}^{\mathrm{SGDw}}\Big(\frac{C_m\delta}{\sqrt{p}}\Big)\Big)^{\frac{\nu-2}{\nu}}$$

$$\le 9C_m^2\delta^2 + \kappa_{\nu;m}^2 2^{\frac{2(\nu-2)}{\nu}}(r_\Psi)^{\frac{(\nu-2)p}{\nu}}\exp\Big(\frac{(\nu-2)p}{\nu}\log(1+2\delta^{-1})-\frac{n(\nu-2)C_m^2\delta^2}{2\nu p\sigma_m^2}\Big).$$

Recall that by assumption, $\frac{\log n}{n} < \frac{\sigma_m^2\zeta_m^2}{p+\nu-2}$. We now choose

$$\delta = \frac{\sigma_m}{C_m}\sqrt{p\Big(p+\frac{2\nu}{\nu-2}\Big)\times\frac{\log n}{n}} = \frac{\sigma_m}{C_m}\sqrt{p\times\frac{(\nu-2)p+2\nu}{\nu-2}\times\frac{\log n}{n}},$$

which implies

$$\inf_{\epsilon>0}\big(\varepsilon_{\nu;n,m,T}^{\mathrm{SGDw}}(\epsilon)\big)^2 \le \big(\varepsilon_{\nu;n,m,T}^{\mathrm{SGDw}}(3C_m\delta)\big)^2$$

$$\le \frac{9\sigma_m^2 p((\nu-2)p+2\nu)\log n}{(\nu-2)n} + \kappa_{\nu;m}^2 2^{\frac{\nu-2}{\nu}}(r_\Psi)^{\frac{(\nu-2)p}{\nu}}(1+2\delta^{-1})^{\frac{(\nu-2)p}{\nu}}\exp\Big(-\frac{(\nu-2)p+2\nu}{2\nu}\log n\Big)$$

$$\le \frac{9\sigma_m^2 p((\nu-2)p+2\nu)\log n}{(\nu-2)n}$$

$$+ \kappa_{\nu;m}^2 2^{\frac{\nu-2}{\nu}}(r_\Psi)^{\frac{(\nu-2)p}{\nu}}\Big(1+\frac{2C_m(\nu-2)^{1/2}}{\sigma_m p^{1/2}((\nu-2)p+2\nu)^{1/2}}\frac{\sqrt{n}}{\sqrt{\log n}}\Big)^{\frac{(\nu-2)p}{\nu}}n^{-\frac{(\nu-2)p+2\nu}{2\nu}}$$

$$\le \frac{9\sigma_m^2 p((\nu-2)p+2\nu)\log n}{(\nu-2)n}$$

$$+ \kappa_{\nu;m}^2 2^{\frac{\nu-2}{\nu}}(r_\Psi)^{\frac{(\nu-2)p}{\nu}}\Big(1+\frac{2C_m(\nu-2)^{1/2}}{\sigma_m p^{1/2}((\nu-2)p+2\nu)^{1/2}}\Big)^{\frac{(\nu-2)p}{\nu}}n^{\frac{(\nu-2)p}{2\nu}-\frac{(\nu-2)p+2\nu}{2\nu}}$$

$$\le \Big(\frac{9\sigma_m^2 p((\nu-2)p+2\nu)}{\nu-2} + \kappa_{\nu;m}^2 2^{\frac{(\nu-2)}{\nu}}(r_\Psi)^{\frac{(\nu-2)p}{\nu}}\Big(1+\frac{2C_m(\nu-2)^{1/2}}{\sigma_m p^{1/2}((\nu-2)p+2\nu)^{1/2}}\Big)^{\frac{(\nu-2)p}{\nu}}\Big)\frac{\log n}{n}.$$

Taking a squareroot across and using $\sqrt{a+b}\le\sqrt{a}+\sqrt{b}$ for $a,b>0$ gives the desired bound. The limiting result follows by substituting this bound into Theorem B.1. $\qquad\square$

*Proof of Lemma B.4.* Let $\delta > 0$ and $N_\delta$ be defined as in the proof of Lemma B.3, with $N_\delta \le \big(r_\Psi(1+2/\delta)\big)^p$. Let $(\psi_l)_{l=1}^{N_\delta}$ be the centers of the covering $\delta$-balls. The covering-ball argument of the proof of Lemma 3.1 of [21] shows that

$$\vartheta_{n,m,T}^{\mathrm{SGDw}}(3C_m\delta) \le \mathbb{P}\Big(\sup_{\psi\in\Psi}\Big\|\frac{1}{n}\sum_{i=1}^n\mathbb{E}\big[\phi\big(K_\psi^m(X_i)\big)\big|X_i\big]-\mathbb{E}\big[\phi(K_\psi^m(X_i))\big]\Big\| > 3C_m\delta\Big)$$

$$\le \sum_{l=1}^{N_\delta}\mathbb{P}\Big(\Big\|\frac{1}{n}\sum_{i=1}^n\mathbb{E}\big[\phi\big(K_{\psi_l}^m(X_i)\big)\big|X_i\big]-\mathbb{E}\big[\phi(K_{\psi_l}^m(X_i))\big]\Big\| \ge C_m\delta\Big).$$

By a Markov's inequality followed by the Burkholder's inequality applied to an average of i.i.d. summands (see e.g. [51] for $\nu > 2$), there exists a constant $C_\nu > 0$ depending only on $\nu$ such that

$$\vartheta_{n,m,T}^{\mathrm{SGDw}}(3C_m\delta) \le \sum_{l=1}^{N_\delta}\frac{\mathbb{E}\big\|\sum_{i=1}^n\mathbb{E}\big[\phi\big(K_{\psi_l}^m(X_i)\big)\big|X_i\big]-\mathbb{E}\big[\phi(K_{\psi_l}^m(X_i))\big]\big\|^\nu}{n^\nu C_m^\nu\delta^\nu}$$

$$\le \sum_{l=1}^{N_\delta}\frac{\mathbb{E}\big\|\mathbb{E}\big[\phi\big(K_{\psi_l}^m(X_1)\big)\big|X_i\big]-\mathbb{E}\big[\phi(K_{\psi_l}^m(X_1))\big]\big\|^\nu}{n^{\nu/2}C_m^\nu\delta^\nu}$$

$$\le \frac{N_\delta\kappa_{\nu;m}^\nu}{n^{\nu/2}C_m^\nu\delta^\nu} \le \frac{(r_\Psi)^p(1+2\delta^{-1})^p\kappa_{\nu;m}^\nu}{n^{\nu/2}C_m^\nu\delta^\nu},$$

where we have used assumption A4 in the last line. This implies

$$\left(\varepsilon_{\nu;n,m,T}^{\text{SGDw}}(3C_m\delta)\right)^2 = 9C_m^2\delta^2 + \kappa_{\nu;m}^2\left(\vartheta_{n,m,T}^{\text{SGDw}}(3C_m\delta)\right)^{(\nu-2)/\nu}$$

$$\leq 9C_m^2\delta^2 + \kappa_{\nu;m}^\nu(r_\Psi)^{\frac{(\nu-2)p}{\nu}}C_m^{-(\nu-2)} \times \frac{(1+2\delta^{-1})^{\frac{(\nu-2)p}{\nu}}}{n^{\frac{\nu-2}{2}}\delta^{\nu-2}} \ .$$

Choosing $\delta = n^{-\frac{(\nu-2)\nu}{2(\nu^2+(\nu-2)p)}} \leq 1$, we get that

$$\inf_{\epsilon>0}\left(\varepsilon_{\nu;n,m,T}^{\text{SGDw}}(\epsilon)\right)^2 \leq \left(\varepsilon_{\nu;n,m,T}^{\text{SGDw}}(3C_m\delta)\right)^2$$

$$\leq 9C_m^2 n^{-\frac{(\nu-2)\nu}{\nu^2+(\nu-2)p}} + \kappa_{\nu;m}^\nu(r_\Psi)^{\frac{(\nu-2)p}{\nu}}C_m^{-(\nu-2)}3^{\frac{(\nu-2)p}{\nu}}n^{-\frac{(\nu-2)\nu}{\nu^2+(\nu-2)p}}$$

$$= \left(9C_m^2 + \kappa_{\nu;m}^\nu(r_\Psi)^{\frac{(\nu-2)p}{\nu}}C_m^{-(\nu-2)}3^{\frac{(\nu-2)p}{\nu}}\right)n^{-\frac{(\nu-2)\nu}{\nu^2+(\nu-2)p}} \ .$$

Taking a squareroot across and using $\sqrt{a+b} \leq \sqrt{a} + \sqrt{b}$ for $a,b>0$ gives the desired bound. The limiting result follows by substituting this bound into Theorem B.1. $\qquad\square$

*Proof of Lemma B.5.* By a Markov's inequality and a Jensen's inequality with respect to the empirical average, we have

$$\vartheta_{n,m,T}^{\text{SGDw}}(\epsilon) \leq \sup_{t\in[T],j\in[N]} \frac{\sum_{i=1}^n \mathbb{E}\left\|\mathbb{E}\left[\phi\left(K_{\psi_{t-1,j}^{\text{SGDw}}}^m(X_i)\right)\Big|X_i,\psi_{t-1,j}^{\text{SGDw}}\right] - \mathbb{E}\left[\phi\left(K_{\psi_{t-1,j}^{\text{SGDw}}}^m(X_1')\right)\Big|\psi_{t-1,j}^{\text{SGDw}}\right]\right\|}{n\epsilon}$$

$$=: \sup_{t\in[T],j\in[N]} \frac{\sum_{i=1}^n A_{tji}}{n\epsilon} \ .$$

Meanwhile by a triangle inequality and the ergodicity assumption, we have

$$A_{tji} \leq \mathbb{E}\left\|\mathbb{E}\left[\phi\left(K_{\psi_{t-1,j}^{\text{SGDw}}}^m(X_i)\right)\Big|X_i,\psi_{t-1,j}^{\text{SGDw}}\right] - \mathbb{E}\left[\phi\left(X_1^{\psi_{t-1,j}^{\text{SGDw}}}\right)\Big|\psi_{t-1,j}^{\text{SGDw}}\right]\right\|$$

$$+ \mathbb{E}\left\|\mathbb{E}\left[\phi\left(X_1^{\psi_{t-1,j}^{\text{SGDw}}}\right)\Big|\psi_{t-1,j}^{\text{SGDw}}\right] - \mathbb{E}\left[\phi\left(K_{\psi_{t-1,j}^{\text{SGDw}}}^m(X_1')\right)\Big|\psi_{t-1,j}^{\text{SGDw}}\right]\right\|$$

$$\leq 2\tilde{C}_K\tilde{\alpha}^m \ .$$

This implies $\vartheta_{n,m,T}^{\text{SGDw}}(\epsilon) \leq 2\tilde{C}_K\tilde{\alpha}^m\epsilon^{-1}$, and therefore

$$\left(\varepsilon_{\nu;n,m,T}^{\text{SGDw}}(\epsilon)\right)^2 \leq \epsilon^2 + 2^{\frac{\nu-2}{\nu}}\kappa_{\nu;m}^2(\tilde{C}_K)^{\frac{\nu-2}{\nu}}\tilde{\alpha}^{\frac{(\nu-2)m}{\nu}}\epsilon^{-\frac{\nu-2}{\nu}} \ .$$

Choosing $\epsilon = \tilde{\alpha}^{(\nu-2)m/(3\nu-2)}$ gives

$$\inf_{\epsilon>0}\left(\varepsilon_{\nu;n,m,T}^{\text{SGDw}}(\epsilon)\right)^2 \leq \left(1 + 2^{\frac{\nu-2}{\nu}}\kappa_{\nu;m}^2(\tilde{C}_K)^{\frac{\nu-2}{\nu}}\right)\tilde{\alpha}^{\frac{2(\nu-2)m}{3\nu-2}} \ .$$

Taking a squareroot across and using $\sqrt{a+b} \leq \sqrt{a} + \sqrt{b}$ for $a,b>0$ gives the desired bound. The limiting result follows by substituting this bound into Theorem B.1. $\qquad\square$

