# OpenReview forum: "Near-Optimality of Contrastive Divergence Algorithms"
_NeurIPS.cc/2024/Conference — NeurIPS 2024 poster_

### Official Review · Reviewer_b7nK · 2024-07-10

**Soundness:** 3
**Presentation:** 3
**Contribution:** 3
**Rating:** 6
**Confidence:** 2

**Summary:**

The paper provides a non-asymptotic analysis of the Contrastive Divergence (CD) algorithms, demonstrating their ability to achieve parametric convergence rates and near-optimal asymptotic variance in the online setting. It also shows near-parametric rates in the offline setting, extending previous results. The analysis is applicable to unnormalized exponential families, which, despite their flexibility, cover only log-densities with linear relationships on the model parameters.
The analysis also covers various data batching schemes, including fully online and minibatch methods.
The study highlights the importance of extending these results to more general models in future work. Additionally, the assumption of $\chi^2$-regularity of the exponential family model is crucial for mapping Markov contractions back to the parameter space. Future research could consider alternative conditions on the Markov Chain Monte Carlo (MCMC) chains, such as geometric ergodicity, to avoid the presence of potentially large regularity constants.

**Strengths:**

**Originality**: The paper provides a novel non-asymptotic analysis of the Contrastive Divergence (CD) algorithm, significantly extending the understanding of its convergence properties. Unlike prior work that focused on asymptotic convergence rates, this study shows that CD can achieve the parametric rate $O(n^{-1/2})$, which is a substantial improvement over the previously established rate of $O(n^{-1/3})$. This originality is further enhanced by applying the analysis to various data batching schemes, including online and minibatch settings.

**Quality**: The theoretical contributions are robust and well-supported by rigorous mathematical analysis. The paper meticulously outlines the assumptions required for their results, such as $\chi^2$-regularity, and provides clear justifications for these choices. The comprehensive approach to evaluating the performance of CD in different batching schemes adds depth to the quality of the research.

**Clarity**: The paper is clearly written and well-structured, making it accessible to a broad audience, including those who may not be deeply familiar with the nuances of CD or unnormalized models. The authors effectively explain complex concepts and provide a logical flow from the introduction to the conclusion. The distinction between the contributions of this work and previous research is clearly articulated, enhancing the reader's understanding of the paper's significance.

**Significance**: The findings have significant implications for the field of machine learning, particularly in the training of unnormalized models. By demonstrating that CD can achieve near-optimal asymptotic variance and parametric convergence rates, the paper provides valuable insights that can influence future research and practical applications. The potential to extend these results to more general models opens up new avenues for exploration and underscores the broader impact of this work.

Overall, the paper makes a substantial contribution to the theoretical understanding of the CD algorithm, providing new insights that are both original and significant. The high quality of the analysis and the clarity of the presentation further enhance the impact of the research.

**Weaknesses:**

While I acknowledge that I am not an expert in this specific area, I have some suggestions that could improve the paper towards its stated goals. Although I did not thoroughly check the proofs due to the extensive 30-page appendix, here are my detailed comments:

**Experimental Validation**: While the primary focus of this paper is theoretical, it would be beneficial to include an intuitive demonstration of the $O(n^{-1/2})$ rate, if possible. This could help bridge the gap between theory and practical understanding, making the results more accessible and relatable to a broader audience.

**Clarity and Structure**: The paper currently presents a substantial amount of theoretical content, which can be overwhelming, especially for readers who may not be deeply familiar with the notation and concepts used. Highlighting the new techniques that significantly improve upon previous work would make the contributions clearer.
Consider focusing on one main theorem with detailed lemmas and techniques that build towards it. This approach could provide a more structured and digestible presentation, making it easier for readers to follow the logical progression of the work.

**Assumptions and Practicality**: The paper requires a slightly stronger moment assumption (Assumption A4) to achieve better results. Providing more detailed examples and explanations that demonstrate the practical acceptability of this assumption would be valuable. This could help convince readers that the stronger assumption is reasonable and applicable in real-world scenarios.

I welcome further discussion on these points to ensure a comprehensive evaluation. Thanks!

**Questions:**

See the weaknesses part.

**Limitations:**

Yes, the authors adequately addressed the limitations in conclusion part. The paper is theoretical, and I believe there is no need to confirm societal impact.

---

> ### Author Rebuttal · Authors · 2024-08-07
>
> We thank the reviewer for their helpful comments. We address their concerns below.
>
> **Regarding the techincality of the paper**  We agree that Section 4 is technical, and that it is crucial to increase its accessbility to a broad readership. We will take multiple actions to improve the accessibility of Section 4, as discussed in the general comment.
>
>
> **Regarding Assumption 4**
> Assumption A4 is a moment assumption on the MCMC sample $\phi(K^m_\psi(x))$. This assumption will hold as soon as the tails of this random variable are light enough. For example, if $\phi(K^m_\psi(x))$ has a sub-Gaussian tail, A4 automatically holds for any positive $\nu$ with $\kappa_{\nu;m} = O(\sqrt{\nu})$; if $\phi(K^m_\psi(x))$ has a sub-exponential tail, A4 holds for any positive $\nu$ with $\kappa_{\nu;m} = O(\nu)$. In fact, we just require A4 to hold for some fixed $\nu > 2$. Characterizing the tail of this random variable can often be done by inspecting the Markov kernel used. We will add this discussion in the camera-ready version of the paper.
>
> **Regarding empirical validation** We thank the reviewer for their suggestions. We agree that experimental validation of the theory is helpful and will add a synthetic experiment where the ground truth is known in order to exemplify the claims made in our theory.
> We follow the setup of [21, Section 4.1] and consider the following model:
> \begin{equation*} \label{}
> \begin{aligned}
>     p(x; \psi) = \mathcal  N(\psi, \Sigma), \quad \quad \Sigma = \begin{bmatrix} 1 & 0.5 \\\\ 0.5 & 1 \end{bmatrix}
> \end{aligned}
> \end{equation*}
> for $\Psi = \left \lbrack -0.5, 0.5 \right \rbrack^2$, which we make an unnormalized exponential family for $\phi(x) = \Sigma^{-1} x$, $\log Z(\psi) = \frac{1}{2} \psi^T \Sigma^{-1} \psi$, implying $\nabla_{  }^2 \log Z(\psi) = \Sigma^{-1}$. It can be shown (see discussion below) that assumptions A1-A6 are verified for this model. Moreover, the quantities involved in the bounds can be computed explicitly (or upper bounded) explicitly, which allows us to validate our bounds on this simple example.
>
> **Experimental setup** We run 500 different experiments with different seeds to compute an empirical estimate of $\delta_n = \mathbb{ E } \| \widehat{ \psi }_n - {\psi}^{\star} \|^2$ up to $n = 2000$ samples, where $\widehat{ \psi }_n$  is the estimate of $\psi^{\star}$ obtained by the online CD, online CD with averaging, and offline CD estimators, respectively. We then plot this empirical estimate as a function of the number of samples $n$ , and compare it to both our upper bounds of our paper obtained in each setting, as well the Cramer-Rao lower bound. Our learning rate schedule is $C n^{-\beta}$, with $C = 0.1$ and $\beta = 0.6$. For offline CD, we make $5000$ gradient steps using the full dataset, and run the experiments for 10 evenly log-spaced values of $ n $ between $ 100 $ and $2000$. We use $m =20$ MCMC steps.
>
>
> **Plotting lower and upper bounds** In the pdf included in the general comment, we plot:
> - The full non-asymptotic upper bound related to the last-iterate online CD estimator
> - The leading term of the non-asymptotic upper bound related to the averaged online CD estimator (for simplicity, due to the complexity of the higher-order terms)
> - The non-asymptotic in $n$, but asymptotic in $T$ upper bound related to the offline CD estimator (for similar reasons).
>
> We see that the bounds for online CD are tight, while the bounds for offline CD are looser, which is due to the large multiplicative constant in front of the leading term. However, our bounds for offline CD are still able to provably explain how offline CD is more robust to transient effects related to bad initialization and smoothness, as mentioned in the rebuttal to reviewer ouZy. Moreover, we see that offline CD performs competitively compared to online CD, which calls for more investigations of the offline setting. Further work will investigate how to tighten these bounds by, in particular, attempting to improve the constants.
>
> **Additional details for the model**
> It can be shown [21] that $\sigma^2 = 2.67$, $\mu = 0.67$, and $C_{\chi} \approx 1.84$, and that $\alpha \leq 1 - \lambda_{\min_{  }}(\Sigma^{-1} / 2) = 0.67$. For $m \geq  \lceil 3.5 \rceil = 4$, Theorem 3.2 will hold. Using the formulas present in the proofs, and the formulas for the cumulants of a Gaussain variable, we can show that $||\log Z||_{3, \infty} \approx 5.95$.
> For online CD with averaging, we also need $m(n) > \frac{(1 - \beta) \log n}{2\lvert \log \alpha \rvert }$. Assuming $n = 2000$ and $\beta =0.6$, we need $m(n) > 3.8$, which will hold for $m(n) \geq 5$. Finally, for offline CD, Assumption A4 to A6 will be satisfied [21].
>
> Moreover, it can be shown that $k_{\psi}^{m}(X)$ is a Gaussian random variable with mean $\psi ( 1 - \rho^{2m}) + {\psi}^{\star}$ and (component-wise) variances $1$, yielding (using formulas for the centered moments of Gaussian random variables and $(a^2 + b^2)^3 \leq 4 (a^{6} +  b^{6})$), we obtain $\kappa_{m, 6} \leq \lambda_{\max_{  }}(\Sigma ^{-1}) (\mathbb{ E } \| k_{\psi}^{m}(X) - \mathbb{ E } k_{\psi}^{m}(X) \|^{6})^{1 / 6} \leq 4 \cdot 12^{1 / 6} \approx 4.6$. Finally, it can also be shown that $\mathbb E [ k_{\psi}^{m}(X) | X]$ is Gaussian with mean $\psi ( 1 - \rho^{2m}) + \rho^{2m}X$ and component-wise variances $1 - \rho^{4m}$. Straightforward derivations can show that assumption A6 is verified for $\sigma_m = 2$.
>
> ---
>
> We have done our best to address your questions, and we would be happy to address further questions in the discussion. If we have done so, we would be grateful if you would consider increasing your score.

---

> > ### Comment · Reviewer_b7nK · 2024-08-10
> >
> > Thank you for the response. I maintain my current evaluation of the paper, as it is highly technical and covers areas I am unfamiliar with. I believe the paper could be improved based on the current modifications shown in the rebuttal. Good luck to you.

---

### Official Review · Reviewer_Fohj · 2024-07-13

**Soundness:** 3
**Presentation:** 2
**Contribution:** 4
**Rating:** 6
**Confidence:** 3

**Summary:**

This paper analyzes the contrastive divergence algorithm for learning exponential family distributions.
Both online and offline versions are studied. The main contribution of this paper is to establish the parametric convergence rate $O(n^{-1/2})$ of the estimator, which improves the previously best result of $O(n^{-1/3})$.

**Strengths:**

Efficiently learning unnormalized model, especially exponential family distributions, is an important problem.
Analyzing the CD algorithm is known to be difficult, and this paper seems to establish the best result, proving the parametric rate both in online and offline settings.
I like the gentle introduction of the CD in the first three pages.

**Weaknesses:**

- That said, the paper is extremely technical, loaded with heavy notation. Though I am aware of some related works on learning unnormalized models, this paper is hard to appreciate. It is hard to penetrate without extreme familiarity with the previous analysis in the literature and all the optimization analysis tools. I guess this paper could be certainly of some audience, but it'd be great if the authors can provide some more high-level explanations, pointing out some difficulties in the analysis, required techniques, and so on.
- In particular, assumptions are not with concrete examples. Can you provide a few examples of exponential family distributions that satisfy and don't satisfy each assumption?

**Questions:**

- Is it proper to say "near-optimality", if this refers to the parametric rate $O(n^{-1/2})$, and it indeed achieves CRLB up to a factor of 4? It's still not efficient due to the multiplicative factor. Is this dependence usually called "nearly optimal"? For example, compared to the NCE estimators, can you argue that what's guaranteed with the CD estimator is provably better in a certain regime? This kind of insights would be very useful.

**Limitations:**

There is no specific limitation noted in the paper.
It'd be very helpful if the authors can provide a comparison of assumptions and attained rates from previous papers to the ones in the paper, side-by-side.

---

> ### Author Rebuttal · Authors · 2024-08-07
>
> We thank the reviewer for their comments and suggestions. We address their concerns below.
>
> **On the technicality of the paper** We agree that the paper in its current form is technical, and agree that increasing the accessibility of our work to a broad readership is important. The main message is the following: under suitable assumptions,
> - Online CD can converge at a speed $O(n^{-1/2})$ (optimal rate), and the asymptotic variance of averaged online CD is $4$ times the CRLB. However, both variants contain transient higher order terms scaling exponentially badly with the smoothness parameters of the problem and linearly with the initialization $\sqrt \delta_0$.
> - Offline CD, on the other hand, converges at a near-optimal $O(\sqrt{\log n} / \sqrt{n})$ rate, but, for a fixed sample size $n$, the higher-order terms vanish as the number of gradient steps $T \to \infty$. Consequently, offline CD can be provably more robust than online CD to smoothness and initialization (see the rebuttal to reviewer ouZy for a detailed discussion).
>
> In conclusion, online CD is asymptotically near-optimal, and asymptotically better than offline CD, while offline CD may perform better in the non-asymptotic regime, where higher-order terms due to initialization and smoothness play a non-negligible role. This matches the expectation that multiple passes through the  data can improve statistical efficiency for CD. This discussion is summarized in the following table (which we thank the reviewer for suggesting)
>
> | Algorithm   | Rate | Leading term constant | Higher order terms |
> | -------- | ------- | --------------------- | --------------------- |
> | Online CD ($\beta=1$, Last iterate, A1 to A3 ours)  | $1/\sqrt{n}$ if $\tilde{\mu}_m C>2$, $1/n^{-\tilde{\mu}_m C / 4}$ otherwise   | $2\tilde{\sigma}_m C \frac{\tilde{\mu}_m C}{\tilde{\mu}_m C - 2}$ |  Exponential in $L$, linear in $\delta_0$
> | Online CD (Averaged iterates, A1 to A3, ours) | $1/\sqrt{n}$                                           | $2\sqrt{\text{tr}(\mathcal I(\psi^\star)^{-1})}$ | Exponential in $L$, linear in $\delta_0$
> | Offline CD (Last iterate, $B=n$, $\nu - 2 = 2p$, A1 to A6, ours)   | $\sqrt{\log(n)}/\sqrt{n}$  | Indep. of $L$, sublinear in $\delta_0$  | $o_{T}(1)$
> | Offline CD ($B=n$, Averaged iterates, [21] A1 to A6 with A4 only for $\nu = 2$)  | $n^{-1/3}$ (Asymptotic)  | $?$ | $?$
>
> We will improve the accessibility of the paper by taking the steps outlined in the general comment.
>
>
> **On the term near-optimality**: We use "near-optimality" to describe online CD because the factor of 4 scaling the CRLB is a universal constant, (independent of the parameters of the problems like smoothness or dimension).
>
> **Comparison with NCE** We thank the reviewer for suggesting to compare our results with other estimation methods for unnormalized models. The asymptotic variance of NCE is provided in [6, Theorem 3]. Its expression depends on the noise distribution used. More explicit formulas can be obtained for specific choices of the noise distribution:
> - If the noise distribution is  Gaussian, then it can be shown that for specific choices of data distributions, the asymptotic variance of NCE grows exponentially with the dimension of the data [5]. This is in contrast with our obtained bounds which do not contain any explicit exponential scaling with the dimension.
> - If the noise distribution is the unknown data distribution, then as the number of noise samples grows to $+\infty$,  the asymptotic variance of NCE approaches 2 times the CRLB [6, Corollary 7]. However, attaining that limit requires knowing the data distribution, which defeats the purpose of estimation.
>
> (We additionally compare with score matching in the next comment)
>
> We can thus see that both NCE can significantly depart from optimality (not just by a universal constant), which is unlike (averaged online) CD. On the other hand, CD requires assumptions for its bounds to hold, and in particular a lower bound on the number of MCMC steps, which can be large in some instances. CD can thus be more computationally expensive than these methods. Our point is thus not to argue that CD is uniformly better than NCE, but is able to trade-off computational power for statistical efficiency, which is useful when acquiring samples is costly.
>
> **On examples satisfying the assumptions:** We discuss the assumptions in m
> - Assumption A1 and A2 describe the exponential family. A1 will hold by putting a hard limit on the parameter norm. A2 will hold as soon as the domain (not $\Psi$) of the exponential family is the whole of $\mathbb R^d$.
> - Assumption A3 to A6 describe the Markov kernel, with A4, A6 focusing on its tails. A4 and A6 will hold for subexponential ones, such as MALA or Random Walk Metropolis Hasting. A5 is a smoothness constraint which should hold in most cases for compact $\Psi$. As discussed to reviewer yRNH, a Gaussian model with an unknown mean lying in a bounded subset, but known covariance matrix. This model satisfies all required assumptions, as discussed in the rebuttal to reviewer yRHN. The other examples of [21] also satisfy all assumptions.
>
>
> On the other hand, A Gaussian model with arbitrary precision matrix will not satisfy Assumption 1 (or Assumption A5), as the parameter set is not compact, and the log-likelihood function is not smooth. Heavy-tailed proposals (like a Student-t distribtution) are likely to not satisfy A6. Regarding Assumption A3, we expect that this assumption will often be verified. However, in some instances, $\alpha$ may be very close to 1, requiring many MCMC steps ($|\log \alpha |$)  for our bounds to hold. Instances where spectral gaps a very close to 1 include multi modal distributions with unknown mode proportions ([8, Example 2]).
>
> ---
>
> We have done our best to address your questions, and we would be happy to address further questions in the discussion. If we have done so, we would be grateful if you would consider increasing your score.

---

> ### Author Response · Authors · 2024-08-07
> **Discussion of proofs, comparision with score matching**
>
> In the following comment, we compare our bounds with existing ones for score matching, and provide a high level discussion of the proofs of the theoretical results in the manuscript.
>
> **Regarding score matching** The asymptotic variance of score-matching was analyzed in [8], where it is shown that:
> - Under a Poincaré Inequality, the asymptotic variance of score matching scales as the square of CRLB (times a smoothness term) (See [8, Theorem 2]).
> - For some specific exponential families, the asymptotic variance of score matching can scale exponentially badly with the parameters of the problem (see [8, Corollary 1]).
>
>
> **Regarding high-level explanation of the proofs** As suggested by the reviewer, to further improve the accessibility of the paper, we provide below a high-level discussion of each result of the main body, and the main proof ingredients used. The overall approach leading the online and offline bounds consists in using non-asymptotic arguments from the convex optimization literature, combined with a control of the bias (and the correlations in the offline setting) introduced by using the CD gradients. While the online setting was not studied in prior analyses of CD, the offline setting was studied in [21]. Our approach significantly departs from [21], which obtained its results by considering constant step sizes $\eta_t = \eta$, thanks to which the sequence of iterates $\psi_t$ becomes a Markov Chain. [21] then obtain convergence by showing the recurrence of this Markov chain in a ball containing the MLE. We now discuss the online and the offline results in more detail.
>
> **High level discussion of the online result proofs** In the online setting, the fact that every sample $x_t$ is only used once makes $\psi_{t}$ and $x_{t}$ conditionally independent given past information. Thanks to this conditional independence, the final recursion is free of correlation terms, and we only need to control the bias and variance induced by using CD instead of standard SGD. The lack of correlations allows to obtain an optimal rate of convergence. A fine control of the bias and the variance, as done in Theorem 3.3, allows to show the that the averaged CD iterates are near-optimal estimators. This correlation-free setting also allows to get rid of multiple assumptions compared the offline analysis of [21].
>
> - Lemma 3.1 is a recursion on the iterates describing the distance of of the estimated parameter to the true parameter during optimization. This recursion is obtained by combining standard arguments in the convex optimization litterature (See [22, Equation 16]) with a control of the approximation error due to using the CD gradient instead of the true likelihood gradient. This control uses arguments similar in spirit to [21, Lemma 3.2] although with considerable simplification thanks to the online setup.
> - Theorem 3.2 provides a bound on the distance of the n-th iterate of the previous sequence, by "unrolling" the recursion. The form of the recursion allows to directly invoke [20] to unroll the recursion, allowing for a very short argument.
> - Theorem 3.3 provides a bound on $\mathbb E ||1/n \sum_{t=1}^{n} \psi_t - \psi^\star ||^2$, the distance of the averaged CD iterates to the true parameter. The proof consists in adapting the argument of [20, Theorem 3], with multiple modifications to account for the fact that
>   - The CD gradient is an unbiased estimate of a *biased* gradient. Consequently, a different decomposition to the one in the proof of [20, Theorem 3] must be used (l. 1133 to l. 1139 of our manuscript)
>   - The CD gradient contains an additional source of noise coming from from using one MCMC sample to approximate $\mathbb E_{p_{\psi^\star}} \phi$. However (and this is key to our final result), we relate this additional noise to the Fisher Information of the model at the true parameter, which is why the CRLB is multiplied by factor of 4.
>   - While averaging the CD iterates yields an estimator with smaller variance, it induces the accumulation of a bias term, which we can control by scaling the number of MCMC steps with $\log n$.
>
>
> **High-level discussion of the offline results proof** In the offline setting, already studied in [21], the conditional independence structure of the online setting disappears. Instead, the iterates $\psi_t$ are correlated with the data points $x_t$, and these correlations need to be controlled to obtain our final bounds. From a technical point of view, the control of these correlations constitutes the main difference between the online and the offline setting (see Lemma E.2), and requires arguments that are specific to our setting, and not standard in the SGD literature. The rest of the arguments used to obtain our bounds (Lemma E.1, E.3 and E.4) are either adaptations from classical results in the SGD literature, or similar in spirit to the one developed in the online section (with adaptations to the offline setting).

---

> > ### Comment · Reviewer_Fohj · 2024-08-12
> >
> > I really appreciate the authors' detailed response.
> > As most of the reviewers raised concerns of its extreme technicality, the paper will be better appreciated by a larger audience, after a through revision as explained and promised by the authors.
> > I'd like to say that the added simulation is also nice, in that it confirms the theory by a simple example.
> > I believe that the authors will do the best in revising the manuscript.
> > I will keep the score as is.

---

### Official Review · Reviewer_ouZy · 2024-07-13

**Soundness:** 3
**Presentation:** 2
**Contribution:** 3
**Rating:** 6
**Confidence:** 4

**Summary:**

The paper studies parameter estimation problem in naturally-parametrized exponential families using the contrastive divergence methods, i.e., SGD where the stochastic gradients are estimated via MCMC. Both online and offline variants of the algorithms are considered. Building upon existing works on stochastic approximation, the paper establishes upper bounds on the last iterate and averaged iterate for online algorithms. In the offline setting, the paper establishes a near-parametric rate under certain tail assumptions.

**Strengths:**

Contrastive divergence algorithms have been used in practice, and it is interesting to provide some theoretical guarantees. The paper made solid contribution by proving sharp rate of convergence under various settings. The results provide methodological guidance on the choice of tuning parameters such as stepsizes and Monte Carlo trajectory length. The analysis in the online setting is clean and conceptually straightforward, while the results are strong.

**Weaknesses:**

On the technical side, the results are mostly applying existing techniques directly. The online results are mostly from Bach and Moulines (2011), with a careful analysis of the biases.

I'm not totally convinced by the usefulness of the offline algorithm. It seems that the analysis is much more involved and requries additional technical assumptions. On the other hand, the results are weaker than the online setting.

**Questions:**

- Unlike Bach and Moulines (2011), the claim of matching Cram\'{e}r--Rao lower bound in Theorem 3.3 involves an additional factor of 2. I think this is unavoidable because of the additional variance from the Monte Carlo sample. This needs to be discussed more thoroughly.
- The writing of Section 4 needs some improvement. The notations are extremely heavy, making the main message difficult to follow. I would suggest simplifying the expression under optimal choice of parameters, and present the main result clearly. The heavy notations can be introduced in the proof sections.
Minor comments:
- Eq (5). Definition of $\mu$ is incorrect. Strong convexity parameter should be the minimum eigenvalue, instead of the operator norm.

**Limitations:**

The paper addressed limitations of the results adequately.

---

> ### Author Rebuttal · Authors · 2024-08-07
>
> We thank the reviewer for their constructive review. We address the reviewer's concerns below.
>
> **Regarding the benefits of offline CD** We agree that the analysis of offline CD is more delicate than the one of online CD. On the other hand, we stress that while online CD is asymptotically optimal, *offline CD can perform better than online CD in the nonasymptotic regime*: indeed,
> - Online CD **without averaging** (Theorem 3.2) is sensitive to the choice of the initial step size: below $2 \tilde{\mu}_m$, the convergence rate is smaller than $O(n^{-1/2})$. Above $2 \tilde{\mu}_m$, the bound contains a transient higher-order term which scales exponentially with the smoothness $L$ of the log-likelihood gradient and linearily with the initialization $\sqrt \delta_0$. This higher-order term can affect the performance of the estimator in the non-asymptotic regime.
> - Similarly, for online CD **with averaging**, the $o(1/n^{-1/2})$ term in Theorem 3.3 contains multiple higher-order terms (See Lemma D4, D.5, D.6) that scale exponentially with $L$ and other smoothness parameters of the problem, and linearily with $\sqrt \delta_0$.
> - Offline CD (Theorem 4.2) also contains similar transient terms as in the online case. However, in contrast to online CD, these terms vanish for a *fixed* number of samples $n$ by making the number of gradient steps $T \to \infty$. In Lemma 4.3, for instance, the only remaining term is $C'(\kappa_m, p, \nu, m, \Psi, \beta)(\sqrt{\log(n) / n} + 1 / \sqrt{n})$ (assuming $B=n$), with  (see Lemma B.3, full statement)
>     \begin{align*}
>     C'(p,\nu,m,\Psi,\beta) = \frac{8(1+ 5 \sigma + 5\kappa_m)}{\tilde \mu_m} \times
>     \Big( 3 \sigma_m \sqrt{p(p+\nu-2)} +  \kappa_m
>     (2 r_\Psi^p )^{\frac{1}{\nu-2}} \Big( 1 + \frac{2 C}{\sigma_m (p^2 + (\nu-2)p)^{1/2}} \Big)^{\frac{p}{\nu-2}}
>     \Big)
> \end{align*}
> For Markov kernels with light enough tails, we can set $\nu - 2 = 2 p$. Considering the worst-case initialization scenario  $\delta_0 = r_{\Psi}^2$, we obtain
> \begin{align*}
>     C'(p,\nu,m,\Psi,\beta)= \frac{8(1+ 5 \sigma + 5\kappa_m)}{\tilde \mu_m} \times \Big( 3 \sigma_m \sqrt{p(p+2p)} +  \kappa_m 2^{\frac{1}{2p}}\delta_0^{\frac{1}{4}} \Big( 1 + \frac{2 C}{\sigma_m (p^2 + 2pp)^{1/2}} \Big)^{1/2} \Big).
> \end{align*}
>
> Compared with the online CD bounds we see multiple benefits:
> - This bound depends only sub-linearily on $\sqrt{\delta_0}$, unlike some of the constants of the online CD bounds, which depend linearily on it. Moreover, for subexponential $k_\psi^m$, the dependence can be arbitrarily weakened to $\delta_0^{1/2a}$ by setting $\nu - 2 = a p$ for some $a > 2$. Offline CD can thus be provably less sensitive to the effect of initialization than online CD, in moderate sample size regimes where $\sqrt{\log n / n} \simeq 1/\sqrt{n}$ and higher-order terms still matter.
> - Moreover, with such choices, $C'$ only depends polynomialy on the parameters of the problem and is independent of L. This is in contrast with the constants of online CD, which contain terms that exponentially grow with the initial learning rate $C$ and $L$, translating the fact that a too large $C$ can lead to a transient exponential blowup of the loss. By allowing $T \to \infty$, offline CD thus prevents this possible exponential blowup.
>
>
> In summary, in subexponential tail scenarios, offline CD allows to control smoothness and initialization related transient terms of the online CD bounds at the cost of increasing compute ($T \gg n$, compared to $T=n$ for online CD), and asymptotic suboptimality. This is one reason why we believe this result is a valuable addition to the paper, despite the technicality of the proof. While alluded to in the end of Section 3, we agree that this argument is not presented with sufficient clarity the paper, and we thank the reviewer for challenging us in that regard. To address this, we will highlight the dependency of the higher-order term of Theorem 3.3 in the parameters of the problem, and will make a thorough comparison between the online and offline CD bounds in Section 4, as done above. We welcome further discussion with the reviewer on this topic.
>
>
> **Regarding the clarity of section 4**: We agree that Section 4 is technical, and that it is crucial to increase its accessbility to a broad readership. To this end, we will take the following actions:
> - Introduce Lemma 4.3 (which is our strongest final result for offline CD) as the main result (Theorem) of Section 4. Consequently, the tail probability terms and Theorem 4.2 will only appear as part of a proof sketch for this Lemma.
> - Delegate Lemma 4.4 to the appendix, with a mention in the main paper.
> - Rely on the extra page for camera-ready papers to add proof sketches for section 3 and 4  and summary tables for rates (provided to reviewer Fohj).
>
>
> **Other questions**
> - We thank the reviewer for spotting the incorrect definition of $\mu$, and will fix this in the updated version of the paper. The extreme-value theorem argument justifying its existence still applies, thanks to [56].
> - The reviewer is correct on the origin of the inflated factor of 4 in our bound. We will add a discussion about it in the paper.
>
> [56] G. Harris and C. Martin. The roots of a polynomial vary continuously as a function of the coefficients. Proc. Amer. Math. Soc. 100 (1987), no. 2, 390–392.
>
> ---
>
> We have done our best to address your questions, and we would be happy to address further questions in the discussion. If we have done so, we would be grateful if you would consider increasing your score.

---

> > ### Comment · Reviewer_ouZy · 2024-08-13
> > **Response**
> >
> > Thanks for addressing my concerns. I appreciate the authors's efforts on improving the presentation of Section 4. I have the following questions:
> > - The drawbacks of online CD (dependence on initial distance, exponential high-order terms) can be overcome by more careful choice of stepsize and tail averaging techniques. These are not discussed and the comparison to offline CD is unfair.
> > - For offline CD, it would be helpful to make it clear what the rate, leading-order term, and high-order terms are, and whether they're optimal.
> > I think there could be a publishable paper here, but the current version looks unclear. So I'll keep my score.

---

> > > ### Author Response · Authors · 2024-08-13
> > > **Response to reviewer's questions**
> > >
> > > We thank the reviewer for their response. We answer their questions below.
> > >
> > > **Regarding our comparison between online and offline CD, and its fairness**:
> > >
> > > - First, the bounds provided in our paper are **the only currently existing nonasymptotic bounds for the CD algorithm** (and the contribution of our work). We do not believe that comparing these two bounds (and stating when offline CD is likely to be more efficient than online CD) as done in the paper and in the rebuttal is unfair to online CD. Note that we take into account all step size schedules of the form $Ct^{-\beta}$ in our comparison.
> > > - Second, while adapting the guarantees of **other** online unbiased SGD algorithms to the CD setting is an interesting avenue for future work (and we are happy to mention that in the paper), the bounds are **not** expected to transfer without fundamental modifications from unbiased SGD to CD: for instance, the tail-averaging algorithm of [58] (Equation 6a) contains averaging **in its update rule**, which may introduce CD-specific correlations that are precisely the ones requiring a tail control as done in Section 4, and that lead to the additional $\log (n)$ factor in the rate. The rate of such variants may thus be **worse** than (averaged) online CD. Offline CD, as we show, provides a simple optimization algorithm with an optimal (up to a log factor) convergence rate (sharply improving over [21]) and that is robust to multiple hyper-parameters of the problem, unlike averaged online CD. Trying to extend the guarantees of other online SGD algorithms to the CD setting would be interesting for future work, but the absence of such results in our paper does not take away the value of our analysis of offline CD, and its benefits compared to online CD.
> > >
> > > **Regarding the rates, leading order terms, and high-order terms of offline CD**: all such quantities are given in the current manuscript. The rates (depending on the setting) are given in Lemma 4.3 and Lemma 4.4 of the main body. The leading order terms are made explicit (to keep the main body simple) in Lemma B.3 and B.4 of the appendix. The higher order terms are given in Theorem B.2 of the appendix, but importantly vanish for fixed $n$ as $T \to \infty$. The rate is optimal up to a $\log(n)$ factor, while the multiplicative constant may be improvable. We will mention these points and will make clearer references to these results in the main body.
> > >
> > > We thank the reviewer for engaging with us and our manuscript. We are happy to answer any remaining questions they may have.
> > >
> > > [58] ROOT-SGD: Sharp Nonasymptotics and Asymptotic Efficiency in a Single Algorithm, Li et. Al

---

### Official Review · Reviewer_yRNH · 2024-07-21

**Soundness:** 3
**Presentation:** 3
**Contribution:** 2
**Rating:** 6
**Confidence:** 2

**Summary:**

The paper delivers a detailed non-asymptotic analysis of the Contrastive Divergence (CD) algorithm applied to unnormalized exponential family distributions. It significantly advances the understanding of CD by showing that, under certain regularity conditions, the algorithm can achieve the parametric convergence rate of \(O(n^{-1/2})\). This finding represents a notable improvement over the previously established convergence rate of \(O(n^{-1/3})\). The paper's analysis is rooted in a meticulous exploration of how CD's bias and variance components affect its convergence behavior, providing deeper insights into the algorithm's performance under various conditions.

**Strengths:**

- The paper offers a thorough theoretical analysis, providing new insights into the convergence behaviors of CD, particularly highlighting how CD approaches the true data distribution parameters under different data batching schemes, including fully online and minibatch settings.
- By establishing that CD's asymptotic variance is close to the Cramér-Rao lower bound, the paper positions CD as a near-optimal method for training unnormalized models, which are prevalent in fields such as image and text processing, protein modeling, and neuroscience.
- The introduction of new primal-dual treeplex norms and a novel regret lower bound specifically tailored for sequence-form strategy spaces in online learning scenarios represent significant theoretical advancements.

**Weaknesses:**

The paper focuses heavily on theoretical analysis with less emphasis on empirical validation. Including comprehensive experiments that benchmark CD against other estimation techniques across various real-world datasets could strengthen the claims.

The paper focuses heavily on theoretical analysis with less emphasis on empirical validation. Including comprehensive experiments that benchmark CD against other estimation techniques across various real-world datasets could strengthen the claims.

**Questions:**

Could you provide more detailed explanations or examples of the "specific regularity conditions" under which CD achieves the improved convergence rate of $O(n^{-1/2})$?

Are there plans to complement the theoretical findings with empirical validations?

**Limitations:**

The authors listed the limitations and future directions in their discussion section

---

> ### Author Rebuttal · Authors · 2024-08-07
>
> We thank the reviewer for their helpful comments. We address their concerns below.
>
> **Regarding the regularity conditions required to achieve the $O(n^{-1/2})$ rate** One of the assumptions made in our theorems regarding online CD  is that the parameter space $\Psi$ is compact. Under this assumption, the log-partition function is smooth (e.g. $C^\infty$) and with bounded derivatives. This boundedness is what we mean by "regularity conditions", and is critical to obtain our upper bound on the variance of the averaged online CD iterates (see Lemma D.1 and D.2 of the appendix).
>
> **Regarding empirical validation** We thank the reviewer for their suggestions. We agree that experimental validation of the theory is helpful and will add a synthetic experiment where the ground truth is known in order to exemplify the claims made in our theory.
> We follow the setup of [21, Section 4.1] and consider the following model:
> \begin{align}
>     p(x; \psi) = \mathcal  N(\psi, \Sigma), \quad \quad \Sigma = \begin{bmatrix} 1 & 0.5 \\\\ 0.5 & 1 \end{bmatrix}
> \end{align}
>
> for $\Psi = \left \lbrack -0.5, 0.5 \right \rbrack^2$, which is an exponential family for $\phi(x) = \Sigma^{-1} x$, $\log Z(\psi) = \frac{1}{2} \psi^T \Sigma^{-1} \psi$,  implying $\nabla_{  }^2 \log Z(\psi) = \Sigma^{-1}$. It can be shown (see discussion below) that assumptions A1-A6 are verified for this model. While other, more complicated models could be shown to verify our assumptions, for this specific model, the quantities involved in the bounds can be computed explicitly (or upper bounded) explicitly, which allows us to validate our bounds.
>
> **Experimental setup** We run 500 different experiments with different seeds to compute an empirical estimate of $\delta_n = \mathbb{ E } \| \widehat{ \psi }_n - {\psi}^{\star} \|^2$ up to $n = 2000$ samples, where $\widehat{ \psi }_n$  is the estimate of $\psi^{\star}$ obtained by the online CD, online CD with averaging, and offline CD estimators, respectively. We then plot this empirical estimate as a function of the number of samples $n$ , and compare it to both our upper bounds of our paper obtained in each setting, as well the Cramer-Rao lower bound. Our learning rate schedule is $C n^{-\beta}$, with $C = 0.1$ and $\beta = 0.6$. For offline CD, we make $5000$ gradient steps using the full dataset, and run the experiments for 10 evenly log-spaced values of $ n $ between $ 100 $ and $2000$. We use $m =20$ MCMC steps.
>
>
> **Plotting lower and upper bounds** In the pdf included in the general comment, we plot:
> - The full non-asymptotic upper bound related to the last-iterate online CD estimator
> - The leading term of the non-asymptotic upper bound related to the averaged online CD estimator (for simplicity, due to the complexity of the higher-order terms)
> - The non-asymptotic in $n$, but asymptotic in $T$ upper bound related to the offline CD estimator (for similar reasons).
>
> In this example, we see that the bounds for online CD are tight. While the rate of offline CD is near-optimal (and thus tight), the bounds are looser than in the online case, which is due to the multiplicative constant in front of the leading term. However, our bounds for offline CD are still able to provably explain how offline CD is more robust to transient effects related to bad initialization and smoothness, as mentioned in the rebuttal to reviewer ouZy. Moreover, we see that offline CD performs competitively compared to online CD, which calls for more investigations of the offline setting. Further work will investigate how to tighten these bounds by, in particular, attempting to improve the constants.
>
>
> **Additional details for the model**
> It can be shown [21] that $\sigma^2 = 2.67$, $\mu = 0.67$, and $C_{\chi} \approx 1.84$, and that $\alpha \leq 1 - \lambda_{\min_{  }}(\Sigma^{-1} / 2) = 0.67$. For $m \geq  \lceil 3.5 \rceil = 4$, Theorem 3.2 will hold. Using the formulas present in the proofs, and the formulas for the cumulants of a Gaussian random variable, we can show that $||\log Z||_{3, \infty} \approx 10.9$.
> For online CD with averaging, we also need $m(n) > \frac{(1 - \beta) \log n}{2\lvert \log \alpha \rvert }$. Assuming $n = 2000$ and $\beta =0.6$, we need $m(n) > 3.8$, which will hold for $m(n) \geq 4$. Finally, for offline CD, Assumption A4 to A6 will be satisfied [21].
>
> It can be shown that $k_{\psi}^{m}(X)$ is a Gaussian random variable with mean $\psi ( 1 - \rho^{2m}) + {\psi}^{\star}$ and (component-wise) variances $1$, yielding (using formulas for the centered moments of Gaussian random variables and $(a^2 + b^2)^3 \leq 4 (a^{6} +  b^{6})$), we obtain $\kappa_{m, 6} \leq \lambda_{\max_{  }}(\Sigma ^{-1}) (\mathbb{ E } \| k_{\psi}^{m}(X) - \mathbb{ E } k_{\psi}^{m}(X) \|^{6})^{1 / 6} \leq 4 \cdot 12^{1 / 6} \approx 4.6$. Finally, it can also be shown that $\mathbb E [ k_{\psi}^{m}(X) | X]$ is Gaussian with mean $\psi ( 1 - \rho^{2m}) + \rho^{2m}X$ and component-wise variances $1 - \rho^{4m}$. Straightforward derivations can show that assumption A6 is verified for $\sigma_m = 2$.
>
> ---
>
> We have done our best to address your questions, and we would be happy to address further questions in the discussion. If we have done so, we would be grateful if you would consider increasing your score.

---

> ### Comment · Reviewer_yRNH · 2024-08-14
>
> Thank you for the response. I have read the rebuttal and decide to keep my score.

---

### Official Review · Reviewer_Hdtk · 2024-07-22

**Soundness:** 4
**Presentation:** 4
**Contribution:** 2
**Rating:** 5
**Confidence:** 3

**Summary:**

This paper studies the problem of finding a maximum-likelihood estimator for an exponential family. The paper proves guarantees for the Contrastive Divergence (CD) algorithm, which iteratively performs stochastic gradient descent on the cross-entropy loss between the estimated model and empirical data distribution. However, each step of gradient descent is performed by estimating the gradient with an MCMC algorithm.

The paper proves:
* Thm 3.2: Online CD (i.e., using a fresh sample on each step), with enough MCMC steps, leads to $O(n^{-1/2})$ error rate with $n$ samples
* Thm 3.3: Online CD with Polyak-Ruppert averaging achieves (up to a constant factor of <= 2), an asymptotically optimal rate.
* Thm 4.2: Offline CD (i.e., using a batch of samples at each step) leads to $O((\log(n) / n)^{1/2} + 1/\sqrt{B})$ error rate with batch size B.

**Strengths:**

This paper improves on the results of Jiang et al. [21], which had previously shown a weaker rate under stronger assumptions. The math that I checked appears correct, and the paper is very well written and presented. Many variations of CD (online, online with averaging, and offline) are analyzed. The best rates possible are also proved.

**Weaknesses:**

My main concern is on whether the subject of this paper will be of sufficient interest to a large enough portion of the NeurIPS audience, or whether a different venue/journal with a more statistical audience might be more appropriate. The
results of this paper are technical bounds on how CD converges for maximum-likelihood estimation in exponential families; and the CD algorithm is not (currently) a focus of the machine learning community. Furthermore, the rate improvement proved is relatively modest -- from the previously known $O(n^{-1/3})$ rate in Jiang et al. [21] to the current best possible $O(n^{-1/2})$ rate, so to some extent the qualitative nature of the result was previously known.

Very minor:
* Assumption A.4, which is needed for Theorem 4.2 is stronger than what Jiang et al. [21] assumes (see Remark 4.1). But Theorem 4.2 is stronger than what is proved in Jiang et al. [21].
* The bound in Line 23 of Theorem 3.2 is quite elaborate and difficult to check/use. However, the most important case of $\\beta = 1$ is conveniently solved.
* On line 95, the idiom “silver lining” is used incorrectly. It should indicate something positive, rather than something negative.

**Questions:**

* What is $\\tilde{\\mu}$ in Theorem 3.2? Do you mean $\\tilde{\\mu}_m$?

**Limitations:**

Yes

---

> ### Author Rebuttal · Authors · 2024-08-07
>
> We thank the reviewer for their comments and in particular for challenging us regarding the significance of our work and its relevance to the NeurIPS community, which will help better position our contributions. We explain further the relevance and significance of our results below. We will make these points clearer in the camera-ready version of the paper.
>
> **On the relevance of our work to the NeurIPS community**
>
> - The broader question our work addresses is the following: *What are the statistical estimation methods that are **sample efficient**, e.g. that require as few samples as possible to reach a good accuracy*? We believe that this question resonates which a sizeable part of the NeurIPS community, especially in fields where samples are expensive to acquire (making sample efficiency crucial). An example of such fields is **Simulation-Based Inference** (SBI), a set of methods performing parameter inference in scientific models with intractable likelihoods. SBI is popular in multiple scientific disciplines such as particle physics and neuroscience, and has attracted significant attention in the NeurIPS community. The data space is challenging, yet often lower-dimensional than domains like images (which are now indeed dominated by other estimation methods like diffusion models), and acquiring samples often requires running expensive simulations, making sample-efficient methods particularly important. Interestingly, [20] designed an SBI method using CD which achieves state-of-the-art sample efficiency for multiple problems. Given such facts, we believe that the results established in our work work are relevant to inform modern machine learning applications and to the NeurIPS community in general.
> - Moreover, we believe that the sample-efficiency properties of CD established in our work yield an important conclusion: in exponential families, CD is near-asympotically **statistically** efficient, which shows the potential of CD in statistical estimation. In order for CD to be **computationally** efficient and thus usable in practice, the number of MCMC steps $m$ should not be too large: this calls for the design of fast-mixing, model-dependent Markov kernels, thanks to which CD may be shown to be an end-to-end efficient estimation technique.
>
>
> **On the significance of the theoretical improvements**
>
> - With the improvement from the $O(n^{-1/3})$-consistency shown in [21] to the optimal $O(n^{-1/2})$ rate established in our work, we place CD at the same level as other well-known statistical estimation methods for unnormalized models like Noise-Contrastive estimation and score matching, which both enjoy $O(n^{-1/2})$ consistency. Thanks to this new bound, CD is now proved to be asymptotically competitive with such methods, making CD a theoretically valid alternative in large data regimes. This conclusion is only possible because of our new bounds, and could not have been reached from the results of [21]. We thus believe that this rate improvement is important.
> - In addition to improving the **rate** of convergence of CD, we also show that the asymptotic variance of averaged online CD (e.g the multiplicative constant associated with the leading $n^{-1/2}$ term) is optimal up to a factor 4. This result improves over the asymptotic variance of score-matching (which includes other problem-dependent smoothness factors and a suboptimal Fisher information exponent [8]) , and over the  Noise-Contrastive Estimation result (which requires knowing the unknown data density to reach similar results [6], and otherwise can scale exponentially poorly with dimension [5]). Analyses of asymptotic variances are common in the machine learning literature, have important empirical implications regarding the sample-efficiency of the associated methods. We believe that our results complement other known results in an informative manner. We are happy to make these points clearer in the camera-ready version of the paper.
>
> **Regarding assumption A4:** The reviewer is correct. We we provide some additional comments regarding cases where A4 is satisfied for $\nu = 2$ (which is needed to obtain the asymptotic consistency of [21]) but not for any $\nu > 2$ (which is needed for our results to hold). In such cases, the Markov kernel is such that $\phi(K^m_\psi(x))$ has only finite second moment and unbounded higher moments, which has a much heavier tail than most typical practical regimes (for example, this implies infinite kurtosis). In particular, violation of A4 is a known pathological regime in the Central Limit Theorem literature, where the empirical average of these random variables converges to a normal distribution, but obtaining finite-sample bounds is difficult (see e.g. [57]). Since, unlike [21],  our results involve finite-sample controls on sample averages of gradients in a much more complicated optimization regime, we do not expect to be able to circumvent such known difficulties. One may indeed try to relax the bounded $\nu$-th moment assumption by a more involved analysis of the dependence, but we believe that our assumption  suffices for many practical applications: Indeed, A4 for any $\nu > 2$  will be satisfied for Markov kernels with subexponential tails. Moreover, the tail of a Markov kernel can often be checked based on its specific form.
>
> We apologize that these points were not made sufficiently clear in the submission, and will add these comments in the camera-ready version  for contextualizing A4.
>
> [57] Friedman, N., Katz et. al Convergence rates for the central limit theorem. PNAS
>
> **Minor** We also thank the reviewer for pointing out some minor inaccuracies. In theorem 4.2, we $\tilde{\mu}$ was indeed supposed to be $\tilde{\mu}_m$. We will correct the use of "silver-lining".
>
> ---
>
> We have done our best to address your questions, and we would be happy to address further questions in the discussion. If we have done so, we would be grateful if you would consider increasing your score.

---

> > ### Comment · Reviewer_Hdtk · 2024-08-12
> > **Response**
> >
> > Thank you for addressing my concerns. I am still hesitant about NeurIPS being a good audience, but I will raise my score since I would not be upset if this paper gets accepted.

---

### Author Rebuttal · Authors · 2024-08-07

We thank the reviewers for their comments, which were very helpful to improving the current manuscript. We have addressed the concerns of each reviewer in their respective threads. We provide below a summary of the most common reviewer's comments, and of our actions taken to address them.


**Regarding offline CD and Section 4** Multiple reviewers (ouzY, Fohj, Hdtk) found Section 4 (which contains the bounds for offline CD) difficult to follow because of the technicality of the theorems. To address this issue, we will simplify Section 4 by taking the following steps:
- We will cast lemma 4.3 (The strongest bound we achieved in this section) as the main result of the section, making it a Theorem. Theorem 4.2 and the introduction of the tail probability term preceding it will be discussed as part of the proof sketch of Lemma (Theorem to-be) 4.3. We will also delegate Lemma 4.4 (which achieves worst upper bounds, but requires fewer assumptions) to the appendix.
- We will provide an extended comparison between the rates achieved in the online case and those achieved in the offline case. The main takeaway, as presented in detail in the response to Reviewer ouzY, is that while the bounds of (averaged) online CD are asymptotically near optimal, and asymptotically surpass those of offline CD, the bounds of offline CD can be tighter than the online ones in the non-asymptotic regime, where transient terms present in the online CD bounds that scale with the smoothness and initialization parameters can be dominant. In contrast, the transient terms present in the offline CD bounds can be made arbitrarily small by increasing the number of gradient steps $T$, while the dominant term remains robust to smoothness and initialization.
- Finally, we will rely on the extra page for camera-ready papers to add proof discussions for section 3 and 4, and a summary table for rates. We provided both proof discussion and tables to reviewer Fohj.

**Regarding empirical validation** Multiple reviewers (yRNH, b7nK) have suggested to add an experimental validation of the bounds, which is currently absent in the paper. To address this, we have added an experiment showing the behavior of online and offline CD for a Gaussian model (described in detail to reviewer yRNH) which we show satisfies all our assumptions. While other (more complicated) models and samplers would satisfy our assumptions, all constants involved in the assumptions can be explicitly computed (or upper bounded) for this model. In each setting (online, online averaged, and offline), we plot the behavior the empirical version of $\mathbb E \| \hat{\psi}_n - \psi^\star \|^2$ compared to our upper bounds and the lower bounds. A more thorough discussion and comparison of the methods is made in the rebuttal to reviewer yRNH.

We have done our best to address the reviewer's concerns, and welcome further discussion with the reviewers if needed.

---

### Decision · Program_Chairs · 2024-09-25

**Decision:**

Accept (poster)

**Comment:**

This paper studies the maximum-likelihood estimator for an exponential family. It provides guarantees for the Contrastive Divergence algorithm, which uses stochastic gradient descent on the cross-entropy loss between the model estimate and empirical data.
The reviewers found the presentation clear and the contribution valuable for the NeurIPS community. I recommend accepting this paper